# PRISM: Distribution-free Adaptive Computation of Matrix Functions for Accelerating Neural Network Training

**Shenghao Yang** [1 2]  **Zhichao Wang** [1 2]  **Oleg Balabanov** [1 2]  **N. Benjamin Erichson** [1 3]  **Michael W. Mahoney** [1 2 3]

## Abstract

Matrix functions such as square root, inverse roots, and orthogonalization play a central role in preconditioned gradient methods for neural network training. This has motivated the development of iterative algorithms that avoid explicit eigendecompositions and rely primarily on matrix multiplications, making them well suited for modern GPU accelerators. We present PRISM[1] (Polynomial-fitting and Randomized Iterative Sketching for Matrix functions computation), a general framework for accelerating iterative algorithms for computing matrix functions. PRISM combines adaptive polynomial approximation with randomized sketching: at each iteration, it fits a polynomial surrogate to the current spectrum via a sketched least-squares problem, adapting to the instance at hand with minimal overhead. We apply PRISM to accelerate Newton–Schulz-like iterations for matrix square roots and orthogonalization, which are core primitives in machine learning. Unlike prior methods, PRISM requires no explicit spectral bounds or singular value estimates; and it adapts automatically to the evolving spectrum. Empirically, PRISM accelerates training when integrated into Shampoo and Muon optimizers.

## 1. Introduction

Matrix functions are extensively used in scientific and engineering applications, and they are of increasing interest in machine learning (ML). Applications range from computational fluid dynamics (Ndjinga, 2008; Castro et al., 2016)

and computational chemistry (Lin et al., 2009b;a) to optimal transport (Janati et al., 2020; Minh, 2022), Gaussian processes and Bayesian inference (Mallasto & Feragen, 2017; Pleiss et al., 2020), uncertainty quantification (Chen et al., 2019), computer vision (Wang et al., 2021; Song et al., 2021; 2023), and fast optimizers for training deep neural networks (Carlson et al., 2015; Gupta et al., 2018; Yao et al., 2020; 2021; Jordan et al., 2024; Ahn et al., 2025). As such, the ability to compute simple matrix functions, such as square root, inverse root, and orthogonalization (polar decomposition), in a fast and numerically stable manner, can lead to substantial improvements.

Among the many alternatives for computing matrix functions, iterative algorithms that rely primarily on General Matrix Multiplications (GEMMs) are particularly attractive for GPU-accelerated computing environments (Volkov & Demmel, 2008; Markidis et al., 2018; Yan et al., 2020; Amsel et al., 2026; Grishina et al., 2025). Compared with algorithms based on the singular value decomposition (SVD), or those that involve some form of matrix inversion, GEMMs have much better scaling with respect to the size of a matrix. Because of this, iterative algorithms of the form $\boldsymbol{X}_{k+1} = F_k(\boldsymbol{X}_k)$, where computing $F_k(\boldsymbol{X}_k)$ is fast on accelerators, have recently received increasing interest (Amsel et al., 2026; Grishina et al., 2025; Kim et al., 2025). A simple example is to restrict $F_k$ to be a polynomial $p_k^{(d)}$ of fixed degree $d$ at $k$-th iteration, in which case evaluating $F_k(\boldsymbol{X}_k) = p_k^{(d)}(\boldsymbol{X}_k)$ only requires GEMMs.

This general approach has received interest in ML due to the effectiveness of the Muon optimizer (Jordan et al., 2024) for training neural networks. Subsequent work (Amsel et al., 2026; Grishina et al., 2025) explored the acceleration of the initial convergence of Newton-Schulz-like iterative methods for the polar factor $\boldsymbol{U}\boldsymbol{V}^T$ of a matrix $\boldsymbol{A}$, where $\boldsymbol{U}$ and $\boldsymbol{V}$ consist of the left and right singular vectors of $\boldsymbol{A}$, respectively. In particular, for $d \in \{3, 5\}$ and any $K \geq 1$, they showed how to construct a polynomial $p^*$ by composing degree-$d$ polynomials such that:

$$p^* = \underset{p = p_K^{(d)} \circ p_{K-1}^{(d)} \circ \cdots \circ p_1^{(d)}}{\arg\min} \ \underset{\substack{\boldsymbol{A} \in \mathbb{R}^{m \times n}: \\ \boldsymbol{\sigma}(\boldsymbol{A}) \subseteq [\ell, u]}}{\max} \|p(\boldsymbol{A}) - \boldsymbol{U}\boldsymbol{V}^T\|_2,$$

where $\boldsymbol{\sigma}(\boldsymbol{A})$ denotes the set of singular values of $\boldsymbol{A}$. Thus,

---

[1]International Computer Science Institute, Berkeley CA, USA [2]University of California, Berkeley CA, USA [3]Lawrence Berkeley National Laboratory, Berkeley CA, USA. Correspondence to: Shenghao Yang <shenghao.yang@berkeley.edu>, Zhichao Wang <zhichao.wang@berkeley.edu>.

*Proceedings of the 43rd International Conference on Machine Learning*, Seoul, South Korea. PMLR 306, 2026. Copyright 2026 by the author(s).

[1]Code is available at https://github.com/s-h-yang/PRISM

if the largest and smallest singular values of $A$ are known a priori (which, in general, is not the case), the methods proposed by Amsel et al. (2026) and Grishina et al. (2025) provide optimal convergence with respect to the spectral norm error $\|X_k - UV^T\|_2$. In addition to polar decomposition, Kim et al. (2025) considers computing matrix roots and inverses via a Monte-Carlo Tree Search (MTCS) method to construct iterative algorithms of the form $X_{k+1} = r_k(X_k)$, where $r_k$ is either a polynomial (i.e., Newton-Schulz-like) or a rational function (i.e., Newton-like); and, when the underlying distribution of singular values is known, they demonstrated strong performance.

While promising, these recently-introduced methods suffer from several disadvantages, which currently limit the broader applicability of this approach. First, they tend to be solved for a single problem, rather than for a broader class of problems, as is more common in numerical analysis (Higham et al., 2005). In ML, they are just applied and evaluated for one specific use case, e.g., within the Muon optimizer where one only needs to compute the polar factor of the gradient matrix. This can lead to strong performance in one setting, but it obscures the broader applicability of the methodology, and it prevents a "cut-and-paste" approach of using this methodology to new problem classes and application domains. Second, these recently-introduced methods tend to be parameterized in terms of parameters that are themselves as difficult to compute as solving the original problem. In particular, in practice, we typically do not have prior knowledge of the distribution of singular values or even a tight interval that contains them. Obtaining good estimates on the largest and, in particular, the smallest singular values of a matrix can be as costly as computing its polar factor by using the original Newton-Schulz method.

To deal with this, Amsel et al. (2026) suggested fixing the range of singular values to $[\ell, u] = [10^{-3}, 1]$, when the computations are carried out in half precision. However, if the actual range of singular values is much narrower or wider than a predefined interval, which can easily happen, then the convergence behavior can degrade dramatically. An example of this is shown in Figure 1, where we compare Newton-Schulz variants for computing polar factor (i.e., orthogonalization) and square root. Observe that PolarExpress (Amsel et al., 2026) can even *slow down* the convergence of the classical Newton-Schulz, if there is a mismatch between the tightest interval that contains the initial singular values and the interval for which the method is optimized.

These shortcomings highlight the need for a principled adaptive approach for a broader class of problems that effectively generates polynomials $p_k^{(d)}(\cdot; X_0)$ whose coefficients do not assume properties of the spectrum of its input matrix, but instead dynamically adjust to the spectrum. We introduce PRISM (**P**olynomial-fitting and **R**andomized **I**terative

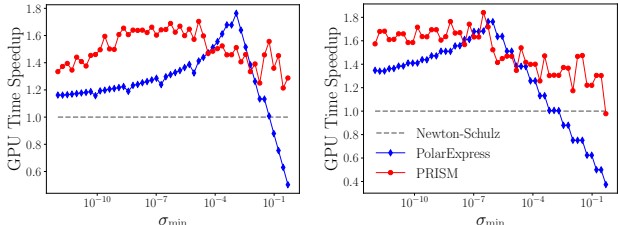

*Figure 1.* Speedup in GPU time over the classical Newton-Schulz for polar decomposition (left) and square root (right). We keep $\sigma_{\max} = 1$ and vary $\sigma_{\min} \in [10^{-12}, 1/2]$. The PolarExpress variant we use is optimized for $\sigma_{\min} = 10^{-3}$ for polar decomposition (and hence it is optimized for $\sigma_{\min} = 10^{-6}$ for square root). All algorithms are run until convergence. In both cases, we see a performance degradation as $\sigma_{\min}$ deviates from the one PolarExpress is optimized for. PRISM (this work) does not require or assume $\sigma_{\min}$ and has a stable speedup across the entire range.

**S**ketching for **M**atrix functions computation), a general framework for accelerating the computation of matrix functions via spectrum-adaptive polynomial updates (Chen et al., 2011) and randomized sketching (Murray et al., 2023).

PRISM targets GPU-friendly iterations and is designed to be broadly applicable, computationally efficient, and robust to variations in spectral structure.

**Contributions.** Summarizing our contributions:

- **General acceleration framework.** PRISM provides a unified, systematic approach for accelerating iterative algorithms for matrix functions, including Newton-Schulz methods for square roots, inverse roots, sign, and polar decomposition; Chebyshev method for the inverse; and inverse Newton for the inverse $p$-th root. See Table 1 for some iterative algorithms accelerated by PRISM.

- **Spectrum-adaptive without prior spectral knowledge.** PRISM dynamically fits polynomial updates to the evolving spectrum of the current iterate without requiring explicit eigenvalue or singular-value information. This leads to an instance-specific and *distribution-free* acceleration of classical iterative algorithms.

- **Efficient randomized polynomial fitting with guarantees.** Using randomized sketching, PRISM reduces the overhead cost of polynomial fitting to $O(n^2 \log n)$, which is nearly negligible compared to the $O(n^3)$ cost of matrix multiplications. At the same time, it preserves the convergence behavior of the underlying iterations, both theoretically and empirically.

- **Empirical validation in neural network optimizers.** We show that PRISM effectively accelerates Newton-Schulz-like algorithms for square roots and polar decomposition on various input matrices, including those with a Marchenko-Pastur law (e.g., neural network weight matrices at initialization) or a heavy-tailed distribution (e.g.,

pre-trained models (Martin & Mahoney, 2021; Hodgkinson et al., 2025)). When integrated into the Shampoo and Muon optimizers, PRISM efficiently accelerates training large neural networks in both cases.

## 2. Related Work

**Iterative algorithms for matrix function computation.** A large body of numerical linear algebra treats matrix functions via iterations that are dominated by matrix–matrix multiplies and thus well–suited to modern accelerators. Classical methods include Newton/Schulz and Padé-type schemes for matrix sign and polar decomposition (Kenney & Laub, 1991; Higham et al., 2004; 2005), stable iterations for the matrix square root and its variants (Higham, 1997), and later improvements using Zolotarev and Halley/QDWH-style rational approximants with careful stability analyses (Nakatsukasa & Freund, 2016; Nakatsukasa & Higham, 2012). These methods motivate designing polynomial or rational updates that (i) avoid explicit inverses or SVD, (ii) converge quickly over a prescribed spectral interval, and (iii) map cleanly to GEMM-dominant kernels (Fan et al., 2019; 2020; Arisaka & Li, 2023; Ndjinga, 2008; Kim et al., 2025).

**Randomized/sketching methods for matrix functions and traces.** Randomized numerical linear algebra (RandNLA) provides subspace embeddings and sketching primitives that reduce dimension, while approximately preserving the spectral structure (Mahoney, 2011; Woodruff, 2014; Drineas & Mahoney, 2018; Nelson & Nguyên, 2013). For quantities involving $f(A)$ or $\mathrm{tr}\, f(A)$, stochastic trace estimation and Krylov/Lanczos quadrature are now standard: Hutchinson/Avron–Toledo and sharper sample-complexity bounds (Avron & Toledo, 2011; Roosta-Khorasani & Ascher, 2015; Cortinovis & Kressner, 2022), and SLQ for $\mathrm{tr}\, f(A)$ with strong empirical and theoretical support (Ubaru et al., 2017; Yao et al., 2020). Variance–reduced estimators, such as Hutch++, further improve sample complexity in the PSD case and beyond (Meyer et al., 2021). Recently, Huang et al. (2025); Refael et al. (2025) applied randomized SVD to efficiently train neural networks.

**Newton–Schulz iteration in deep learning.** In large-scale training, GEMM-only iterations have reappeared as inner loops inside optimizers. Shampoo constructs Kronecker–factored preconditioners using (inverse) matrix square roots (Gupta et al., 2018); and AdaHessian uses randomized curvature (Hutchinson) to approximate Hessian–diagonals (Yao et al., 2021). More recently, *Muon* applied Newton-Schulz-based orthogonalization to momentum matrices to produce direction-only updates for matrix-shaped parameters (Jordan et al., 2024). Two follow-ups design iterations expressly for ML constraints: *PolarExpress* formulates a minimax-optimal polynomial update for the polar/sign problem with GPU-friendly stability and demonstra-

ble Muon gains (Amsel et al., 2026); and *CANS* (Chebyshev-Accelerated Newton–Schulz) uses Chebyshev polynomials and Remez optimization to accelerate early iterations and offer controlled approximate orthogonalization (Grishina et al., 2025). At scale, *DION* shows how to efficiently distribute orthonormalized updates in data-parallel/FSDP systems (Ahn et al., 2025). This line of work motivates adaptive, spectrum-aware polynomial updates that retain the hardware efficiency prized in ML training. However, all of these prior methods only focus on polar decomposition.

**Other ML applications of iterative matrix functions.** Beyond optimization, matrix functions (especially $A^{1/2}$ and $A^{-1/2}$) recur in areas such as computer vision and probabilistic ML. In global covariance pooling and second-order layers, Newton–Schulz/Taylor or Padé-based approximations are competitive with or superior to SVD in speed and accuracy (Li et al., 2018; Song et al., 2021; 2023). In Gaussian–process and Bayesian–optimization pipelines, fast actions of $K^{\pm 1/2}$ via iterative quadrature provide scalable alternatives to dense factorizations (Pleiss et al., 2020). Whitening/Coloring transforms for universal style transfer and de-correlated batch norm similarly rely on differentiable square roots/inverse square roots (Li et al., 2017; Huang et al., 2018; Song et al., 2023). Finally, Riemannian optimization on the Stiefel manifold uses polar-based retractions whose inner loops are identical to the iterations studied here (Absil et al., 2008; Grishina et al., 2025).

## 3. The PRISM Meta-algorithm

We now lay out the high-level structure of PRISM in the form of a meta-algorithm. The PRISM meta-algorithm starts by framing the design of existing or new algorithms as that of iterative polynomial approximation (Part I: Basic setup). Then, a principled acceleration scheme naturally arises, in which randomized sketching is used to efficiently improve polynomial approximation, leading to accelerated convergence behavior (Part II: Acceleration). A variety of classical iterative algorithms for matrix functions fit into the basic setup of PRISM. Consequently, all of these algorithms can be accelerated by deploying the acceleration techniques outlined in Part II of PRISM. In the next paragraph, we demonstrate how Newton-Schulz iterations are derived by following Part I of PRISM. Analogous derivations for Newton (Appendix A.2), Inverse Newton (Appendix A.3), and Chebyshev (Appendix A.4) methods are deferred to the appendix.

**Deriving Newton-Schulz with PRISM Part I.** Let $x \neq 0$ be such that $\mathrm{sign}(x) = \mathrm{sign}(a)$. Then, $\mathrm{sign}(a) = \mathrm{sign}(x) = x(x^2)^{-1/2} = x(1 - \xi)^{-1/2} = xf(\xi)$, where $f(\xi) = (1 - \xi)^{-1/2}$, and where $\xi = 1 - x^2$ measures how close $x$ is to $\mathrm{sign}(x) = \mathrm{sign}(a)$. Therefore, the problem of approximating $\mathrm{sign}(a)$ leads to that of approximating $f(\xi)$.

**PRISM Meta-algorithm for Computing $T(\boldsymbol{A})$**

*Part I: Basic setup*

1. Let $x$ be an estimate of $T(a)$ where $a \in \mathbb{R}$. Write $T(a) = x f(\xi)$ for a residual function $\xi = \xi(x, a)$.

2. Set up the scalar iteration $x_{k+1} = x_k \cdot f_d(\xi(x_k, a))$, where $f_d(\xi)$ is the $d$-th order Taylor's expansion of $f(\xi)$ around $\xi = 0$.

3. To compute $T(\boldsymbol{A})$, run the matrix version,
$$\boldsymbol{X}_{k+1} = \boldsymbol{X}_k f_d(\boldsymbol{R}_k),$$
where $\boldsymbol{R}_k = \xi(\boldsymbol{X}_k, \boldsymbol{A})$ is the residual matrix.

*Part II: Acceleration*

4. **Polynomial fitting:** To accelerate convergence, replace $f_d$ with $g_d(\xi; \alpha) = f_{d-1}(\xi) + \alpha \xi^d$, iterate
$$\boldsymbol{X}_{k+1} = \boldsymbol{X}_k g_d(\boldsymbol{R}_k; \alpha_k^*),$$
where $\alpha_k^* = \arg\min_\alpha \|\xi(\boldsymbol{X}_{k+1}, \boldsymbol{A})\|_F^2$ minimizes the residual norm.

5. **Sketching:** To maintain low cost at every iteration, use $\tilde{\alpha}_k = \arg\min_\alpha \|\boldsymbol{S}\xi(\boldsymbol{X}_{k+1}, \boldsymbol{A})\|_F^2$ in place of $\alpha_k^*$, where $\boldsymbol{S}$ is a low-dimensional sketch matrix.

Using the $d$-th order Taylor polynomial $f_d(\xi)$ around $\xi = 0$, we obtain an iterative procedure $x_{k+1} = x_k f_d(\xi(x_k))$. The matrix version, $\boldsymbol{X}_{k+1} = \boldsymbol{X}_k f_d(\boldsymbol{R}_k)$, where $\boldsymbol{R}_k = \boldsymbol{I} - \boldsymbol{X}_k^2$, gives the generalized Newton-Schulz iteration for matrix sign. When $d = 1$, this reduces to the standard one, $\boldsymbol{X}_{k+1} = \frac{1}{2}\boldsymbol{X}_k - \frac{3}{2}\boldsymbol{X}_k^3$, which convergences quadratically when initialized properly, e.g., when $\boldsymbol{X}_0 = \boldsymbol{A}/\|\boldsymbol{A}\|_F$. When $d \geq 2$, this leads to high-order variants with higher-order convergence rates. We note that this procedure was first described by Kenney & Laub (1991) to derive the more general Padé family of iterative rational methods for computing the matrix sign. In addition, due to the close relationship between iterative algorithms for computing the matrix sign, square roots, and polar factor (Higham et al., 2004; Higham, 1997), the Newton-Schulz variants for computing square roots and polar decomposition can be derived analogously.

The role of Part I in PRISM is to provide a common ground so that different algorithms for computing different matrix functions can all be accelerated in a similar way as outlined in Part II, which serves as PRISM's main algorithmic component. Once a new or existing iterative algorithm is fitted into Part I of PRISM, such as the Newton-Schulz iterations we discussed above, as well as additional examples in Appendix A, one may apply Part II of PRISM to accelerate convergence at low cost.

Since Taylor's polynomial may not provide the best approx-

imation of the target function at individual eigenvalues of the residual matrix $\boldsymbol{R}_k$, a poor fit can consequently result in a slow initial convergence of the corresponding algorithm. PRISM Part II directly addresses this in the following way:

- In order to improve convergence (across iterations) of the algorithm, Step 4 of PRISM replaces the Taylor polynomial with one that better fits to the spectrum. We will require that the residual matrix $\boldsymbol{R}_k = \xi(\boldsymbol{X}_k, \boldsymbol{A})$ be symmetric, and hence by minimizing the squared Frobenius norm, $\alpha_k^*$ effectively fits the candidate polynomial $g_d(\xi; \alpha)$ on the eigenvalues of $\boldsymbol{R}_k$ by minimizing a (nonlinear) least-squares loss. We defer an in-depth discussion of this to Section 4, where we focus on the particular example of matrix sign computation. We will also explain why the candidate polynomial $g_d(\xi; \alpha)$ was chosen to take that particular form.

- In order to speed up the run time (of each iteration) of the algorithm, in Step 5 of PRISM, we use randomized sketching methods from RandNLA to significantly reduce the cost of least-squares polynomial fitting. This is essential to ensure the practicality of PRISM: it accelerates convergence by automatically adapting to the spectrum–without requiring any knowledge on the spectral distribution of the input matrix–at comparably negligible additional cost. We will show that appropriately chosen sketch matrices do not compromise the performance.

## 4. Matrix Sign: a Case Study on How PRISM Accelerates Convergence at Low Cost

In this section, we describe how the PRISM meta-algorithm can be applied to develop an accelerated Newton-Schulz iteration for computing matrix sign. PRISM applies more broadly, but we start here with the matrix sign function in order to present the core ideas in a single setting, without deviating to small algorithmic or technical differences. The matrix sign function is particularly interesting because iterative algorithms for matrix square roots and polar decomposition–two primitive matrix functions that arise in neural network optimizers–are closely related to sign computation, and analogous results readily hold for those algorithms (see Section 5). We will discuss how Part II of PRISM accelerates classical Newton-Schulz iterations. Although the derivations and illustrations presented in this section are specific to matrix sign, analogous results, such as why classical Newton-Schulz is slow and why PRISM accelerates them, hold more generally for other Newton-Schulz-like algorithms, including all those present in Table 1.

For a square matrix $\boldsymbol{A} \in \mathbb{R}^{n \times n}$, the matrix sign function is defined as $\text{sign}(\boldsymbol{A}) = \boldsymbol{A}(\boldsymbol{A}^2)^{-1/2}$. For our analysis we require that $\boldsymbol{A}^2$ is symmetric, which practically covers all relevant use cases and hence we will assume it true through-

out. To simplify notation, we also assume that $\|\boldsymbol{A}\|_2 \leq 1$, with the understanding that such condition is easily satisfied by normalization $\boldsymbol{A} \mapsto \boldsymbol{A}/\|\boldsymbol{A}\|_F$. As derived in Section 3, the Newton-Schulz iteration for matrix sign is

$$\boldsymbol{X}_0 = \boldsymbol{A}, \ \boldsymbol{R}_k = \boldsymbol{I} - \boldsymbol{X}_k^2, \ \boldsymbol{X}_{k+1} = \boldsymbol{X}_k f_d(\boldsymbol{R}_k), \quad (1)$$

and $f_d(\xi)$ is the $d$-th order Taylor approximation of $f(\xi) = (1-\xi)^{-1/2}$ around $\xi = 0$. While $f_d(\xi)$ provides an accurate estimate of $f(\xi)$ near $\xi = 0$, the error increases rapidly as $\xi$ approaches 1. In the early phase of Newton-Schulz, if $\boldsymbol{X}_k$ has eigenvalues close to 0, then the matrix polynomial $f_d(\boldsymbol{I} - \boldsymbol{X}_k^2)$ is not a good approximant of $f(\boldsymbol{I} - \boldsymbol{X}_k^2)$, and thus the convergence of (1) can be slow. To see this more clearly, consider $d = 1$ and the scalar sequence

$$x_{k+1} = x_k f_1(1 - x_k^2) = x_k(1 + \tfrac{1}{2}(1 - x_k^2)).$$

It is easy to verify that, for $x_k$ close to 0,

$$1 - x_{k+1}^2 = \tfrac{3}{4}(1 - x_k^2)^2 + \tfrac{1}{4}(1 - x_k^2)^3 \approx 1 - \tfrac{9}{4}x_k^2,$$

where we used Taylor approximation around $x_k = 0$. This means that even though the sequence is still quadratically convergent since $|1 - x_{k+1}^2| \leq |1 - x_k^2|^2$, the initial convergence rate behaves much like a linear one with a relatively small constant around 9/4. An illustration is provided in Figure 2. The same observation generalizes to high-order Taylor series for $d \geq 1$ and matrix iterations.

### 4.1. Fitting Polynomials to Matrix Spectrum

In order to accelerate the convergence of (1), we replace the Taylor polynomial $f_d$ with a different polynomial $g_d$ by iteratively fitting it to the target function $f(\xi) = (1-\xi)^{-1/2}$ over the spectrum of the current iterate. This is Step 4 of the PRIME meta-algorithm. Since the error $|f(\xi) - f_d(\xi)|$ is proportional to $\xi^{d+1}$, instead of fitting an entirely new polynomial, which can be difficult and costly, we consider the class of degree-$d$ polynomials of the form

$$g_d(\xi; \alpha) = f_{d-1}(\xi) + \alpha \xi^d.$$

That is, we keep all but the coefficient of $\xi^d$ the same and change $\alpha$ so that $g_d(\xi; \alpha)$ is a better fit to the data $\{\lambda_i, f(\lambda_i)\}_{i=1}^n$ in the least-squares sense, where $\lambda_i$ denotes the $i$-th eigenvalue of the residual matrix $\boldsymbol{R} = \boldsymbol{I} - \boldsymbol{X}^2$. We will discuss in more detail shortly, but let us start with a simple example on how this can accelerate convergence. Consider again the case with $d = 1$ and the scalar sequence $x_{k+1} = x_k f_1(1 - \xi_k^2)$. We have seen that if $x_k$ is near 0 then $1 - x_{k+1}^2 \approx 1 - 2.25x_k^2$. If we replace $f_1(\xi) = 1 + \tfrac{1}{2}\xi$ with $g_1(\xi; 1) = 1 + \xi$, then for $x_k$ close to 0,

$$1 - x_{k+1}^2 = (1 - x_k^2)^2 + (1 - x_k^2)^3 - (1 - x_k^2) \approx 1 - 4x_k^2.$$

This shows that with $\alpha_k = 1$, for $x_k$ close to 0, we still maintain quadratic convergence $|1 - x_{k+1}^2| \leq |1 - x_k^2|^2$,

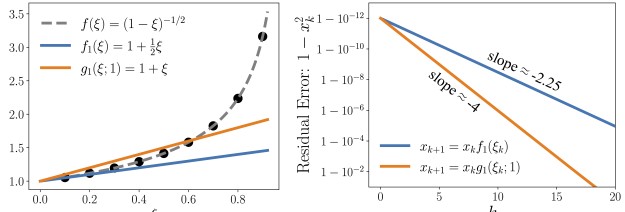

*Figure 2.* Better polynomial approximation leads to faster convergence. Left: Approximating $f(\xi)$ using its Taylor approximation $f_1(\xi)$ around $\xi = 0$ versus the alternative $g_1(\xi; 1)$. Right: The initial convergence behavior in residual error $\xi_k = 1 - x_k^2$ using the standard and "accelerated" Newton-Schulz, respectively, for $x_0 = 10^{-6}$. Using $g_1(\xi; 1)$ leads to an exponential speedup.

and although the local convergence behavior is still much like a linear one, the error $1 - x_{k+1}^2$ diminishes at a rate that is nearly *twice* as rapid. An illustration is provided in Figure 2, where we see that a better fit of $f(\xi)$ for $\xi \gg 0$ leads to a much faster convergence of the resulting sequence.

Step 4 of PRISM changes (1) into

$$\boldsymbol{X}_0 = \boldsymbol{A}, \ \boldsymbol{R}_k = \boldsymbol{I} - \boldsymbol{X}_k^2, \ \boldsymbol{X}_{k+1} = \boldsymbol{X}_k g_d(\boldsymbol{R}_k; \alpha_k^*), \quad (2)$$

$$\alpha_k^* = \arg\min_{\alpha \in [\ell, u]} \|\boldsymbol{I} - \boldsymbol{X}_k^2 g_d(\boldsymbol{R}_k; \alpha)^2\|_F^2$$

$$= \arg\min_{\alpha \in [\ell, u]} \sum_{i=1}^n (1 - (1 - \lambda_{k,i}) g_d(\lambda_{k,i}; \alpha)^2)^2, \quad (3)$$

where $\lambda_{k,1}, \lambda_{k,2}, \ldots, \lambda_{k,n}$ are the eigenvalues of $\boldsymbol{R}_k$. The last equality follows because $\boldsymbol{A}^2$ is symmetric and $\boldsymbol{X}_0 = \boldsymbol{A}$, so $\boldsymbol{R}_k$ is symmetric for all $k$. Therefore, (3) fits the polynomial $g_d(x; \alpha)$ to the set of points $\{(\lambda_{k,i}, f(\lambda_{k,i})\}_{i=1}^n$, and recall that $f(\xi) = (1-\xi)^{-1/2}$. In (3), we impose an interval constraint $\alpha \in [\ell, u]$ to regularize the problem so that $\alpha_k^*$ will not be negatively affected by potential outliers. Without such a constraint, the resulting residual matrix still has a strictly smaller Frobenius norm, and hence the sequence generated by (2) still converges to $\text{sign}(\boldsymbol{A})$. However, it is important to ensure that our choice of $\alpha_k$ indeed accelerates the overall rate of convergence rather than making it slower. A natural condition we would like to guarantee is the sequence of residual matrices $\boldsymbol{R}_k$ should have a strictly decreasing spectral norm. Since the Taylor polynomial $f_d(\xi)$ approximates $f(\xi)$ from below, i.e. $f(\xi) - f_d(\xi) > 0$ for all $\xi \in (0, 1)$, if unconstrained, $\alpha^*$ can be unnecessarily large, causing an oscillating behavior in the spectral norm of the residual matrix, e.g. having $\|\boldsymbol{R}_{k+1}\|_2 > \|\boldsymbol{R}_k\|_2$, which can hurt the overall convergence rate. The interval $[\ell, u]$ we add to (3) should ensure that the resulting iteration in (2) converges at least as fast as the original one in (1). It turns out that there is a principled way to choose $[\ell, u]$ based on the resulting polynomial properties. For example, one may set $[\ell, u] = [1/2, 1]$ for $d = 1$ and $[\ell, u] = [3/8, 29/20]$ for $d = 2$, which cover the 3rd- and 5th-order Newton-Schulz

*Table 1.* PRISM-accelerated algorithms for computing a few primitive matrix functions that arise in neural network optimizers

| Method | Target | Initialization | Iteration[*] | Residual |
|---|---|---|---|---|
| Newton-Schulz (3rd-order) | $\boldsymbol{A}^{1/2}$ $\boldsymbol{A}^{-1/2}$ | $\boldsymbol{X}_0 = \boldsymbol{A}$ $\boldsymbol{Y}_0 = \boldsymbol{I}$ | $\boldsymbol{X}_{k+1} = \boldsymbol{X}_k(\boldsymbol{I} + \alpha_k \boldsymbol{R}_k)$ $\boldsymbol{Y}_{k+1} = (\boldsymbol{I} + \alpha_k \boldsymbol{R}_k)\boldsymbol{Y}_k$ | $\boldsymbol{R}_k = \boldsymbol{I} - \boldsymbol{X}_k \boldsymbol{Y}_k$ |
| Newton-Schulz (5th-order) | $\boldsymbol{A}^{1/2}$ $\boldsymbol{A}^{-1/2}$ | $\boldsymbol{X}_0 = \boldsymbol{A}$ $\boldsymbol{Y}_0 = \boldsymbol{I}$ | $\boldsymbol{X}_{k+1} = \boldsymbol{X}_k(\boldsymbol{I} + \frac{1}{2}\boldsymbol{R}_k + \alpha_k \boldsymbol{R}_k^2)$ $\boldsymbol{Y}_{k+1} = (\boldsymbol{I} + \frac{1}{2}\boldsymbol{R}_k + \alpha_k \boldsymbol{R}_k^2)\boldsymbol{Y}_k$ | $\boldsymbol{R}_k = \boldsymbol{I} - \boldsymbol{X}_k \boldsymbol{Y}_k$ |
| Newton-Schulz (3rd-order) | $\boldsymbol{U}\boldsymbol{V}^T$ | $\boldsymbol{X}_0 = \boldsymbol{A}$ | $\boldsymbol{X}_{k+1} = \boldsymbol{X}_k(\boldsymbol{I} + \alpha_k \boldsymbol{R}_k)$ | $\boldsymbol{R}_k = \boldsymbol{I} - \boldsymbol{X}_k^T \boldsymbol{X}_k$ |
| Newton-Schulz (5th-ordNeer) | $\boldsymbol{U}\boldsymbol{V}^T$ | $\boldsymbol{X}_0 = \boldsymbol{A}$ | $\boldsymbol{X}_{k+1} = \boldsymbol{X}_k(\boldsymbol{I} + \frac{1}{2}\boldsymbol{R}_k + \alpha_k \boldsymbol{R}_k^2)$ | $\boldsymbol{R}_k = \boldsymbol{I} - \boldsymbol{X}_k^T \boldsymbol{X}_k$ |
| Coupled Inverse Newton | $\boldsymbol{A}^{-1/p}$ $(p \geq 1)$ | $\boldsymbol{X}_0 = \boldsymbol{I}$ $\boldsymbol{M}_0 = \boldsymbol{A}$ | $\boldsymbol{X}_{k+1} = \boldsymbol{X}_k(\boldsymbol{I} + \alpha_k \boldsymbol{R}_k)$ $\boldsymbol{M}_{k+1} = (\boldsymbol{I} + \alpha_k \boldsymbol{R}_k)^p \boldsymbol{M}_k$ | $\boldsymbol{R}_k = \boldsymbol{I} - \boldsymbol{X}_k^p \boldsymbol{A}$ |
| DB Newton | $\boldsymbol{A}^{1/2}$ $\boldsymbol{A}^{-1/2}$ | $\boldsymbol{X}_0 = \boldsymbol{A}$ $\boldsymbol{Y}_0 = \boldsymbol{I}$ | $\boldsymbol{X}_{k+1} = (1 - \alpha_k)\boldsymbol{X}_k + \alpha_k \boldsymbol{Y}_k^{-1}$ $\boldsymbol{Y}_{k+1} = (1 - \alpha_k)\boldsymbol{Y}_k + \alpha_k \boldsymbol{X}_k^{-1}$ | $\boldsymbol{R}_k = \boldsymbol{I} - \boldsymbol{X}_k \boldsymbol{Y}_k$ |
| Chebyshev | $\boldsymbol{A}^{-1}$ | $\boldsymbol{X}_0 = \boldsymbol{A}^T$ | $\boldsymbol{X}_{k+1} = \boldsymbol{X}_k(\boldsymbol{I} + \boldsymbol{R}_k + \alpha_k \boldsymbol{R}_k^2)$ | $\boldsymbol{R}_k = \boldsymbol{I} - \boldsymbol{A}\boldsymbol{X}_k$ |

[*]The value of $\alpha_k$ depends on both the input data and the underlying algorithm, see PRISM meta-algorithm for how it is defined/computed.

variants, respectively. We formally state a convergence result in Theorem 4.1 for the case $d = 1$, and leave an analogous result to Appendix D where we discuss how $[\ell, u]$ should be chosen in detail.

**Theorem 4.1.** *Let $\boldsymbol{A} \in \mathbb{R}^{n \times n}$ be such that $0 < \|\boldsymbol{A}\|_2 \leq 1$ and $\boldsymbol{A}^2$ is symmetric. Let $\boldsymbol{X}_0 = \boldsymbol{A}$ and consider the sequence of matrices $\boldsymbol{X}_1, \boldsymbol{X}_2, \ldots$ generated by (2) where $\alpha_k^*$ is determined by (3), with $d = 1$, $\ell = 1/2$ and $u = 1$. We have that $\boldsymbol{X}_k \to \mathrm{sign}(\boldsymbol{A})$ and $\|\boldsymbol{I} - \boldsymbol{X}_k^2\|_2 \leq \|\boldsymbol{I} - \boldsymbol{A}^2\|_2^{2^{k-2}}$.*

***Remark.*** The proof of Theorem 4.1, which we leave to Appendix C.2, is based on demonstrating that the polynomial $h(\xi, \alpha_k^*) = 1 - (1 - \xi)g_d(\xi; \alpha_k^*)^2$, which defines the recurrence relation in the residual space, maintains a good quadratic convergence behavior for all $k$ and all possible initial eigenvalues $\{\lambda_{0,i}\}_{i=1}^n \subseteq [0, 1]^n$. The assumption that $\boldsymbol{A}^2$ is symmetric covers the case $\boldsymbol{A} = \begin{bmatrix} 0 & \boldsymbol{A}' \\ \boldsymbol{I} & 0 \end{bmatrix}$ where $\boldsymbol{A}'$ is symmetric. This will be useful later to apply Theorem 4.1 and get an analogous result for computing matrix square root. Theorem 4.1 indicates that for $d = 1$ and $[\ell, u] = [1/2, 1]$ in the computation of $\alpha_k^*$, (1) converges at least as fast as the classical Newton-Schulz (Higham et al., 2005). The bounds $[\ell, u]$ depend solely on some favorable polynomial properties of $g_d(\xi; \alpha)$ and are independent of the spectrum of the input matrix. We refer the reader to Lemma C.1 for details. For the case $d \geq 2$, the same line of arguments can be applied to find reasonable choices for $[\ell, u]$. These bounds are universal in the sense that they are completely independent from the spectral properties (or any other characteristics such as the size) of the input matrix.

### 4.2. Fast Approximate Polynomial Fitting via Randomized Sketching

For the matrix sign iteration (as well as square roots and polar decomposition which we discuss later) in (2), the loss

function in (3) is a degree-4 polynomial in $\alpha_k$, i.e.,

$$m(\alpha) := \|\boldsymbol{I} - \boldsymbol{X}_k^2 g_d(\boldsymbol{I} - \boldsymbol{X}_k^2; \alpha)^2\|_F^2$$
$$= c_0 + c_1 \alpha + c_2 \alpha^2 + c_3 \alpha^3 + c_4 \alpha^4.$$

The coefficients of $m(\alpha)$ depend on $d$. For example, for $d = 1$ and $g_1(\xi; \alpha) = 1 + \alpha\xi$ we have $c_1 = \mathrm{tr}(4\boldsymbol{R}_k^3 - 4\boldsymbol{R}_k^2)$, $c_2 = \mathrm{tr}(6\boldsymbol{R}_k^4 - 10\boldsymbol{R}_k^3 + 4\boldsymbol{R}_k^2)$, $c_3 = \mathrm{tr}(4\boldsymbol{R}_k^5 - 8\boldsymbol{R}_k^4 + 4\boldsymbol{R}_k^3)$, $c_4 = \mathrm{tr}(\boldsymbol{R}_k^6 - 2\boldsymbol{R}_k^5 + \boldsymbol{R}_k^4)$. For general $d \geq 1$, computing these coefficients requires access to the diagonal entries of $\boldsymbol{R}_k^i$ for $i$ up to $4d + 2$. We will discuss how to speed up this computation using randomized sketching in the following paragraphs, but once we know $c_1, c_2, c_3, c_4$, minimizing $m(\alpha)$ can be done analytically by solving the cubic equation $m'(\alpha) = 0$. We provide more details in Appendix A.1.

In order to obtain the coefficients $c_i$'s of $m(\alpha)$, naively computing $\boldsymbol{R}_k^{4d+2}$ and then evaluating its trace requires at least $\log_2(4d + 2) \geq 2 + \log_2(d)$ matrix multiplications. This can be more expensive than executing a full iteration of classical Newton-Schulz, and thus rendering our acceleration scheme too costly. Ideally, we would like to obtain a good polynomial $g_d(\xi; \alpha_k)$ in sub-cubic time with respect to $n$. To accomplish this, we use randomized sketching and approximately minimize $m(\alpha)$ with a controlled error rate. A matrix $\boldsymbol{S} \in \mathbb{R}^{p \times n}$ is an $(p, \epsilon, \delta)$-Oblivious Subspace Embedding (OSE)[2] if for any fixed $p$-dimensional subspace $\mathbb{V} \subseteq \mathbb{R}^n$, with probability at least $1 - \delta$, for all $\boldsymbol{x} \in \mathbb{V}$ we have $(1 - \epsilon)\|\boldsymbol{x}\|_2^2 \leq \|\boldsymbol{S}\boldsymbol{x}\|_2^2 \leq (1 + \epsilon)\|\boldsymbol{x}\|_2^2$. Instead of (3), we choose $\alpha_k$ by solving the following problem:

$$\tilde{\alpha}_k = \underset{\alpha \in [\ell, u]}{\arg\min} \left\| \boldsymbol{S}_k \left( \boldsymbol{I} - \boldsymbol{X}_k^2 g_d(\boldsymbol{R}_k; \alpha)^2 \right) \right\|_F^2, \quad (4)$$

where $\boldsymbol{S}_k \in \mathbb{R}^{p \times n}$ is an OSE. The loss function in (4) is

---

[2]Subspace embeddings were first introduced by Drineas et al. (2006); they were first used in data-oblivious form by Sarlós (2006); Drineas et al. (2011); and they were popularized in RandNLA by Woodruff (2014).

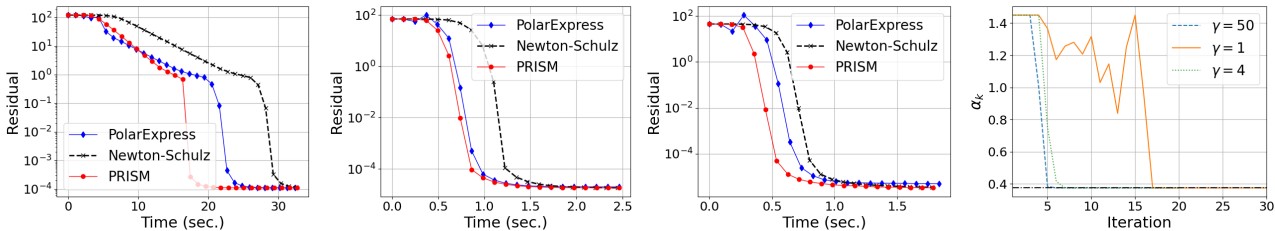

*Figure 3.* Convergence of degree-5 polynomial methods for orthogonalizing a Gaussian random matrix $\boldsymbol{A} \in \mathbb{R}^{n \times m}$ with varying aspect ratio $\gamma = n/m$. The figures from left to right show the Frobenius norm error $\|\boldsymbol{I} - \boldsymbol{X}_k^T \boldsymbol{X}_k\|_F$ for $\gamma = 1, 4, 50$, respectively. The last figure on the right shows the $\alpha_k$'s computed by (4) in PRISM for different aspect ratios at each iteration.

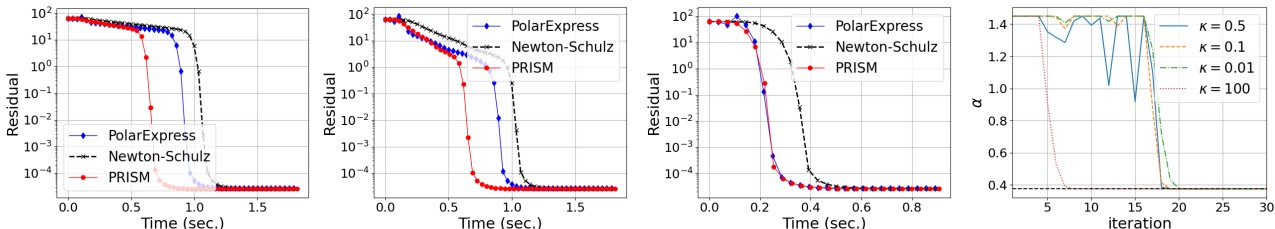

*Figure 4.* Convergence of degree-5 polynomial methods for orthogonalizing random matrices generated by HTMP (Hodgkinson et al., 2025) with different parameter $\kappa$. Smaller $\kappa$ indicates a heavier tail in the spectra. The figures from left to right show the Frobenius norm error $\|\boldsymbol{I} - \boldsymbol{X}_k^T \boldsymbol{X}_k\|_F$ for $\kappa = 0.1, 0.5, 100$, respectively. The rightmost figure shows the $\alpha_k$'s computed by (4) in PRISM.

a degree-4 polynomial in $\alpha$ whose coefficients are linear functions of $\mathrm{tr}(\boldsymbol{S}_k \boldsymbol{R}_k^i \boldsymbol{S}_k^T)$ for $1 \le i \le 4d + 2$. Therefore, computing these coefficients now requires $O(n^2 p)$ time, as opposed to $O(n^3)$ in the previous case. This means that computing $\tilde{\alpha}_k$ takes $O(n^2 p)$ time in total, which can be much less than the $O(n^3)$ complexity of one iteration of Newton-Schulz. There are many plausible choices for the sketch matrix $\boldsymbol{S}_k$, and here simple random Gaussian matrices appear to be sufficient.

One might ask whether we will lose the convergence speed when we replace the exact minimization in (3) with the approximate minimization in (4). For $d = 1$, Theorem 4.2 says that the worst-case convergence rate is essentially the same when $p = O(\log n)$. The proof uses Johnson-Lindenstrauss property of OSE and the strong convexity of $m(\alpha)$ to bound the distance between $\alpha_k^*$ from (3) and $\tilde{\alpha}_k$ from (4), and then shows that the resulting polynomial $g_d(\xi; \tilde{\alpha}_k)$ still induces a similar quadratic convergence property as $g_d(\xi; \alpha_k^*)$. We leave the proof to Appendix C.3. A similar line of arguments should generalize to the case $d \ge 2$. Empirically, for both $d = 1$ and $d = 2$, we observed that the dimension $p$ can be as small as 5 and still the sequence $\boldsymbol{X}_k$ converges as fast as if $\alpha_k^*$ were computed in (3) without sketching.

**Theorem 4.2.** *Let $\boldsymbol{A} \in \mathbb{R}^{n \times n}$ be such that $0 < \|\boldsymbol{A}\|_2 \le 1$ and $\boldsymbol{A}^2$ is symmetric. Let $\boldsymbol{S}_k \in \mathbb{R}^{p \times n}$ be random matrices consisting of i.i.d Gaussian entries $[\boldsymbol{S}_k]_{i,j} \sim \mathcal{N}(1, 1/p)$ and $p \ge 48(\log n + \log(1/\delta) + \log k + 27.6)$. Let $\boldsymbol{X}_0 = \boldsymbol{A}$ and consider the sequence of matrices $\boldsymbol{X}_1, \boldsymbol{X}_2, \dots$ generated by (2) where $\alpha_k^*$ is determined by (4), with $d = 1$, $\ell = 1/2$*

*and $u = 1$. With probability at least $1 - \delta$, we have that $\boldsymbol{X}_k \to \mathrm{sign}(\boldsymbol{A})$ and $\|\boldsymbol{I} - \boldsymbol{X}_k^2\|_2 \le \|\boldsymbol{I} - \boldsymbol{A}^2\|_2^{2^{k-3}}$.*

***Remark.*** Using a recent result on the subspace embedding property of rescaled Gaussian sketch matrices, e.g., see Proposition B.3 in Balabanov et al. (2026), the $\log n$ factor in the sketch dimension $p$ may be replaced by a constant, and consequently a constant sketch dimension $p = O(1)$ may already be sufficient to obtain the worst-cast guarantee in Theorem 4.2. This partially explains why empirically a sketch dimension $p$ as small as 5 works in our case. The overall convergence behavior in practice is surprisingly stable with respect to small sketch matrices.

## 5. PRISM-based Computation of Square Roots, Orthogonalization and Others

The following results of Higham et al. (2004) and Higham (1997) imply that what we derived for matrix sign computation readily extend to square root and orthogonalization.

**Theorem 5.1** ((Higham, 1997)). *Let $\boldsymbol{A} \in \mathbb{R}^{n \times n}$ have no eigenvalues on $\mathbb{R}_-$. Consider any iteration of the form $\boldsymbol{X}_{k+1} = \boldsymbol{X}_k h(\boldsymbol{X}_k^2)$ that converges to $\mathrm{sign}(\boldsymbol{X}_0)$ for $\boldsymbol{X}_0 = \begin{bmatrix} 0 & \boldsymbol{A} \\ \boldsymbol{I} & 0 \end{bmatrix}$ with order of convergence $q$. Then in the coupled iteration $\boldsymbol{X}_{k+1} = \boldsymbol{X}_k h(\boldsymbol{Y}_k \boldsymbol{X}_k)$, $\boldsymbol{Y}_{k+1} = h(\boldsymbol{Y}_k \boldsymbol{X}_k) \boldsymbol{Y}_k$, with $\boldsymbol{X}_0 = \boldsymbol{A}$ and $\boldsymbol{Y}_0 = \boldsymbol{I}$, we have $\boldsymbol{X}_k \to \boldsymbol{A}^{1/2}$ and $\boldsymbol{Y}_k \to \boldsymbol{A}^{-1/2}$, both with order of convergence $q$.*

**Theorem 5.2** ((Higham et al., 2004)). *Let $\boldsymbol{A} \in \mathbb{R}^{m \times n}$ with $m \ge n$ be of rank $n$ and have SVD $\boldsymbol{A} = \boldsymbol{U}\boldsymbol{\Sigma}\boldsymbol{V}^T$.*

*Consider any iteration of the form $X_{k+1} = X_k h(X_k^2)$ that converges to* $\text{sign}(X_0)$ *for* $X_0 = (A^T A)^{1/2}$ *with order of convergence q. Then* $X_{k+1} = X_k h(X_k^T X_k)$ *with* $X_0 = A$ *converges to* $UV^T$ *with order of convergence q.*

Let $A \in \mathbb{R}^{n \times n}$ be a symmetric matrix with positive real eigenvalues, and let $X_0 = \begin{bmatrix} 0 & A \\ I & 0 \end{bmatrix}$. Then $X_0^2$ is symmetric, and thus we can use PRISM to accelerate Newton-Schulz for $\text{sign}(X_0)$. Theorem 5.1 guarantees the following iteration converges to the square roots of $A$:

$$X_0 = A, \; Y_0 = I, \; R_k = I - X_k Y_k$$
$$X_{k+1} = X_k g_d(R_k; \tilde{\alpha}_k), \; Y_{k+1} = g_d(R_k; \tilde{\alpha}_k) Y_k,$$

where $g_d(\xi; \alpha) = f_{d-1}(\xi) + \alpha \xi^d$ and $f_d$ is the $d$-order Taylor polynomial of $f(\xi) = (1 - \xi)^{-1/2}$, and $\tilde{\alpha}_k$ is computed according to (4). For $d = 1$, Theorem 4.2 and Theorem 5.1 imply that it has quadratic convergence in the worst case. Similarly, one may obtain PRISM-accelerated Newton-Schulz for orthogonalization. Table 1 shows these methods for $d \in \{1, 2\}$ in explicit forms, which correspond to accelerated variants of the 3rd- and 5th-order Newton-Schulz iterations, respectively. See Appendix A.1 for details on these algorithms are derived. Pseudocode for each of these PRISM-based algorithms is provided in Appendix B.

Adaptive polynomial acceleration of other algorithms, such as those provided in Table 1, can be obtained analogously to how Part II of PRISM applies to accelerate matrix sign computation. In Appendix A, we provide detailed derivations for every algorithm from Table 1 along with explicit formulas on how to compute $\alpha_k$ in each case.

# 6. Experiments

## 6.1. Empirical Evaluation of Fast Convergence

We empirically test the accelerated convergence of PRISM-based Newton-Schulz (5th-order, cf. Table 1, and also Algorithm 2 in Appendix B) for computing the polar factor of matrix $A \in \mathbb{R}^{n \times m}$, comparing it with the classical Newton-Schulz and PolarExpress (Amsel et al., 2026). Since PRISM has an additional overhead to dynamically compute $\alpha_k$, to make the comparison fair, we measure the wall-clock time used by each algorithm. Convergence with respect to the number of iterations is shown in Appendix F. In Figure 3, we use standard Gaussian random matrices with different aspect ratios $\gamma = n/m$ and compare the convergence of the Frobenius norm error for 5th-order Newton-Schulz, Polar-Express, and PRISM. In Figure 4, we carry out the same experiment for matrices with heavy-tailed spectra. Many recent works (Martin & Mahoney, 2019; 2020; Wang et al., 2023) have observed that the weight and kernel matrices in well-trained neural networks have heavy-tailed spectra. The spectra of gradient matrices in well-trained models often inherit heavy tails. We follow Hodgkinson et al. (2025)

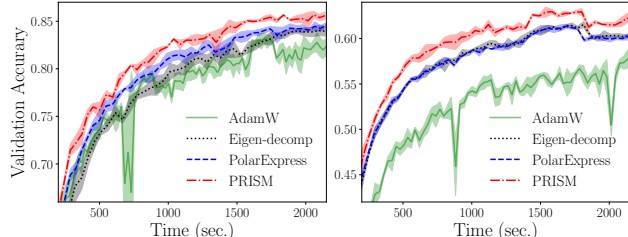

*Figure 5.* Improvement to the Shampoo optimizer in terms of training speed. We compare three methods to compute the inverse root preconditioner inside Shampoo. Left: ResNet-20 on CIFAR10. Right: ResNet-32 on CIFAR100.

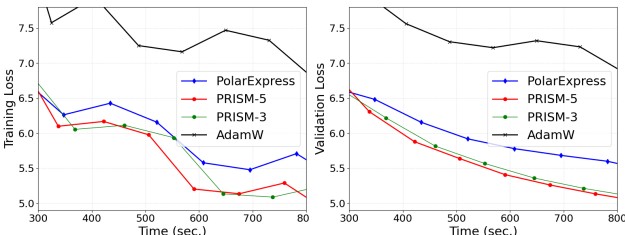

*Figure 6.* Improvement of Muon in terms of training and validation losses for the GPT-2 model. The final validation losses for PolarExpress, PRISM with degree-5 (PRISM-5) and degree-3 (PRISM-3) polynomials, and AdamW are 5.4523, 5.0251, 4.9886, and 6.8689.

to generate high-temperature Marchenko-Pastur (HTMP) random matrices to mimic the heavy-tailed gradient matrices in well-trained neural networks as the input matrix. In Figures 3 and 4 we also plot the evolutions of the coefficient $\alpha_k$ found by PRISM, which exhibit very different trends for different input matrices. Automatically adapting to the input spectra allows PRISM to converge the fastest in our experiments. For additional experiments on square roots, see Appendix F.

## 6.2. Applications to Neural Network Training

We integrate PRISM into neural network optimizers that require frequent computation of matrix functions and compare performance with existing methods. We carry out experiments using the Shampoo (Shi et al., 2023) and Muon (Jordan et al., 2024; Amsel et al., 2026) optimizers.

Shampoo is a preconditioned stochastic gradient method that generalizes AdaGrad (Duchi et al., 2011). For a given weight matrix $W_t$ and gradient $G_t$, the update step with learning rate $\eta$ is $W_{t+1} = W_t - \eta L_t^{-1/p} G_t R_t^{-1/p}$, where $L_t, R_t$ are two preconditioners maintained by Shampoo. Recent work recommended using $p = 2$ (Shi et al., 2023; Morwani et al., 2025) and this is what we use in our experiment. Previous implementations use eigen-decomposition to compute inverse roots $L_t^{-1/2}$ and $R_t^{-1/2}$. We replace this part with PRISM (accelerated 5th-order Newton-Schulz,

cf. Table 1, and also Algorithm 4 in Appendix B) and PolarExpress and compare their performance with eigen-decomposition.[3] We train slightly larger variants of ResNet-20 and ResNet-32 (He et al., 2016) for the CIFAR10 and CIFAR100 datasets. We run 5 iterations for both PolarExpress and PRISM (accelerated 5-th order Newton-Schulz). The validation accuracy throughout the first 50 epochs is shown in Figure 5. The ranking stays the same when we keep training longer.

The Muon optimizer belongs to the family of spectral descent algorithms (Carlson et al., 2015; Riabinin et al., 2025; Su, 2025; Davis & Drusvyatskiy, 2025). It gained popularity as an alternative for training large language models (Liu et al., 2025; Shah et al., 2025; Wen et al., 2025). In Figure 6, we train a GPT-2 model from random initialization with 10 layers, 16 attention heads, and an embedding dimension of 1024, using 200M tokens from the FineWeb dataset. We implement the polar decomposition of gradient matrices inside Muon with PolarExpress, PRISM-based Newton-Schulz with degree-3 and degree-5 polynomials. Additional experimental details can be found in Appendix E.

*Table 2.* NanoGPT trained by PRISM-5 with various iterations

| Iterations | 1 | 3 | 5 | 7 | 9 |
|---|---|---|---|---|---|
| Validation loss | 4.926 | 4.403 | 4.015 | 3.890 | 3.885 |
| Time (min) | 26.5 | 29.2 | 32.3 | 35.3 | 38.2 |
| Avg LMO error | 26.87 | 23.20 | 8.97 | 1.55 | 0.56 |

More iterations of PRISM at each step of training lower the residual error and the weight updates conform more strictly to exact orthogonalitzation. However, this also increases the time spent per training step. Table 2 shows results of training NanoGPT with 124M training parameters on FineWeb for 2200 steps with varying PRISM-5 iterations at each step. Here, Avg LMO error refers to mean Frobenius residual error of polar decomposition of the gradient matrices over layers at final step. Observe that a lower residual directly correlates with a lower validation loss but a strict trade-off exists (Shulgin et al., 2025): moving from 3 to 5 iterations increases total training time by 20% while yielding only marginal improvements in validation loss.

## 7. Conclusion

PRISM frames a wide range of classical iterations under a single meta-algorithmic template (Part I), then accelerates them (Part II) by dynamically fitting polynomial updates to the evolving spectrum of the current residual, without requiring a priori spectral bounds or distributional assump-

tions on singular values. Empirically, PRISM consistently delivers robust speedups across spectra that are common in ML practice, including Marchenko–Pastur-like and heavy-tailed regimes. As a result, PRISM effectively accelerates training when integrated into methods such as Shampoo and Muon. Overall, this provides a practical and general route to instance-adaptive, GPU-friendly matrix function computation, turning spectral adaptivity–previously a source of tuning burden–into a reliable algorithmic primitive for large-scale computations.

## Acknowledgements

MWM would like to acknowledge the support of the Defense Advanced Research Projects Agency (DARPA) under Contract No. HR0011-25-2-0011. SY would also like to acknowledge the support of the Natural Sciences and Engineering Research Council of Canada (NSERC). NBE would also like to acknowledge the support from the U.S. Department of Energy, Office of Advanced Scientific Computing Research, Scientific Discovery through Advanced Computing (SciDAC) program, under Contract Number DE-AC02-05CH11231 at Berkeley Lab. The views, opinions, and/or findings expressed are those of the authors and should not be interpreted as representing the official views or policies of the Department of Defense or the U.S. Government.

## Impact Statement

This paper presents work whose goal is to advance the field of Machine Learning. There are many potential societal consequences of our work, none of which we feel must be specifically highlighted here.

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

# Appendices

## Contents

# A. Deriving PRISM-based Algorithms

## A.1. Newton-Schulz iteration for matrix sign, square roots and polar decomposition

For completeness, we first apply Part I of the PRISM meta-algorithm to derive the Newton-Schulz iteration for computing the matrix sign function, and then we apply Part II to accelerate its convergence. Afterwards, we will invoke Theorem 5.1 and Theorem 5.2 to obtain PRISM-accelerated Newton-Schulz iteration for square roots and polar decomposition, respectively. For the computation of the matrix sign function, we will assume that the input matrix $\boldsymbol{A}$ is such that $\boldsymbol{A}^2$ is symmetric and $\|\boldsymbol{A}\|_2 \leq 1$.

Let $x \neq 0$ be such that $\operatorname{sign}(x) = \operatorname{sign}(a)$. Then we can write

$$\operatorname{sign}(a) = \operatorname{sign}(x) = x(x^2)^{-1/2} = x(1 - \xi)^{-1/2} = xf(\xi),$$

where $f(\xi) = (1 - \xi)^{-1/2}$, and where $\xi = 1 - x^2$ measures how close $x$ is to $\operatorname{sign}(x) = \operatorname{sign}(a)$. Therefore, the problem of approximating $\operatorname{sign}(a)$ leads to that of approximating $f(\xi)$. Using the $d$-th order Taylor polynomial $f_d(\xi)$ around $\xi = 0$, we obtain an iterative procedure

$$x_{k+1} = x_k f_d(\xi(x_k)).$$

The matrix version,

$$\boldsymbol{X}_0 = \boldsymbol{A}, \ \boldsymbol{R}_k = \boldsymbol{I} - \boldsymbol{X}_k^2, \ \boldsymbol{X}_{k+1} = \boldsymbol{X}_k f_d(\boldsymbol{R}_k),$$

is the (generalized) Newton-Schulz iteration for matrix sign (Higham et al., 2005). To accelerate the convergence, Part II of PRISM defines

$$g_d(\xi; \alpha) = f_{d-1}(\xi) + \alpha \xi^d$$

and the following iteration

$$\boldsymbol{X}_0 = \boldsymbol{A}, \ \boldsymbol{R}_k = \boldsymbol{I} - \boldsymbol{X}_k, \ \boldsymbol{X}_{k+1} = \boldsymbol{X}_k g_d(\boldsymbol{X}_k; \alpha_k),$$

$$\text{where } \alpha_k = \arg \min_{\alpha \in [\ell, u]} \left\| \boldsymbol{S}_k \left( \boldsymbol{I} - \boldsymbol{X}_k^2 g_d(\boldsymbol{R}_k; \alpha)^2 \right) \right\|_F^2,$$

and $\boldsymbol{S}_k \in \mathbb{R}^{p \times n}$ with $m \ll n$ is a sketch matrix, e.g., consisting of i.i.d Gaussian entries. For $d = 1$, Theorem 4.2 guarantees quadratic convergence if we set $[\ell, u] = [1/2, 1]$ in the computation of $\alpha_k$. For $d = 2$, empirically we find that $[\ell, u] = [3/8, 29/20]$ always leads to fast convergence.

Denote the optimization objective function in the definition of $\alpha_k$ as

$$m(\alpha) = \left\| \boldsymbol{S}_k \left( \boldsymbol{I} - \boldsymbol{X}_k^2 g_d(\boldsymbol{R}_k; \alpha)^2 \right) \right\|_F^2.$$

Then $m(\alpha)$ is a degree-4 polynomial with respect to $\alpha$, that is

$$m(\alpha) = c_0 + c_1 \alpha + c_2 \alpha^2 + c_3 \alpha^3 + c_4 \alpha^4.$$

The coefficients $c_0, c_1, c_2, c_3, c_4$ depend on $d$. For $d = 1$ we have $g_1(\xi; \alpha) = 1 + \alpha \xi$, a simple calculation yields that

$$c_1 = 4 \operatorname{tr}(\boldsymbol{S}_k \boldsymbol{R}_k^3 \boldsymbol{S}_k^T) - 4 \operatorname{tr}(\boldsymbol{S}_k \boldsymbol{R}_k^2 \boldsymbol{S}_k^T),$$
$$c_2 = 6 \operatorname{tr}(\boldsymbol{S}_k \boldsymbol{R}_k^4 \boldsymbol{S}_k^T) - 10 \operatorname{tr}(\boldsymbol{S}_k \boldsymbol{R}_k^3 \boldsymbol{S}_k^T) + 4 \operatorname{tr} \boldsymbol{S}_k \boldsymbol{R}_k^2 \boldsymbol{S}_k^T),$$
$$c_3 = 4 \operatorname{tr}(\boldsymbol{S}_k \boldsymbol{R}_k^5 \boldsymbol{S}_k^T) - 8 \operatorname{tr}(\boldsymbol{S}_k \boldsymbol{R}_k^4 \boldsymbol{S}_k^T) + 4 \operatorname{tr}(\boldsymbol{S}_k \boldsymbol{R}_k^3 \boldsymbol{S}_k^T),$$
$$c_4 = \operatorname{tr}(\boldsymbol{S}_k \boldsymbol{R}_k^6 \boldsymbol{S}_k^T) - 2 \operatorname{tr}(\boldsymbol{S}_k \boldsymbol{R}_k^5 \boldsymbol{S}_k^T) + \operatorname{tr}(\boldsymbol{S}_k \boldsymbol{R}_k^4 \boldsymbol{S}_k^T).$$

For $d = 2$ and $g_2(\xi; \alpha) = 1 + \frac{1}{2}\xi + \alpha \xi^2$, we have

$$c_1 = \frac{1}{2} \operatorname{tr}(\boldsymbol{S}_k \boldsymbol{R}_k^7 \boldsymbol{S}_k^T) + 2 \operatorname{tr}(\boldsymbol{S}_k \boldsymbol{R}_k^6 \boldsymbol{S}_k^T) + \frac{1}{2} \operatorname{tr}(\boldsymbol{S}_k \boldsymbol{R}_k^5 \boldsymbol{S}_k^T) - 3 \operatorname{tr}(\boldsymbol{S}_k \boldsymbol{R}_k^4 \boldsymbol{S}_k^T),$$
$$c_2 = \frac{3}{2} \operatorname{tr}(\boldsymbol{S}_k \boldsymbol{R}_k^8 \boldsymbol{S}_k^T) + 3 \operatorname{tr}(\boldsymbol{S}_k \boldsymbol{R}_k^7 \boldsymbol{S}_k^T) - \frac{9}{2} \operatorname{tr}(\boldsymbol{S}_k \boldsymbol{R}_k^6 \boldsymbol{S}_k^T) - 4 \operatorname{tr}(\boldsymbol{S}_k \boldsymbol{R}_k^5 \boldsymbol{S}_k^T) + 4 \operatorname{tr}(\boldsymbol{S}_k \boldsymbol{R}_k^4 \boldsymbol{S}_k^T),$$
$$c_3 = 2 \operatorname{tr}(\boldsymbol{S}_k \boldsymbol{R}_k^9 \boldsymbol{S}_k^T) - 6 \operatorname{tr}(\boldsymbol{S}_k \boldsymbol{R}_k^7 \boldsymbol{S}_k^T) + 4 \operatorname{tr}(\boldsymbol{S}_k \boldsymbol{R}_k^6 \boldsymbol{S}_k^T),$$
$$c_4 = \operatorname{tr}(\boldsymbol{S}_k \boldsymbol{R}_k^{10} \boldsymbol{S}_k^T) - 2 \operatorname{tr}(\boldsymbol{S}_k \boldsymbol{R}_k^9 \boldsymbol{S}_k^T) + \operatorname{tr}(\boldsymbol{S}_k \boldsymbol{R}_k^8 \boldsymbol{S}_k^T).$$

For general $d \geq 1$, computing these coefficients requires access to the diagonal entries of $\boldsymbol{S}_k \boldsymbol{R}_k^i \boldsymbol{S}_k^T$ for $i$ up to $4d + 2$. Computing $\boldsymbol{S}_k \boldsymbol{R}_k^i \boldsymbol{S}_k^T$ from right to left (or equivalently, from left to right) as

$$\boldsymbol{S}_k \boldsymbol{R}_k^i \boldsymbol{S}_k^T = \boldsymbol{S}_k \boldsymbol{R}_k(\cdots(\boldsymbol{R}_k(\boldsymbol{R}_k \boldsymbol{S}_k^T)))$$

takes $O(n^2 p)$ time.

Using Theorem 5.1, we get the following PRISM-accelerated Newton-Schulz iteration for computing the square root,

$$\boldsymbol{X}_0 = \boldsymbol{A}, \ \boldsymbol{Y}_0 = \boldsymbol{I}, \ \boldsymbol{R}_k = \boldsymbol{I} - \boldsymbol{X}_k \boldsymbol{Y}_k$$
$$\boldsymbol{X}_{k+1} = \boldsymbol{X}_k g_d(\boldsymbol{R}_k; \alpha_k), \ \boldsymbol{Y}_{k+1} = g_d(\boldsymbol{R}_k; \alpha_k)\boldsymbol{Y}_k,$$
$$\text{where } \alpha_k = \arg\min_{\alpha \in [\ell, u]} \left\| \boldsymbol{S}_k \left( \boldsymbol{I} - \boldsymbol{X}_k^2 g_d(\boldsymbol{R}_k; \alpha)^2 \right) \right\|_F^2.$$

It is straightforward to see that $\alpha_k$ can be computed in the same way as in the sign computation, i.e., by solving the cubic equation $m'(\alpha) = 0$. The coefficients $c_0, c_1, c_2, c_3, c_4, c_5$ of the function $m(\alpha)$ have identical formulas; the only difference is that $\boldsymbol{R}_k = \boldsymbol{I} - \boldsymbol{X}_k \boldsymbol{Y}_k$ rather than the previous $\boldsymbol{R}_k = \boldsymbol{I} - \boldsymbol{X}_k^2$. Again, $[\ell, u] = [1, 1/2]$ is recommended for $d = 1$, which corresponds to the 3rd-order Newton-Schulz iteration in Table 1; and $[\ell, u] = [3/8, 29/20]$ is recommended for $d = 2$, which corresponds to the 5th-order Newton-Schulz iteration in Table 1.

Similarly, using Theorem 5.2, we get the following PRISM-accelerated Newton-Schulz iteration for computing the polar factor $\boldsymbol{U}\boldsymbol{V}^T$, where $\boldsymbol{A} = \boldsymbol{U}\boldsymbol{\Sigma}\boldsymbol{V}^T$ is an SVD. Assume $\boldsymbol{A} \in \mathbb{R}^{m \times n}$ with $m \geq n$,

$$\boldsymbol{X}_0 = \boldsymbol{A}, \ \boldsymbol{R}_k = \boldsymbol{I} - \boldsymbol{X}_k^T \boldsymbol{X}_k, \ \boldsymbol{X}_{k+1} = \boldsymbol{X}_k g_d(\boldsymbol{R}_k; \alpha_k),$$
$$\text{where } \alpha_k = \arg\min_{\alpha \in [\ell, u]} \left\| \boldsymbol{S}_k \left( \boldsymbol{I} - \boldsymbol{X}_k^2 g_d(\boldsymbol{R}_k; \alpha)^2 \right) \right\|_F^2.$$

Again, $\alpha_k$ is computed in the same way, that is, by solving the cubic equation $m'(\alpha) = 0$, where the coefficients of $m(\alpha)$ have the same formulas as in the sign computation; the only difference is that now we have $\boldsymbol{R}_k = \boldsymbol{I} - \boldsymbol{X}_k^T \boldsymbol{X}_k$ instead of $\boldsymbol{R}_k = \boldsymbol{I} - \boldsymbol{X}_k^2$. Since Newton-Schulz for polar decomposition shares identical convergence behavior as Newton-Schulz for sign computation, the same constraint on $\alpha$ is recommended. That is, $[\ell, u] = [1, 1/2]$ for $d = 1$, which corresponds to the 3rd-order Newton-Schulz iteration in Table 1; and $[\ell, u] = [3/8, 29/20]$ for $d = 2$, which corresponds to the 5th-order Newton-Schulz iteration in Table 1.

### A.2. DB Newton iteration for matrix square roots

We first apply Part I of the PRISM meta-algorithm to derive the Newton iteration for computing the $p$-th root of a matrix, and consequently we obtain the Newton iteration for matrix square root as a special case for $p = 2$. Then we apply Part II to accelerate its computation. We will assume that the input matrix $\boldsymbol{A}$ is symmetric since this is the most relevant case, for example, arising as the Hessian or the covariance matrix.

Let $a, x > 0$, and write

$$a^{1/p} = x(x^{-p}a)^{1/p} = xf(\xi),$$

where $f(\xi) = (1 - \xi)^{1/p}$ and $\xi = 1 - x^{-p}a$. This reduces the problem of approximating $a^{1/p}$ to that of approximating $f(\xi)$. Using the first-order Taylor approximation $f_1(\xi)$ around $\xi = 0$, i.e., $f_1(\xi) = 1 - \frac{1}{p}\xi$, we get an iterative procedure

$$x_{k+1} = x_k f_1(1 - x^{-p}a) = \frac{1}{p}((p-1)x_k + x_k^{1-p}a).$$

The matrix version,

$$\boldsymbol{X}_{k+1} = \frac{1}{p}((p-1)\boldsymbol{X}_k + \boldsymbol{X}_k^{1-p}\boldsymbol{A}),$$

is the Newton iteration for the $p$-th root (Higham et al., 2005). Part II of PRISM can be applied to accelerate Newton iteration for matrix $p$-th root for any $p \geq 2$, but here we focus on the special case $p = 2$, with the understanding that analogous results hold more generally for $p \geq 2$. The Newton iteration for matrix square root is thus

$$\boldsymbol{X}_0 = \boldsymbol{A}, \ \boldsymbol{X}_{k+1} = \frac{1}{2}\boldsymbol{X}_k + \frac{1}{2}\boldsymbol{X}_k^{-1}\boldsymbol{A}.$$

By applying PRISM, we can accelerate its convergence by executing the following iteration

$$\boldsymbol{X}_0 = \boldsymbol{A},\ \boldsymbol{X}_{k+1} = (1 - \alpha_k)\boldsymbol{X}_k + \alpha_k \boldsymbol{X}_k^{-1}\boldsymbol{A},\ \text{where } \alpha_k = \arg\min \|\xi(\boldsymbol{X}_{k+1}, \boldsymbol{A})\|_F^2$$

and $\xi(\boldsymbol{X}_k, \boldsymbol{A}) = \boldsymbol{I} - \boldsymbol{X}_{k+1}^{-1}\boldsymbol{A}$ is the residual matrix at iteration $k + 1$. In practice, coupled versions of Newton iteration are often preferred to improve numerical stability. By introducing $\boldsymbol{Y}_k = \boldsymbol{A}^{-1}\boldsymbol{X}_k$ to the standard Newton iteration, we obtain the DB Newton iteration due to Denman & Beavers (1976),

$$\boldsymbol{X}_{k+1} = \frac{1}{2}\boldsymbol{X}_k + \frac{1}{2}\boldsymbol{Y}_k^{-1},\ \boldsymbol{X}_0 = \boldsymbol{A},$$

$$\boldsymbol{Y}_{k+1} = \frac{1}{2}\boldsymbol{Y}_k + \frac{1}{2}\boldsymbol{X}_k^{-1},\ \boldsymbol{Y}_0 = \boldsymbol{I}.$$

DB Newton requires performing two matrix inversions at each iteration. To reduce the number of matrix inversions at each iteration from two to one, we may introduce $\boldsymbol{M}_k = \boldsymbol{X}_k\boldsymbol{Y}_k$ and obtain the following product form of DB Newton (Cheng et al., 2001),

$$\boldsymbol{M}_{k+1} = \frac{1}{2}\boldsymbol{I} + \frac{1}{4}\boldsymbol{M}_k + \frac{1}{4}\boldsymbol{M}_k^{-1},\ \boldsymbol{M}_0 = \boldsymbol{A},$$

$$\boldsymbol{X}_{k+1} = \frac{1}{2}\boldsymbol{X}_k + \frac{1}{2}\boldsymbol{X}_k\boldsymbol{M}_k^{-1},\ \boldsymbol{X}_0 = \boldsymbol{A},$$

$$\boldsymbol{Y}_{k+1} = \frac{1}{2}\boldsymbol{Y}_k + \frac{1}{2}\boldsymbol{Y}_k\boldsymbol{M}_k^{-1},\ \boldsymbol{Y}_0 = \boldsymbol{I}.$$

Introducing $\boldsymbol{M}_k = \boldsymbol{X}_k\boldsymbol{Y}_k$ to the PRISM-accelerated version of Newton iteration, we get

$$\boldsymbol{M}_{k+1} = 2\alpha_k(1 - \alpha_k)\boldsymbol{I} + (1 - \alpha_k)^2\boldsymbol{M}_k + \alpha_k^2\boldsymbol{M}_k^{-1},\ \boldsymbol{M}_0 = \boldsymbol{A},$$

$$\boldsymbol{X}_{k+1} = (1 - \alpha_k)\boldsymbol{X}_k + \alpha_k\boldsymbol{X}_k\boldsymbol{M}_k^{-1},\ \boldsymbol{X}_0 = \boldsymbol{A},$$

$$\boldsymbol{Y}_{k+1} = (1 - \alpha_k)\boldsymbol{Y}_k + \alpha_k\boldsymbol{Y}_k\boldsymbol{M}_k^{-1},\ \boldsymbol{Y}_0 = \boldsymbol{I},$$

$$\text{where } \alpha_k = \arg\min \|\boldsymbol{I} - \boldsymbol{M}_{k+1}\|_F^2.$$

Given $\boldsymbol{M}_k$, the function $\|\boldsymbol{I} - \boldsymbol{M}_{k+1}\|_F^2$ is a degree-4 polynomial with respect to $\alpha_k$, i.e.

$$\|\boldsymbol{I} - \boldsymbol{M}_{k+1}\|_F^2 = m(\alpha_k) = c_0 + c_1\alpha_k + c_2\alpha_k^2 + c_3\alpha_k^3 + c_4\alpha_k^4,$$

where

$$c_1 = \text{tr}(-4\boldsymbol{I} + 8\boldsymbol{M}_k - 4\boldsymbol{M}_k^2),$$

$$c_2 = \text{tr}(10\boldsymbol{I} - 14\boldsymbol{M}_k + 6\boldsymbol{M}_k^2 - 2\boldsymbol{M}_k^{-1}),$$

$$c_3 = \text{tr}(-12\boldsymbol{I} + 12\boldsymbol{M}_k - 4\boldsymbol{M}_k^2 + 4\boldsymbol{M}_k^{-1}),$$

$$c_4 = \text{tr}(6\boldsymbol{I} - 4\boldsymbol{M}_k + \boldsymbol{M}_k^2 - 4\boldsymbol{M}_k^{-1} + \boldsymbol{M}_k^{-2}).$$

Using the linearity of matrix trace and the fact that for symmetric matrix $\boldsymbol{A}$,

$$\text{tr}(\boldsymbol{A}^2) = \sum_{i,j} \boldsymbol{A}_{i,j}^2,$$

all of these coefficients can be efficiently computed in $O(n^2)$ time, without having to perform matrix multiplications. Therefore, the optimal $\alpha_k$ that minimizes the Frobenius norm of the residual matrix for the next iterate can be efficiently computed without sketching. We note that this is a distinct difference compared with Newton-Schulz-like algorithms for computing square roots. In addition, unlike Newton-Schulz iteration for square root which has a local convergence region, since Newton iteration for matrix square root is globally convergent, we do not need to impose any interval constraint on the coefficient $\alpha_k$ when we solve the optimization problem.

**Remark on computing the matrix inverse $\boldsymbol{M}_k^{-1}$.** When the input matrix $\boldsymbol{A}$ is symmetric, one can easily verify that $\boldsymbol{M}_k$ is symmetric for all $k$. Therefore, $\boldsymbol{M}_k^{-1}$ can be computed via triangular solve from the Cholesky factorization of $\boldsymbol{M}_k$. This can greatly improve the practical runtime of the Newton iteration. In Figure 11, we show that PRISM-based Newton iteration can outperform PRISM-based Newton-Schulz by a good margin.

### A.3. Inverse Newton iteration for inverse $p$-th root

We first apply Part I of the PRISM meta-algorithm to derive inverse Newton iteration for computing the inverse $p$-th root of a matrix, and then we apply Part II to accelerate its computation. We will again assume that the input matrix $A$ is symmetric.

Let $a, x > 0$, and write

$$a^{-1/p} = x(x^p a)^{-1/p} = xf(\xi),$$

where $f(\xi) = (1 - \xi)^{-1/p}$ and $\xi = 1 - x^p a$. This reduces the problem of approximating $a^{-1/p}$ to that of approximating $f(\xi)$. Using the first-order Taylor approximation $f_1(\xi)$ around $\xi = 0$, i.e., $f_1(\xi) = 1 + \frac{1}{p}\xi$, we get an iterative procedure

$$x_{k+1} = x_k f_1(1 - x^p a) = \frac{1}{p}((p + 1)x_k - x_k^{p+1} a).$$

The matrix version,

$$X = A, \ X_{k+1} = \frac{1}{p}((p + 1)X_k - X_k^{p+1} A),$$

is the inverse Newton iteration for the inverse $p$-th root (Higham et al., 2005). When $p = 1$, this is a variant of the Newton-Schulz iteration for matrix inverse (Higham et al., 2005). In practice, by introducing $M_k = X_k^p A$, the following coupled inverse Newton iteration is preferred to improve numerical stability (Higham et al., 2005),

$$X_{k+1} = X_k \left( \frac{(p+1)I - M_k}{p} \right), \ X_0 = \frac{1}{c}I,$$

$$M_{k+1} = \left( \frac{(p+1)I - M_k}{p} \right)^p M_k, \ M_0 = \frac{1}{c^p}A,$$

where a good choice of $c$ to guarantee convergence is

$$c = \left( \frac{2\|A\|_F}{p + 1} \right)^{1/p}.$$

By noting that the residual matrix

$$R_k = \xi(X_k, A) = I - X_k^p A = I - M_k,$$

the above coupled inverse Newton iteration can be equivalently written as

$$R_k = I - M_k, \ X_{k+1} = X_k \left( I + \frac{1}{p}R_k \right), \ X_0 = \frac{1}{c}I,$$

$$M_{k+1} = \left( I + \frac{1}{p}R_k \right)^p M_k, \ M_0 = \frac{1}{c^p}A.$$

Applying PRISM to this, we get

$$R_k = I - M_k, \ X_{k+1} = X_k(I + \alpha_k R_k), \ X_0 = \frac{1}{c}I,$$

$$M_{k+1} = (I + \alpha_k R_k)^p M_k, \ M_0 = \frac{1}{c^p}A,$$

$$\text{where } \alpha_k = \underset{\alpha \in [\ell, u]}{\arg\min} \left\| S_k \left( R_k + \sum_{i=1}^p \binom{p}{i} \alpha^i (R_k^{i+1} - R_k^i) \right) \right\|_F^2,$$

and $S_k \in \mathbb{R}^{m \times n}$ where $m \ll n$ is a sketch matrix, e.g., consisting of i.i.d Gaussian entries. Denote the optimization objective function in the definition of $\alpha_k$ as

$$m(\alpha) = \left\| S_k \left( R_k + \sum_{i=1}^p \binom{p}{i} \alpha^i (R_k^{i+1} - R_k^i) \right) \right\|_F^2.$$

The function $m(\alpha)$ is a polynomial of degree $2p$ with respect to $\alpha$. For $p = 1$, we have

$$m(\alpha) = c_0 + c_1\alpha + c_2\alpha^2$$

where

$$c_1 = 2\operatorname{tr}(\boldsymbol{S}_k\boldsymbol{R}_k^3\boldsymbol{S}_k^T) - 2\operatorname{tr}(\boldsymbol{S}_k\boldsymbol{R}_k^2\boldsymbol{S}_k^T),$$
$$c_2 = \operatorname{tr}(\boldsymbol{S}_k\boldsymbol{R}_k^4\boldsymbol{S}_k^T) - 2\operatorname{tr}(\boldsymbol{S}_k\boldsymbol{R}_k^3\boldsymbol{S}_k^T) + \operatorname{tr}(\boldsymbol{S}_k\boldsymbol{R}_k^2\boldsymbol{S}_k^T).$$

For $p = 2$, we have

$$m(\alpha) = c_0 + c_1\alpha + c_2\alpha^2 + c_3\alpha^3 + c_4\alpha^4,$$

where

$$c_1 = 4\operatorname{tr}(\boldsymbol{S}_k\boldsymbol{R}_k^3\boldsymbol{S}_k^T) - 4\operatorname{tr}(\boldsymbol{S}_k\boldsymbol{R}_k^2\boldsymbol{S}_k^T),$$
$$c_2 = 6\operatorname{tr}(\boldsymbol{S}_k\boldsymbol{R}_k^4\boldsymbol{S}_k^T) - 10\operatorname{tr}(\boldsymbol{S}_k\boldsymbol{R}_k^3\boldsymbol{S}_k^T) + 4\operatorname{tr}(\boldsymbol{S}_k\boldsymbol{R}_k^2\boldsymbol{S}_k^T),$$
$$c_3 = 4\operatorname{tr}(\boldsymbol{S}_k\boldsymbol{R}_k^5\boldsymbol{S}_k^T) - 8\operatorname{tr}(\boldsymbol{S}_k\boldsymbol{R}_k^4\boldsymbol{S}_k^T) + 4\operatorname{tr}(\boldsymbol{S}_k\boldsymbol{R}_k^3\boldsymbol{S}_k^T),$$
$$c_4 = \operatorname{tr}(\boldsymbol{S}_k\boldsymbol{R}_k^6\boldsymbol{S}_k^T) - 2\operatorname{tr}(\boldsymbol{S}_k\boldsymbol{R}_k^5\boldsymbol{S}_k^T) + \operatorname{tr}(\boldsymbol{S}_k\boldsymbol{R}_k^4\boldsymbol{S}_k^T).$$

In both cases, computing the coefficients of $m(\alpha)$ takes $O(mn^2)$ time. Once the coefficients are known, $\alpha_k$ that minimizes $m(\alpha)$ can be computed analytically by solving $m'(\alpha) = 0$.

For $p \geq 3$, the coefficients of $m(\alpha) = \sum_{i=0}^{2p} c_i\alpha^i$ can be computed similarly in $O(mn^2)$ time, but minimizing $m(\alpha)$ requires a numerical optimization algorithm. Since this is a scalar polynomial function, this can be done by numerically solving $m'(\alpha) = 0$ and then evaluating $m(\alpha)$ at the roots of its derivative. Numerically computing the roots of $m'(\alpha)$ by computing the eigenvalues of the companion matrix takes $O(p^3)$ times.

## A.4. Chebyshev's iteration for inverse

We first apply Part I of the PRISM meta-algorithm to derive Chebyshev's iteration for computing matrix inverse, and then we apply Part II to accelerate its computation. Here, we do not require the input matrix $\boldsymbol{A}$ to be symmetric, but we will assume that $\|\boldsymbol{A}\|_2 \leq 1$, which is easily satisfied by normalizing $\boldsymbol{A} \mapsto \boldsymbol{A}/\|\boldsymbol{A}\|_F$ for a general full-rank square matrix $\boldsymbol{A}$.

Let $a, x \neq 0$, and write

$$a^{-1} = x(ax)^{-1} = xf(\xi),$$

where $f(\xi) = (1 - \xi)^{-1}$ and $\xi = 1 - ax$. Using the second-order Taylor approximation $f_2(\xi)$ around $\xi = 0$, i.e., $f_2(\xi) = 1 + \xi + \xi^2$, we get an iterative procedure

$$x_{k+1} = x_k f_2(1 - ax) = 3x_k - 3x_k a x_k + x_k a x_k a_k.$$

The matrix version,

$$\boldsymbol{X}_0 = \boldsymbol{A}^T, \ \ \boldsymbol{X}_{k+1} = \boldsymbol{X}_k(\boldsymbol{I} + \boldsymbol{R}_k + \boldsymbol{R}_k^2) = 3\boldsymbol{X}_k - 3\boldsymbol{X}_k\boldsymbol{A}\boldsymbol{X}_k + \boldsymbol{X}_k\boldsymbol{A}\boldsymbol{X}_k\boldsymbol{A}\boldsymbol{X}_k,$$

gives Chebyshev's iteration, where $\boldsymbol{R}_k = \boldsymbol{I} - \boldsymbol{A}\boldsymbol{X}_k$.

Applying Part II of PRISM, we get the following accelerated version,

$$\boldsymbol{X}_0 = \boldsymbol{A}^T, \ \ \boldsymbol{R}_k = \boldsymbol{I} - \boldsymbol{A}\boldsymbol{X}_k, \ \ \boldsymbol{X}_{k+1} = \boldsymbol{X}_k(\boldsymbol{I} + \boldsymbol{R}_k + \alpha_k\boldsymbol{R}_k^2),$$
$$\text{where } \alpha_k = \arg\min_{\alpha \in [\ell, u]} \left\| \boldsymbol{S}_k\left(\boldsymbol{R}_k^2 - \alpha(\boldsymbol{R}_k^2 - \boldsymbol{R}_k^3)\right) \right\|_F^2.$$

Write

$$m(\alpha) = \left\| \boldsymbol{S}_k\left(\boldsymbol{R}_k^2 - \alpha(\boldsymbol{R}_k^2 - \boldsymbol{R}_k^3)\right) \right\|_F^2 = c_0 + c_1\alpha + c_2\alpha^2,$$

we have

$$c_1 = -2\operatorname{tr}(\boldsymbol{S}_k\boldsymbol{R}_k^4\boldsymbol{S}_k^T) + 2\operatorname{tr}(\boldsymbol{S}_k\boldsymbol{R}_k^5\boldsymbol{S}_k^T)$$
$$c_2 = \operatorname{tr}(\boldsymbol{S}_k\boldsymbol{R}_k^4\boldsymbol{S}_k^T) - 2\operatorname{tr}(\boldsymbol{S}_k\boldsymbol{R}_k^5\boldsymbol{S}_k^T) + \operatorname{tr}(\boldsymbol{S}_k\boldsymbol{R}_k^6\boldsymbol{S}_k^T).$$

Therefore, in the PRISM-accelerated Chebyshev's iteration, $\alpha_k$ can be computed in closed-form by solving $m'(\alpha) = 0$. Empirically, we found that enforcing $\alpha_k \in [\ell, u] = [1/2, 2]$ is sufficient to ensure fast convergence.

# B. Pseudocode of Newton-Schulz Variants for Polar Decomposition and Square Roots

We provide a pseudocode for the PRISM-based Newton-Schulz iterations (3rd- and 5th-order, respectively) for computing the polar factor and square roots, respectively. These follow closely with what we implemented in our experiments. Other PRISM-based algorithms, such as PRISM-based Inverse Newton for the inverse 4th root, and PRISM-based Newton iteration for square roots, can be implemented in a similar fashion. Note that, for each of these algorithms, we have already provided all necessary details in Appendix A. This section shows a bit more details (and in a more compact form, so it is easier to navigate) to make it easier for an actual implementation of PRISM-based Newton-Schulz iterations from Appendix A.1.

**Notation:** We use $\odot$ to denote element-wise multiplication. For a matrix $\boldsymbol{A}$, we use $\mathrm{sum}(\boldsymbol{A})$ to denote the sum of all elements of $\boldsymbol{A}$.

**Practical consideration on the sketch matrix.** Even though our theory (e.g. Theorem 4.2) requires a different sketch matrix $\boldsymbol{S}_k$ at every iteration, in practice we never encounter an issue by using a single sketch matrix $\boldsymbol{S}$ throughout all iterations. In fact, using a single, fixed sketch matrix does not seem to harm the practical performance at all. We thus conjecture that the theoretical requirement for a different sketch matrix at every iteration may just be an artifact of our proof approach, but not really necessary. Therefore, in a practical implementation, we suggest using a fixed sketch matrix to avoid having to sample a different one at every iteration. There, in the pseudocode, we sample a fixed Gaussian sketch matrix only once at the beginning. Additionally, we fix the sketch dimension to be constant at 16. This appears to be more than enough for all practical purposes, including input matrices of dimension up to 10,000 x 10,000.

**Practical consideration on computing** $\mathrm{tr}(\boldsymbol{S}\boldsymbol{A}^k\boldsymbol{S}^T)$. If $\boldsymbol{A}$ is symmetric, then for any $i, j, k \geq 1$ such that $i + j = k$, we have the identity

$$\mathrm{tr}(\boldsymbol{S}\boldsymbol{A}^k\boldsymbol{S}^T) = \mathrm{sum}((\boldsymbol{A}^i\boldsymbol{S}) \odot (\boldsymbol{A}^j\boldsymbol{S})).$$

We use this to slightly reduce the number of matrix-matrix multiplications, for example, when computing $\mathrm{tr}(\boldsymbol{S}\boldsymbol{A}^{10}\boldsymbol{S}^T)$, we just need to first compute $\boldsymbol{A}^5\boldsymbol{S} = \boldsymbol{A}(\boldsymbol{A}(\boldsymbol{A}(\boldsymbol{A}(\boldsymbol{A}\boldsymbol{S}))))$, and then use $\mathrm{tr}(\boldsymbol{S}\boldsymbol{A}^{10}\boldsymbol{S}^T) = \mathrm{sum}((\boldsymbol{A}^5\boldsymbol{S}) \odot (\boldsymbol{A}^5\boldsymbol{S}))$.

## B.1. Algorithm 1: PRISM Newton-Schulz (3rd order) for polar decomposition

**Input:** Matrix $\boldsymbol{A} \in \mathbb{R}^{m \times n}$, where $m \geq n$. If $m < n$, take the transpose $\boldsymbol{A}^T$. Constants $\ell = 0.5$ and $u = 1$. Number of iterations $K \geq 1$.
**Output:** The polar factor $\boldsymbol{O} = \boldsymbol{U}\boldsymbol{V}^T$ of $\boldsymbol{A}$, where $\boldsymbol{U} \in \mathbb{R}^{m \times n}$ and $\boldsymbol{V} \in \mathbb{R}^{n \times n}$ contain the left and right singular vectors of $\boldsymbol{A}$.

1. Sample a Gaussian sketch matrix $\boldsymbol{S} \in \mathbb{R}^{n \times 16}$, i.e., each entry $\boldsymbol{S}_{ij}$ is i.i.d. Gaussian with mean 0 and variance 1/16.

2. Initialize $\boldsymbol{X}_0 = \boldsymbol{A}/\|\boldsymbol{A}\|_F$.

3. For $k = 0, 1, 2, \ldots, K-1$:

    (a) Compute the residual $\boldsymbol{R}_k = \boldsymbol{I} - \boldsymbol{X}_k^T\boldsymbol{X}_k$.
    (b) Compute the matrix sketches $\tilde{\boldsymbol{R}}_k^{(1)} = \boldsymbol{R}_k\boldsymbol{S}$ and $\tilde{\boldsymbol{R}}_k^{(i+1)} = \boldsymbol{R}_k\tilde{\boldsymbol{R}}_k^{(i)}$ for $i = 1, 2, 3$.
    (c) Compute the trace estimates

    $$r_k^{(2)} = \mathrm{sum}(\tilde{\boldsymbol{R}}_k^{(1)} \odot \tilde{\boldsymbol{R}}_k^{(1)}), \quad r_k^{(3)} = \mathrm{sum}(\tilde{\boldsymbol{R}}_k^{(2)} \odot \tilde{\boldsymbol{R}}_k^{(1)}), \quad r_k^{(4)} = \mathrm{sum}(\tilde{\boldsymbol{R}}_k^{(2)} \odot \tilde{\boldsymbol{R}}_k^{(2)}),$$
    $$r_k^{(5)} = \mathrm{sum}(\tilde{\boldsymbol{R}}_k^{(2)} \odot \tilde{\boldsymbol{R}}_k^{(3)}), \quad r_k^{(6)} = \mathrm{sum}(\tilde{\boldsymbol{R}}_k^{(3)} \odot \tilde{\boldsymbol{R}}_k^{(3)}).$$

    (d) Compute $\alpha_k$ via polynomial fitting, i.e.,

    $$\alpha_k = \underset{\alpha \in [\ell, u]}{\arg\min} \; c_k^{(1)}\alpha + c_k^{(2)}\alpha^2 + c_k^{(3)}\alpha^3 + c_k^{(4)}\alpha^4,$$

    where

    $$c_k^{(1)} = 4r_k^{(3)} - 4r_k^{(2)},$$
    $$c_k^{(2)} = 6r_k^{(4)} - 10r_k^{(3)} + 4r_k^{(2)},$$
    $$c_k^{(3)} = 4r_k^{(5)} - 8r_k^{(4)} + 4r_k^{(3)},$$
    $$c_k^{(4)} = r_k^{(6)} - 2r_k^{(5)} + r_k^{(4)}.$$

(e) Update $\boldsymbol{X}_{k+1} = \boldsymbol{X}_k(\boldsymbol{I} + \alpha_k \boldsymbol{R}_k)$.

4. Return $\boldsymbol{X}_K$.

## B.2. Algorithm 2: PRISM Newton-Schulz (5th order) for polar decomposition

**Input:** Matrix $\boldsymbol{A} \in \mathbb{R}^{m \times n}$, where $m \geq n$. If $m < n$, take the transpose $\boldsymbol{A}^T$. Constants $\ell = 0.375$ and $u = 1.45$. Number of iterations $K \geq 1$.
**Output:** The polar factor $\boldsymbol{O} = \boldsymbol{U}\boldsymbol{V}^T$ of $\boldsymbol{A}$, where $\boldsymbol{U} \in \mathbb{R}^{m \times n}$ and $\boldsymbol{V} \in \mathbb{R}^{n \times n}$ contain the left and right singular vectors of $\boldsymbol{A}$.

1. Sample a Gaussian sketch matrix $\boldsymbol{S} \in \mathbb{R}^{n \times 16}$, i.e., each entry $\boldsymbol{S}_{ij}$ is i.i.d. Gaussian with mean 0 and variance 1/16.

2. Initialize $\boldsymbol{X}_0 = \boldsymbol{A}/\|\boldsymbol{A}\|_F$.

3. For $k = 0, 1, 2, \ldots, K-1$:

   (a) Compute the residual $\boldsymbol{R}_k = \boldsymbol{I} - \boldsymbol{X}_k^T \boldsymbol{X}_k$.
   (b) Compute the matrix sketches $\tilde{\boldsymbol{R}}_k^{(1)} = \boldsymbol{R}_k \boldsymbol{S}$ and $\tilde{\boldsymbol{R}}_k^{(i+1)} = \boldsymbol{R}_k \tilde{\boldsymbol{R}}_k^{(i)}$ for $i = 1, 2, 3, 4$.
   (c) Compute the trace estimates

$$
\begin{aligned}
r_k^{(2)} &= \mathrm{sum}(\tilde{\boldsymbol{R}}_k^{(1)} \odot \tilde{\boldsymbol{R}}_k^{(1)}), \quad r_k^{(3)} = \mathrm{sum}(\tilde{\boldsymbol{R}}_k^{(2)} \odot \tilde{\boldsymbol{R}}_k^{(1)}), \quad r_k^{(4)} = \mathrm{sum}(\tilde{\boldsymbol{R}}_k^{(2)} \odot \tilde{\boldsymbol{R}}_k^{(2)}), \\
r_k^{(5)} &= \mathrm{sum}(\tilde{\boldsymbol{R}}_k^{(2)} \odot \tilde{\boldsymbol{R}}_k^{(3)}), \quad r_k^{(6)} = \mathrm{sum}(\tilde{\boldsymbol{R}}_k^{(3)} \odot \tilde{\boldsymbol{R}}_k^{(3)}), \quad r_k^{(7)} = \mathrm{sum}(\tilde{\boldsymbol{R}}_k^{(3)} \odot \tilde{\boldsymbol{R}}_k^{(4)}), \\
r_k^{(8)} &= \mathrm{sum}(\tilde{\boldsymbol{R}}_k^{(4)} \odot \tilde{\boldsymbol{R}}_k^{(4)}), \quad r_k^{(9)} = \mathrm{sum}(\tilde{\boldsymbol{R}}_k^{(4)} \odot \tilde{\boldsymbol{R}}_k^{(5)}), \quad r_k^{(10)} = \mathrm{sum}(\tilde{\boldsymbol{R}}_k^{(5)} \odot \tilde{\boldsymbol{R}}_k^{(5)}).
\end{aligned}
$$

   (d) Compute $\alpha_k$ via polynomial fitting, i.e.,

$$
\alpha_k = \arg\min_{\alpha \in [\ell, u]} c_k^{(1)}\alpha + c_k^{(2)}\alpha^2 + c_k^{(3)}\alpha^3 + c_k^{(4)}\alpha^4,
$$

   where

$$
\begin{aligned}
c_k^{(1)} &= 0.5 r_k^{(7)} + 2 r_k^{(6)} + 0.5 r_k^{(5)} - 3 * r_k^{(4)}, \\
c_k^{(2)} &= 1.5 r_k^{(8)} + 3 r_k^{(7)} - 4.5 r_k^{(6)} - 4 r_k^{(5)} + 4 r_k^{(4)}, \\
c_k^{(3)} &= 2 r_k^{(9)} - 6 r_k^{(7)} + 4 r_k^{(6)}, \\
c_k^{(4)} &= r_k^{(10)} - 2 r_k^{(9)} + r_k^{(8)}.
\end{aligned}
$$

   (e) Update $\boldsymbol{X}_{k+1} = \boldsymbol{X}_k(\boldsymbol{I} + 0.5\boldsymbol{R}_k + \alpha_k \boldsymbol{R}_k^2)$.

4. Return $\boldsymbol{X}_K$.

## B.3. Algorithm 3: PRISM Newton-Schulz (3rd order) for square roots

**Input:** Symmetric matrix $\boldsymbol{A} \in \mathbb{R}^{n \times n}$. Constants $\ell = 0.5$ and $u = 1$. Number of iterations $K \geq 1$.
**Output:** $\boldsymbol{A}^{1/2}$ and $\boldsymbol{A}^{-1/2}$.

1. Sample a Gaussian sketch matrix $\boldsymbol{S} \in \mathbb{R}^{n \times 16}$, i.e., each entry $\boldsymbol{S}_{ij}$ is i.i.d. Gaussian with mean 0 and variance 1/16.

2. Let $c = \|\boldsymbol{A}\|_F$. Initialize $\boldsymbol{X}_0 = \boldsymbol{A}/c, \boldsymbol{Y}_0 = \boldsymbol{I}$.

3. For $k = 0, 1, 2, \ldots, K-1$:

   (a) Compute the residual $\boldsymbol{R}_k = \boldsymbol{I} - \boldsymbol{Y}_k^T \boldsymbol{X}_k$.
   (b) Compute the matrix sketches $\tilde{\boldsymbol{R}}_k^{(1)} = \boldsymbol{R}_k \boldsymbol{S}$ and $\tilde{\boldsymbol{R}}_k^{(i+1)} = \boldsymbol{R}_k \tilde{\boldsymbol{R}}_k^{(i)}$ for $i = 1, 2, 3$.

(c) Compute the trace estimates

$$r_k^{(2)} = \text{sum}(\tilde{\boldsymbol{R}}_k^{(1)} \odot \tilde{\boldsymbol{R}}_k^{(1)}), \quad r_k^{(3)} = \text{sum}(\tilde{\boldsymbol{R}}_k^{(2)} \odot \tilde{\boldsymbol{R}}_k^{(1)}), \quad r_k^{(4)} = \text{sum}(\tilde{\boldsymbol{R}}_k^{(2)} \odot \tilde{\boldsymbol{R}}_k^{(2)}),$$
$$r_k^{(5)} = \text{sum}(\tilde{\boldsymbol{R}}_k^{(2)} \odot \tilde{\boldsymbol{R}}_k^{(3)}), \quad r_k^{(6)} = \text{sum}(\tilde{\boldsymbol{R}}_k^{(3)} \odot \tilde{\boldsymbol{R}}_k^{(3)}).$$

(d) Compute $\alpha_k$ via polynomial fitting, i.e.,

$$\alpha_k = \underset{\alpha \in [\ell, u]}{\arg\min} \, c_k^{(1)}\alpha + c_k^{(2)}\alpha^2 + c_k^{(3)}\alpha^3 + c_k^{(4)}\alpha^4,$$

where

$$c_k^{(1)} = 4r_k^{(3)} - 4r_k^{(2)},$$
$$c_k^{(2)} = 6r_k^{(4)} - 10r_k^{(3)} + 4r_k^{(2)},$$
$$c_k^{(3)} = 4r_k^{(5)} - 8r_k^{(4)} + 4r_k^{(3)},$$
$$c_k^{(4)} = r_k^{(6)} - 2r_k^{(5)} + r_k^{(4)}.$$

(e) Update

$$\boldsymbol{Z}_k = \boldsymbol{I} + \alpha_k \boldsymbol{R}_k, \quad \boldsymbol{X}_{k+1} = \boldsymbol{X}_k \boldsymbol{Z}_k, \quad \boldsymbol{Y}_{k+1} = \boldsymbol{Z}_k \boldsymbol{Y}_k.$$

.

4. Return $\sqrt{c}\boldsymbol{X}_K \approx \boldsymbol{A}^{1/2}, \sqrt{c}^{-1}\boldsymbol{Y}_K \approx \boldsymbol{A}^{-1/2}$.

## B.4. Algorithm 4: PRISM Newton-Schulz (5th order) for square roots

**Input:** Symmetric matrix $\boldsymbol{A} \in \mathbb{R}^{n \times n}$. Constants $\ell = 0.5$ and $u = 1$. Number of iterations $K \geq 1$.
**Output:** $\boldsymbol{A}^{1/2}$ and $\boldsymbol{A}^{-1/2}$.

1. Sample a Gaussian sketch matrix $\boldsymbol{S} \in \mathbb{R}^{n \times 16}$, i.e., each entry $\boldsymbol{S}_{ij}$ is i.i.d. Gaussian with mean 0 and variance 1/16.

2. Let $c = \|\boldsymbol{A}\|_F$. Initialize $\boldsymbol{X}_0 = \boldsymbol{A}/c, \boldsymbol{Y}_0 = \boldsymbol{I}$.

3. For $k = 0, 1, 2, \ldots, K - 1$:

   (a) Compute the residual $\boldsymbol{R}_k = \boldsymbol{I} - \boldsymbol{Y}_k^T \boldsymbol{X}_k$.
   (b) Compute the matrix sketches $\tilde{\boldsymbol{R}}_k^{(1)} = \boldsymbol{R}_k \boldsymbol{S}$ and $\tilde{\boldsymbol{R}}_k^{(i+1)} = \boldsymbol{R}_k \tilde{\boldsymbol{R}}_k^{(i)}$ for $i = 1, 2, 3, 4$.
   (c) Compute the trace estimates

$$r_k^{(2)} = \text{sum}(\tilde{\boldsymbol{R}}_k^{(1)} \odot \tilde{\boldsymbol{R}}_k^{(1)}), \quad r_k^{(3)} = \text{sum}(\tilde{\boldsymbol{R}}_k^{(2)} \odot \tilde{\boldsymbol{R}}_k^{(1)}), \quad r_k^{(4)} = \text{sum}(\tilde{\boldsymbol{R}}_k^{(2)} \odot \tilde{\boldsymbol{R}}_k^{(2)}),$$
$$r_k^{(5)} = \text{sum}(\tilde{\boldsymbol{R}}_k^{(2)} \odot \tilde{\boldsymbol{R}}_k^{(3)}), \quad r_k^{(6)} = \text{sum}(\tilde{\boldsymbol{R}}_k^{(3)} \odot \tilde{\boldsymbol{R}}_k^{(3)}), \quad r_k^{(7)} = \text{sum}(\tilde{\boldsymbol{R}}_k^{(3)} \odot \tilde{\boldsymbol{R}}_k^{(4)}),$$
$$r_k^{(8)} = \text{sum}(\tilde{\boldsymbol{R}}_k^{(4)} \odot \tilde{\boldsymbol{R}}_k^{(4)}), \quad r_k^{(9)} = \text{sum}(\tilde{\boldsymbol{R}}_k^{(4)} \odot \tilde{\boldsymbol{R}}_k^{(5)}), \quad r_k^{(10)} = \text{sum}(\tilde{\boldsymbol{R}}_k^{(5)} \odot \tilde{\boldsymbol{R}}_k^{(5)}).$$

   (d) Compute $\alpha_k$ via polynomial fitting, i.e.,

$$\alpha_k = \underset{\alpha \in [\ell, u]}{\arg\min} \, c_k^{(1)}\alpha + c_k^{(2)}\alpha^2 + c_k^{(3)}\alpha^3 + c_k^{(4)}\alpha^4,$$

   where

$$c_k^{(1)} = 0.5r_k^{(7)} + 2r_k^{(6)} + 0.5r_k^{(5)} - 3 * r_k^{(4)},$$
$$c_k^{(2)} = 1.5r_k^{(8)} + 3r_k^{(7)} - 4.5r_k^{(6)} - 4r_k^{(5)} + 4r_k^{(4)},$$
$$c_k^{(3)} = 2r_k^{(9)} - 6r_k^{(7)} + 4r_k^{(6)},$$
$$c_k^{(4)} = r_k^{(10)} - 2r_k^{(9)} + r_k^{(8)}.$$

   (e) Update

$$\boldsymbol{Z}_k = \boldsymbol{I} + 0.5\boldsymbol{R}_k + \alpha_k \boldsymbol{R}_k^2, \quad \boldsymbol{X}_{k+1} = \boldsymbol{X}_k \boldsymbol{Z}_k, \quad \boldsymbol{Y}_{k+1} = \boldsymbol{Z}_k \boldsymbol{Y}_k.$$

   .

4. Return $\sqrt{c}\boldsymbol{X}_K \approx \boldsymbol{A}^{1/2}, \sqrt{c}^{-1}\boldsymbol{Y}_K \approx \boldsymbol{A}^{-1/2}$.

# C. Proof of Theorems in the Main Text

## C.1. Technical lemma

The proofs rely heavily on the following lemma which summarizes some important properties of the polynomial

$$h(x, \alpha) = 1 - (1 - x)(1 + \alpha x)^2.$$

**Lemma C.1.** *The function* $h(x, \alpha) = 1 - (1 - x)(1 + \alpha x)^2$ *has the following properties:*

1. $h(x, \alpha) \in [-1/5, x^2]$ *for all* $x \in [1/2, 1]$ *and for all* $\alpha \in [1/2, 1]$;

2. $h(x, \alpha) \in [-1/5, 1/4]$ *for all* $x \in [-1/5, 1/2]$ *and for all* $\alpha \in [1/2, 1]$;

3. *Let* $n \geq 1$ *and* $x_1, x_2, \ldots, x_n \in [-1/4, 1/4]$ *and* $\alpha^* = \arg \min_{\alpha \in [1/2, 1]} \sum_{i=1}^{n} h(x_i, \alpha)^2$, *we have*

$$\max_i |h(x_i, \alpha^*)| \leq C \max_i x_i^2 \text{ for some constant } C < 1.71.$$

4. *Let* $n \geq 1$ *and* $x_1, x_2, \ldots, x_n \in [-1/4, 1/4]$, *not all 0, and* $\alpha^* = \arg \min_{\alpha \in [1/2, 1]} \sum_{i=1}^{n} h(x_i, \alpha)^2$ *and* $\tilde{\alpha} \in [1/2, 1]$ *be such that* $\sum_{i=1}^{n} h(x_i, \tilde{\alpha})^2 \leq (1 + \gamma) \sum_{i=1}^{n} h(x_i, \alpha^*)^2$ *for some* $\gamma \geq 0$. *Then*

$$|\alpha^* - \tilde{\alpha}| < \sqrt{\gamma} D \max_i |x_i| \text{ for some constant } D < 0.51.$$

5. *Let* $n \geq 1$ *and* $x_1, x_2, \ldots, x_n \in [-1/4, 1/4]$, *and let* $\tilde{\alpha} \in [1/2, 1]$ *be such that* $\sum_{i=1}^{n} h(x_i, \tilde{\alpha})^2 \leq (1 + \gamma) \sum_{i=1}^{n} h(x_i, \alpha)^2$ *for all* $\alpha \in [1/2, 1]$, *where* $\gamma < 1.38$. *Then*

$$\max_i |h(x_i, \tilde{\alpha})| \leq E \max_i x_i^2 \text{ for some constant } E < 2.95.$$

We break the proof of Lemma C.1 into separate claims and prove each claim separately.

**Claim 1.** $h(x, \alpha) \in [-1/5, x^2]$ *for all* $x \in [1/2, 1]$ *and for all* $\alpha \in [1/2, 1]$.

*Proof.* Differentiate $h(x, \alpha)$ with respect to $\alpha$, we get that for all $x \in [1/2, 1]$ and for all $\alpha \in [1/2, 1]$,

$$\frac{\partial h}{\partial \alpha}(x, \alpha) = -2x(1 - x)(1 + \alpha x) \leq 0.$$

Therefore, the function $\alpha \mapsto h(x, \alpha)$ is monotonically decreasing on the interval $\alpha \in [1/2, 1]$. In particular,

$$h(x, 1) \leq h(x, \alpha) \leq h(x, 1/2), \ \forall \alpha \in [1/2, 1], \ \forall x \in [1/2, 1].$$

Thus, it suffices to show that

$$h(x, 1/2) \leq x^2 \text{ and } h(x, 1) \geq -1/5, \ \forall x \in [1/2, 1].$$

It is easy to verify that for all $x \in [1/2, 1]$,

$$h(x, 1/2) = \frac{3}{4}x^2 + \frac{1}{4}x^3 \leq x^2.$$

To see that for all $x \in [1/2, 1]$ one also has

$$h(x, 1) = -x + x^2 + x^3 \geq -1/5,$$

we note that the derivative with respect to $x$,

$$\frac{\partial h}{\partial x}(x, 1) = -1 + 2x + 3x^2 \geq 0, \ \forall x \in [1/2, 1],$$

so $h(x, 1)$ is increasing on the interval $[1/2, 1]$, and thus for all $x \in [1/2, 1]$ one has

$$h(x, 1) \geq h(1/2, 1) = -\frac{1}{2} + \frac{1}{4} + \frac{1}{8} = -\frac{1}{8} \geq -\frac{1}{5}.$$

This completes the proof. $\square$

**Claim 2.** $h(x, \alpha) \in [-1/5, 1/4]$ *for all* $x \in [-1/5, 1/2]$ *and for all* $\alpha \in [1/2, 1]$.

*Proof.* Define
$$g(x, \alpha) = (1 - x)(1 + \alpha x)^2,$$
so that $h(x, \alpha) = 1 - g(x, \alpha)$. The required result is equivalent to
$$\frac{3}{4} \le g(x, \alpha) \le \frac{6}{5},$$
which we will show in the next. Differentiate $g(x, \alpha)$ with respect to $\alpha$,
$$\frac{\partial g}{\partial \alpha}(x, \alpha) = 2x(1 - x)(1 + \alpha x).$$

For $x \in [-1/5, 1/2]$ and $\alpha \in [1/2, 1]$ we have $1 - x \ge 1/2 > 0$ and $1 + \alpha x \ge 4/5 > 0$. Hence the sign of $\partial g/\partial \alpha$ is the sign of $x$. Consequently,

- if $x \in [0, 1/2]$, then $g(x, \alpha)$ is increasing in $\alpha$;
- if $x \in [-1/5, 0]$, then $g(x, \alpha)$ is decreasing in $\alpha$.

Therefore, for fixed $x$, the extrema of $g(x, \alpha)$ on $\alpha \in [1/2, 1]$ are attained at $\alpha = 1/2$ or $\alpha = 1$. Define
$$\phi(x) = g(x, 1/2) = 1 - \frac{3}{4}x^2 - \frac{1}{4}x^3,$$
$$\psi(x) = g(x, 1) = 1 + x - x^2 - x^3.$$

We consider two cases depending on if $x \in [0, 1/2]$ of $x \in [-1/5, 0]$. If $x \in [0, 1/2]$, then the minimum occurs at $\alpha = 1/2$ and the maximum occurs at $\alpha = 1$. That is,
$$\phi(x) \le g(x, \alpha) \le \psi(x).$$

We have
$$\phi'(x) = -3x(2 + x)/4 \le 0, \ \forall x \in [0, 1/2],$$
so $\phi(x)$ is decreasing on $[0, 1/2]$. Hence
$$g(x, \alpha) \ge \phi(x) \ge \phi(1/2) = \frac{25}{32} > \frac{3}{4}, \ \forall x \in [0, 1/2], \ \forall \alpha \in [1/2, 1].$$

Furthermore,
$$\psi'(x) = 1 - 2x - 3x^2 = (1 - 3x)(1 + x)$$
has a root at $x = 1/3$ in the interval $[0, 1/2]$. This gives
$$g(x, \alpha) \le \psi(x) \le \max\{\psi(0), \psi(1/3), \psi(1/2)\} = \frac{32}{27} < \frac{6}{5}, \ \forall x \in [0, 1/2], \ \forall \alpha \in [1/2, 1].$$

On the other hand, if $x \in [-1/5, 0]$, then the minimum occurs at $\alpha = 1$ and the maximum occurs at $\alpha = 1/2$. That is,
$$\psi(x) \le g(x, \alpha) \le \phi(x).$$

Over the interval $[-1/5, 0]$, the derivative $\psi'(x) = (1 - 3x)(1 + x) \ge 0$, so $\psi(x)$ is increasing, and hence
$$g(x, \alpha) \ge \psi(x) \ge \psi(-1/5) = \frac{96}{125} > \frac{3}{4}, \ \forall x \in [-1/5, 0], \ \forall \alpha \in [1/2, 1].$$

Similarly, the derivative $\phi'(x) = -3x(2 + x)/4 \le 0$, so $\phi(x)$ is decreasing over the interval $[-1/5, 0]$, and hence
$$g(x, \alpha) \le \phi(x) \le \phi(0) = 1 < \frac{6}{5}, \ \forall x \in [-1/5, 0], \ \forall \alpha \in [1/2, 1].$$

We obtain the required result by combining both cases. $\square$

**Claim 3.** *Let $n \geq 1$ and $x_1, x_2, \ldots, x_n \in [-1/4, 1/4]$ and $\alpha^* = \arg\min_{\alpha \in [1/2,1]} \sum_{i=1}^{n} h(x_i, \alpha)^2$, we have*

$$\max_i |h(x_i, \alpha^*)| \leq C \max_i x_i^2 \text{ for some constant } C < 1.71.$$

*Proof.* We assume that $\max_i x_i^2 > 0$, as otherwise $x_i = h(x_i, \alpha) = 0$ for all $i$ and for all $\alpha$, and therefore the result holds trivially. To prove the result when not all $x_i$'s are 0, we will construct a worst-case configuration in which the ratio $\max_i |h(x_i, \alpha^*)| / \max_i x_i^2$ is maximized as much as possible, and then we will bound the maximum ratio based on the configuration. The proof uses various monotone behaviors of $h(x, \alpha)$ with respect to $x$ or $\alpha$ and reducing the analysis to characterizing the worst-case ratio around an "outlier", or "anchor point", $x_o = -1/4$.

We start by characterizing the regions on which $|h(x, \alpha)|$ is increasing or decreasing. Fix $\alpha$ in $[1/2, 1]$ and consider

$$g_\alpha(x) = h(x, \alpha).$$

Then we have that
$$g'_\alpha(x) = 1 - 2\alpha + 4\alpha x - 2\alpha^2 x + 3\alpha^2 x^2 = (1 + \alpha x)(1 - 2\alpha + 3\alpha x),$$

and
$$\frac{d}{dx}|h(x, \alpha)| = \text{sign}(g_\alpha(x)) \cdot g'_\alpha(x).$$

By factoring
$$g_\alpha(x) = x\left(\alpha^2 x^2 + (2\alpha - \alpha^2)x + (1 - 2\alpha)\right)$$

we see that the three roots of $g_\alpha(x)$ are

$$r_-(\alpha) = \frac{\alpha - 2 - \sqrt{\alpha(\alpha + 4)}}{2\alpha}, \quad r_0(\alpha) = 0, \quad r_+(\alpha) = \frac{\alpha - 2 + \sqrt{\alpha(\alpha + 4)}}{2\alpha}.$$

Because $\alpha \in [1/2, 1]$ we get
$$r_-(\alpha) < 0 < r_+(\alpha).$$

Since $g_\alpha(x)$ is a cubic function with positive leading coefficient,

$$\text{sign}(g_\alpha(x)) = \begin{cases} 1, & \text{if } x \in (r_-(\alpha), 0) \cup (r_+(\alpha), +\infty), \\ -1, & \text{if } x \in (-\infty, r_-(\alpha)) \cup (0, r_+(\alpha)). \end{cases}$$

On the other hand, the two roots of $g'_\alpha(x)$ are

$$x_-(\alpha) = -\frac{1}{\alpha}, \quad x_+(\alpha) = \frac{2\alpha - 1}{3\alpha},$$

and since $\alpha \in [1/2, 1]$ we get that

$$g'_\alpha(x) > 0 \text{ for } x \in (-\infty, x_-(\alpha)) \cup (x_+(\alpha), +\infty),$$
$$g'_\alpha(x) < 0 \text{ for } x \in (x_-(\alpha), x_+(\alpha)).$$

Combining these, we get for a fixed $\alpha \in [1/2, 1]$,

- $|h(x, \alpha)|$ is decreasing with respect to $x$ on $(-\infty, r_-(\alpha)) \cup (x_-(\alpha), 0) \cup (x_+(\alpha), r_+(\alpha))$;

- $|h(x, \alpha)|$ is increasing with respect to $x$ on $(r_-(\alpha), x_-(\alpha)) \cup (0, x_+(\alpha)) \cup (r_+(\alpha), +\infty)$.

For $x_1, x_2, \ldots, x_n \in [-1/4, 1/4]$, denote
$$M = \max_i |x_i| \leq 1/4,$$

so we have
$$x_i \in [-M, M] \text{ for all } i.$$

Since $r_-(\alpha) \leq x_-(\alpha) \leq -1 < -M$ for every $\alpha \in [1/2, 1]$, the monotone behavior of $|h(x, \alpha)|$ implies that

$$\frac{\max_i |h(x_i, \alpha^*)|}{\max_i x_i^2} = \frac{\max_i |h(x_i, \alpha^*)|}{M^2}$$

$$\leq \max \left\{ \frac{|h(-M, \alpha^*)|}{M^2}, \frac{|h(\min\{x_+(\alpha^*), M\}, \alpha^*)|}{M^2}, \frac{|h(M, \alpha^*)|}{M^2} \right\}.$$

To bound the above quantity, for each choice of a potential "outlier"

$$x_o \in \{-M, \ \min\{x_+(\alpha^*), M\}, \ M\},$$

we find an upper bound on the maximum possible value of $|h(x_o, \alpha^*)|$ as a function of $M$, and then maximize the ratio $|h(x_o, \alpha^*)|/M^2$ over $M \in (0, 1/4]$. It turns out that, the case

$$x_o = -M.$$

gives rise to the worst-case ratio. We will focus on this case for the rest of the proof. The other cases all follow from a similar line of reasoning.

Since $-1/4 \leq x_o < 0$, it is straightforward to check that

$$\frac{d}{d\alpha} h(x_o, \alpha)^2 = -4x_o(1 - x_o)(1 + \alpha x_o)h(x_o, \alpha) \geq 0$$

for all $\alpha \in [1/2, 1]$. This means that the function $|h(x_o, \alpha)|$ is increasing with respect to $\alpha$ for $\alpha \in [1/2, 1]$, and hence $|h(x_o, \alpha^*)|$ is maximized if $\alpha^*$ is away from 1/2 as far as possible. Recall that

$$\alpha^* = \arg\min_{\alpha \in [1/2, 1]} \sum_{i=1}^n h(x_i, \alpha)^2.$$

Using the definition of $\alpha^*$, we now determine how large $\alpha^*$ can be, and consequently we derive an upper bound on $|h(x_o, \alpha^*)|$. Fix an arbitrary $x \in [-M, M]$ and consider

$$f_x(\alpha) = h(x, \alpha)^2.$$

We will show that $f_x(\alpha)$ is increasing on the interval $[\beta(x), 1]$ where

$$\beta(x) = \frac{1/\sqrt{1-x} - 1}{x},$$

and hence conclude that $a^* \leq \beta(M)$. Note that if $x = 0$ then $f_x(\alpha) = 0$ for all $\alpha$. So we consider two cases depending on the sign of $x$. If $x \in (0, 1/4]$, then by analyzing the sign of

$$f_x'(\alpha) = -4x(1 - x)(1 + \alpha x)h(x, \alpha)$$

we get that $\text{sign}(f_x'(\alpha)) = -\text{sign}(h(x, \alpha))$ for $\alpha \in [1/2, 1]$. It then follows from a straightforward analysis of $\text{sign}(h(x, \alpha))$ that

$$\text{sign}(f_x'(\alpha)) = \begin{cases} -1, & \text{if } \alpha \in [1/2, \beta(x)], \\ 1, & \text{if } \alpha \in [\beta(x), 1]. \end{cases}$$

Similarly, if $x \in [-1/4, 0)$, then for all $\alpha \in [1/2, 1]$ we have

$$\text{sign}(f_x'(\alpha)) = \text{sign}(h(x, \alpha)) = 1.$$

Combining both cases, we get that $f_x(\alpha)$ is monotonically increasing with respect to $\alpha$ for all $x \in [-1/4, 1/4]$ and for all $\alpha \in [\beta(x), 1]$. Because $\beta(x) \leq \beta(M)$ for all $x \in [-M, M]$, we must have

$$\alpha^* \leq \beta(M),$$

as otherwise one may take $\alpha = \beta(M) < \alpha^*$ to get $\sum_{i=1}^n h(x_i, \alpha) < \sum_{i=1}^n h(x_i, \alpha^*)$, contradicting the definition of $\alpha^*$. Therefore, by combining the monotone increasing property of $|h(x_o, \alpha)|$ over the interval $\alpha \in [1/2, 1]$, we get

$$|h(x_o, \alpha^*)| \leq |h(x_o, \beta(M))|.$$

Since $x_o = -M$, all it left is to compute

$$\max_{M \in (0, 1/4]} \frac{|h(-M, \beta(M))|}{M^2}, \text{ where } \beta(M) = \frac{1/\sqrt{1-M} - 1}{M}.$$

Because $|h(-M, \beta(M))| \geq 0$ for all $M \in (0, 1/4]$, we will equivalently consider $h(-M, \beta(M))$ in place of $|h(-M, \beta(M))|$. Define

$$R(M) = \frac{h(-M, \beta(M))}{M^2}.$$

Consider the change of variable

$$u = \frac{1}{\sqrt{1-M}}$$

which maps $M \in (0, 1/4]$ to $u \in (1, 2/\sqrt{3}]$, and

$$h(-M, \beta(M)) = 1 - (1+M)(1 - \beta(M)M) = 1 - (1+M)(2-u)^2.$$

Dividing by $M^2$ and then using $M = 1 - 1/u^2$, we get

$$R(M) = r(u) = -\frac{2u^2(u^2 - 2u - 2)}{(u+1)^2}.$$

And since

$$r'(u) = -\frac{4u(u+2)(u^2 - u - 1)}{(u+1)^3} > 0$$

for all $u \in [1, 2/\sqrt{3}]$, the function $r(u)$ is strictly increasing. Therefore, the maximum of $R(M)$ is attained at

$$M = 1 - \frac{1}{(2/\sqrt{3})^2} = \frac{1}{4}.$$

Finally, for $M = 1/4$, we have

$$\beta(M) = \beta(1/4) = \frac{1/\sqrt{1 - 1/4} - 1}{1/4} = \frac{8}{\sqrt{3}} - 4 \in [1/2, 1],$$

and

$$R(M) = R(1/4) = \frac{h(-1/4, \beta(1/4))}{1/16} = 16\left(-\frac{17}{3} + \frac{10}{\sqrt{3}}\right) < 1.71$$

This is an upper bound of $R(M)$ for all $M \in (0, 1/4]$, and hence the proof is complete. $\square$

**Claim 4.** *Let $n \geq 1$ and $x_1, x_2, \ldots, x_n \in [-1/4, 1/4]$, not all 0, and $\alpha^* = \arg\min_{\alpha \in [1/2, 1]} \sum_{i=1}^n h(x_i, \alpha)^2$ and $\tilde{\alpha} \in [1/2, 1]$ is such that $\sum_{i=1}^n h(x_i, \tilde{\alpha})^2 \leq (1+\gamma) \sum_{i=1}^n h(x_i, \alpha^*)^2$ for some $\gamma \geq 0$. Then*

$$|\alpha^* - \tilde{\alpha}| < \sqrt{\gamma} D \max_i |x_i| \text{ for some constant } D < 0.51.$$

*Proof.* Denote

$$L(\alpha) = \sum_{i=1}^n h(x_i, \alpha)^2 \quad \text{and} \quad S = \sum_{i=1}^n x_i^2.$$

Since not all $x_i$'s are 0, we know that $S > 0$. To bound the distance between $\alpha^*$ and $\tilde{\alpha}$, we will use the strong convexity of the function $L(\alpha)$. We start by computing a strong convexity parameter of $L(\alpha)$ by lower bounding its second-order derivative

$L''(\alpha)$. The computation is elementary and relies on characterizing the behaviors of a couple of related polynomials in $x$ and $\alpha$.

For fixed $x$, define
$$f_x(\alpha) = h(x, \alpha).$$

We have
$$f_x'(\alpha) = -2x(1-x)(1+\alpha x), \quad f_x''(\alpha) = -2x^2(1-x).$$

Therefore,
$$\frac{d^2}{d\alpha^2}\left(f_x(\alpha)^2\right) = 2f_x'(\alpha)^2 + 2f_x(\alpha)f_x''(\alpha) = 4x^2(1-x)\left(3(1-x)(1+ax)^2 - 1\right).$$

Let us write this as
$$\frac{d^2}{d\alpha^2}\left(f_x(\alpha)^2\right) = x^2 g(x, \alpha), \text{ where } g(x, \alpha) = 4(1-x)\left(3(1-x)(1+\alpha x)^2 - 1\right),$$

and hence
$$L''(\alpha) = \sum_{i=1}^{n} x_i^2 g(x_i, \alpha).$$

We will show that
$$g(x, \alpha) \geq g(1/4, 1/2) = \frac{1419}{256}, \quad \forall x \in [-1/4, 1/4], \ \forall \alpha \in [1/2, 1].$$

Rewrite $g(x, \alpha)$ as
$$g(x, \alpha) = 12(1-x)^2(1+\alpha x)^2 - 4(1-x).$$

Differentiate $g(x, \alpha)$ with respect to $\alpha$,
$$\frac{\partial g}{\partial \alpha}(x, \alpha) = 24(1-x)^2 \, x \, (1+\alpha x).$$

For $x \in [-1/4, 1/4]$ and $\alpha \in [1/2, 1]$ we have $1 - x \geq 3/4 > 0$ and $1 + \alpha x \geq 3/4 > 0$, so the sign of $\partial g/\partial \alpha$ is the sign of $x$:

- if $x > 0$, then $\partial g/\partial \alpha > 0$ and $g(x, \alpha)$ is increasing in $\alpha$, hence it attains minimum at $\alpha = 1/2$;

- if $x < 0$, then $\partial g/\partial \alpha < 0$ and $g(x, \alpha)$ is decreasing in $\alpha$, hence it attains minimum at $\alpha = 1$.

This means that the minimum value of $g(x, \alpha)$ for $x \in [-1/4, 1/4]$ and $\alpha \in [1/2, 1]$ is attained on one of the following two sets
$$\{(x, 1/2) : x \in [0, 1/4]\} \text{ and } \{(x, 1) : x \in [-1/4, 0]\}.$$

We examine each case separately. Consider the case $\alpha = 1/2$ and $x \in [0, 1/4]$. Define
$$g_1(x) = g(x, 1/2) = 12(1-x)^2(1+x/2)^2 - 4(1-x) = 3x^4 + 6x^3 - 9x^2 - 8x + 8.$$

We have that
$$g_1'(x) = 12x^3 + 18x^2 - 18x - 8 \text{ and } g_1''(x) = 36x^2 + 36x - 18.$$

The function $g_1''(x)$ has two roots $x_- = (-1 - \sqrt{3})/2$ and $x_+ = (-1 - \sqrt{3})/2$. Since $x_- < 0$ and $x_+ > 1/4$ and $g_1''$ is a convex quadratic function, we know that $g''(x) < 0$ for all $x \in [0, 1/4]$, which implies that $g_1'$ is strictly decreasing on $[0, 1/4]$. Therefore, the maximum of $g_1'(x)$ on $[0, 1/4]$ occurs at $x = 0$. Since
$$g_1'(0) = -8 < 0$$

we get $g_1'(x) < 0$ on $[0, 1/4]$, so $g_1$ is strictly decreasing, and hence
$$\min_{x \in [0, 1/4]} g(x, 1/2) = \min_{x \in [0, 1/4]} g_1(x) = g_1(1/4) = \frac{1419}{256}.$$

On the other hand, consider the case $\alpha = 1$ and $x \in [-1/4, 0]$. Define

$$g_2(x) = g(x, 1) = 12(1-x)^2(1+x)^2 - 4(1-x) = 12x^4 - 24x^2 + 4x + 8.$$

Then

$$g_2'(x) = 48x^3 - 48x + 4 \text{ and } g_2''(x) = 144x^2 - 48.$$

Since $g_2''(x) = 48(3x^2 - 1) < 0$ for $x \in [-1/4, 0]$, we know that $g_2'(x)$ is strictly decreasing, so

$$g_2'(x) \geq g_2'(0) = 4 > 0 \text{ for } x \in [-1/4, 0].$$

This means that $g_2(x)$ is strictly increasing on $[-1/4, 0]$, and consequently

$$\min_{x \in [-1/4, 0]} g(x, 1) = \min_{x \in [-1/4, 0]} g_2(x) = g_2(-1/4) = \frac{355}{64}.$$

Combining both cases we get that

$$g(x, \alpha) \geq \frac{1419}{256} \text{ for } x \in [-1/4, 1/4] \text{ and } \alpha \in [1/2, 1].$$

Consequently, we get

$$L''(\alpha) \geq \frac{1419}{256} \sum_{i=1}^{n} x_i^2 = \frac{1419}{256} S.$$

Hence $L$ is $\mu$-strongly convex on $[1/2, 1]$ with

$$\mu = \frac{1419}{259} S.$$

The strong convexity of $L(\alpha)$ implies that for all $\alpha \in [1/2, 1]$,

$$L(\alpha) \geq L(\alpha^*) + \frac{\mu}{2}(\alpha - \alpha^*))^2.$$

Applying this at $\alpha = \tilde{\alpha}$ gives

$$\frac{\mu}{2}(\tilde{\alpha} - \alpha^*)^2 \leq L(\tilde{\alpha}) - L(\alpha^*).$$

Since $L(\tilde{\alpha}) \leq (1 + \gamma)L(\alpha^*)$,

$$L(\tilde{\alpha}) - L(\alpha^*) \leq \gamma L(\alpha^*),$$

and therefore

$$|\tilde{\alpha} - \alpha^*| \leq \sqrt{\frac{2\gamma L(\alpha^*)}{\mu}} = \sqrt{\frac{512}{1419}} \sqrt{\frac{\gamma L(\alpha^*)}{S}}.$$

Now, since

$$h(x, 1/2) = x^2 \left( \frac{3}{4} + \frac{x}{4} \right),$$

so

$$h(x, 1/2)^2 = x^4 \left( \frac{3}{4} + \frac{x}{4} \right)^2 \leq \frac{169}{256} x^4 \text{ for } x \in [-1/4, 1/4].$$

Therefore, by invoking the definition of $\alpha^*$ and $M$ and $S$ we get

$$L(\alpha^*) \leq L(1/2) = \sum_{i=1}^{n} h(x, 1/2)^2 \leq \frac{169}{256} \sum_{i=1}^{n} x_i^4 \leq \frac{169}{256} M^2 \sum_{i=1}^{n} x_i^2 = \frac{169}{256} M^2 S.$$

It then follows that

$$|\tilde{\alpha} - \alpha^*| \leq \sqrt{\frac{512}{1419}} \sqrt{\frac{\gamma L(\alpha^*)}{S}} \leq \sqrt{\frac{338}{1419}} \sqrt{\gamma} M < 0.51 \sqrt{\gamma} M.$$

This proves the claim. $\qquad\square$

**Claim 5.** *Let $n \geq 1$ and $x_1, x_2, \ldots, x_n \in [-1/4, 1/4]$, and let $\tilde{\alpha} \in [1/2, 1]$ be such that $\sum_{i=1}^{n} h(x_i, \tilde{\alpha})^2 \leq (1 + \gamma) \sum_{i=1}^{n} h(x_i, \alpha)^2$ for all $\alpha \in [1/2, 1]$, where $\gamma < 1.38$. Then*

$$\max_i |h(x_i, \tilde{\alpha})| \leq E \max_i x_i^2 \text{ for some constant } E < 2.95.$$

*Proof.* The proof of this claim follows from the same line of arguments as used in the proof of Claim 3, and then we apply the distance bound of Claim 4 to get the final result. By carefully analyzing the monotone behaviors of $|h(x, \alpha)|$ for $x \in [-1/4, 1/4]$ and $\alpha \in [1/2, 1]$ as in the proof of Claim 3, we get that

$$\frac{\max_i |h(x_i, \tilde{\alpha})|}{\max_i x_i^2} = \max_{M \in (0, 1/4]} \frac{|h(-M, \tilde{\alpha})|}{M^2}$$

where $M = \max_i x_i^2$. Again, as in the proof of Claim 3, where we have showed that $a^* \leq \beta(M)$, because the function $|h(-M, \alpha)|$ is monotonically increasing with respect to $\alpha$ for $\alpha \in [1/2, 1]$, we need to determine how large $\tilde{\alpha}$ can be. Using the result of Claim 4 and the assumption that $\gamma < 1.38$, we get

$$\tilde{\alpha} \leq \alpha^* + 0.51\sqrt{1.38}M \leq \beta(M) + 0.6M.$$

Therefore, we maximize the worst-case ratio to get

$$\max_{M \in (0, 1/4]} \frac{|h(-M, \tilde{\alpha})|}{M^2} \leq \max_{M \in (0, 1/4]} \frac{h(-M, \beta(M) + 0.6M)}{M^2}$$
$$= \frac{h(-1/4, \beta(1/4) + 3/20)}{1/16}$$
$$= \frac{157}{\sqrt{3}} - \frac{84187}{960} < 2.95.$$

This finishes the proof. $\qquad\square$

### C.2. Proof of Theorem 4.1

Let $\boldsymbol{A} \in \mathbb{R}^{n \times n}$ be such that $\|\boldsymbol{A}\|_2 \leq 1$ and $\boldsymbol{A}^2$ is symmetric. Let $\boldsymbol{X}_0 = \boldsymbol{A}$ and let $\boldsymbol{X}_1, \boldsymbol{X}_2, \ldots$ be the sequence generated by Equation (2) with $d = 1$, where $\alpha_k$ is determined by Equation (3) with $\ell = 1/2$ and $u = 1$. Denote $\boldsymbol{R}_k = \boldsymbol{I} - \boldsymbol{X}_k^2$. Since $d = 1$, Equation (2) simplifies to

$$\boldsymbol{X}_{k+1} = \boldsymbol{X}_k(\boldsymbol{I} + \alpha_k(\boldsymbol{I} - \boldsymbol{X}_k^2)) = \boldsymbol{X}_k(\boldsymbol{I} + \alpha_k \boldsymbol{R}_k),$$

and hence

$$\boldsymbol{R}_{k+1} = \boldsymbol{I} - \boldsymbol{X}_{k+1}^2 = \boldsymbol{I} - (\boldsymbol{I} - \boldsymbol{R}_k)(\boldsymbol{I} + \alpha_k \boldsymbol{R}_k)^2.$$

Define $h(x, \alpha) = 1 - (1 - x)(1 + \alpha x)^2$ so that we can write the above recurrence relation with respect to $\boldsymbol{R}_k$ succinctly as

$$\boldsymbol{R}_{k+1} = h(\boldsymbol{R}_k, \alpha_k).$$

In order to see that

$$\|\boldsymbol{R}_k\|_2 \leq \|\boldsymbol{R}_0\|_2^{2^{k-2}},$$

where $\alpha_k$ is computed according to Equation (3), we rely on the properties of $h$ in Lemma C.1. Because $\boldsymbol{A}^2$ is symmetric, $\boldsymbol{R}_0 = \boldsymbol{I} - \boldsymbol{A}^2$ is also symmetric. Since $\boldsymbol{R}_{k+1} = h(\boldsymbol{R}_k, \alpha_k)$ and $h(x, \alpha)$ is a polynomial in $x$, it follows that $\boldsymbol{R}_k$ is symmetric for all $k$. Therefore,

$$\|\boldsymbol{R}_k\|_2 = \max_i |\lambda_{k,i}| \text{ for all } k,$$

where $\lambda_{k,i}$ denotes the $i$-th eigenvalue of $\boldsymbol{R}_k$. We will assume without loss of generality that the eigenvalues are ordered in a way such that $\lambda_{k+1,i} = h(\lambda_{k,i}, \alpha_k)$. For $k = 0$, because $\|\boldsymbol{X}_0\|_2 \leq 1$ and $\boldsymbol{X}_0^2$ is symmetric, the eigenvalues of $\boldsymbol{X}_0^2$ are all real-valued and lie in the interval $[0, 1]$, and therefore $0 \leq \lambda_{0,i} < 1$ for all $i$. Using Lemma C.1, we get that

- $\|\boldsymbol{R}_{k+1}\|_2 \leq \|\boldsymbol{R}_k\|_2^2$ if $\|\boldsymbol{R}_k\|_2 \geq 1/2$;

- $\|\boldsymbol{R}_{k+1}\|_2 \le 1/4$ if $\|\boldsymbol{R}_k\|_2 \le 1/2$;

- $\|\boldsymbol{R}_{k+1}\|_2 \le 1.71\|\boldsymbol{R}_k\|_2^2$ if $\|\boldsymbol{R}_k\|_2 \le 1/4$.

Let $k_1$ be such that $\|\boldsymbol{R}_{k_1}\|_2 \le 1/4 < \|\boldsymbol{R}_{k_1-1}\|_2$. Because $\|\boldsymbol{R}_{k_1-1}\|_2 > 1/4$, we must have

$$\|\boldsymbol{R}_{k_1-2}\|_2 \ge \sqrt{\|\boldsymbol{R}_{k_1-1}\|_2} > \sqrt{1/4} = 1/2 > 1.71\|\boldsymbol{R}_{k_1}\|_2.$$

Then by induction we have that for $k_2 \ge 0$,

$$\|\boldsymbol{R}_{k_1+k_2}\|_2 \le \left(1.71\|\boldsymbol{R}_{k_1}\|_2\right)^{2^{k_2}} \le \left(\|\boldsymbol{R}_{k_1-2}\|_2\right)^{2^{k_2}} \le \|\boldsymbol{R}_0\|_2^{2^{k_1+k_2-2}}.$$

Finally, the convergence to $\text{sign}(\boldsymbol{A})$ can be established following the same argument as in the proof of Theorem 3.1 and Theorem 5.2 of Kenney & Laub (1991). For completeness we repeat the main arguments below. Let

$$S = \{x : |1 - x^2| < 1\}, \ S_+ = \{x \in S : \text{Re}(x) > 0\}, \ S_- = \{x \in S : \text{Re}(x) < 0\}.$$

Denote

$$p_{k,d}(x) = xg_d(1 - x^2; \alpha_k).$$

Let $x_{k,i}$ denote the $i$-th eigenvalue of $\boldsymbol{X}_k$ and assume without loss of generality that the indices are ordered such that $x_{k+1,i} = p_{k,d}(x_{k,i})$. Then since $\|\boldsymbol{R}_k\|_2 < 1$ for all $k$, we get that

$$1 - x_{k,i}^2 < 1 \text{ and } 1 - p_{k,d}(x_{k,i})^2 = 1 - x_{k+1,i}^2 < 1, \text{ for all } k.$$

So $p_{k,d}$ maps $S$ into $S$ for all $k$. Since $S_+ \cap S_- = \emptyset$ and each is a connected set, $p_{k,d}(S_+)$ must lie entirely in either $S_+$ or $S_-$, because $p_{k,d}$ is a continuous mapping. But since $1 \in S_+$ and $p_{k,d}(1) = 1$, we must have $p_{k,d}(S_+) \subseteq S_+$ for all $k$. Similarly, $p_{k,d}(S_-) \subseteq S_-$ for all $k$. Thus, by induction, we have that if $x_{0,i} \in S_+$ then $x_{k,i} \in S_+$ for all $k$. Since $\lim_{k \to +\infty} \|\boldsymbol{R}_k\|_2 = 0$, which means $\lim_{k \to +\infty} 1 - x_{k,i}^2 = 0$ for all $i$, we must have

$$\lim_{k \to +\infty} x_{k,i} = \text{sign}(x_{0,i})$$

for all i. Then, using Lemma 5.1 of Kenney & Laub (1991) and the definition of matrix sign in terms of the Jordan form, we get $\boldsymbol{X}_k \to \text{sign}(\boldsymbol{X}_0)$.

### C.3. Proof of Theorem 4.2

The basic setup is the same as in the proof of Theorem 4.1. For the reader's convenience, we repeat the same setup here. Let $\boldsymbol{A} \in \mathbb{R}^{n \times n}$ be such that $\|\boldsymbol{A}\|_2 \le 1$ and $\boldsymbol{A}^2$ is symmetric. Let $\boldsymbol{X}_0 = \boldsymbol{A}$ and let $\boldsymbol{X}_1, \boldsymbol{X}_2, \ldots$ be the sequence generated by Equation (2) with $d = 1$, where $\alpha_k$ is determined by Equation (4) with $\ell = 1/2$ and $u = 1$. With a slight abuse of notation, we will use $\ell$ as indices of iteration counter from now on. Fix $k \ge 0$ and let $0 \le \ell \le k$. Denote $\boldsymbol{R}_\ell = \boldsymbol{I} - \boldsymbol{X}_\ell^2$. Since $d = 1$, Equation (2) simplifies to

$$\boldsymbol{X}_{\ell+1} = \boldsymbol{X}_\ell(\boldsymbol{I} + \alpha_\ell(\boldsymbol{I} - \boldsymbol{X}_\ell^2)) = \boldsymbol{X}_\ell(\boldsymbol{I} + \alpha_\ell\boldsymbol{R}_\ell),$$

and hence

$$\boldsymbol{R}_{\ell+1} = \boldsymbol{I} - \boldsymbol{X}_{\ell+1}^2 = \boldsymbol{I} - (\boldsymbol{I} - \boldsymbol{R}_\ell)(\boldsymbol{I} + \alpha_\ell\boldsymbol{R}_\ell)^2.$$

As in the proof of Theorem 4.1, define $h(x, \alpha) = 1 - (1 - x)(1 + \alpha x)^2$ so that we may write the recurrence relation with respect to $\boldsymbol{R}_\ell$ succinctly as

$$\boldsymbol{R}_{\ell+1} = h(\boldsymbol{R}_\ell, \alpha_\ell).$$

In order to see that

$$\|\boldsymbol{R}_k\|_2 \le \|\boldsymbol{R}_0\|_2^{2^{k-3}},$$

when $\alpha_\ell, 0 \le \ell \le k$, is computed as in Equation (4), we rely again on the properties of $h$ in Lemma C.1. Because $\boldsymbol{A}^2$ is symmetric, $\boldsymbol{R}_0 = \boldsymbol{I} - \boldsymbol{A}^2$ is also symmetric. Since $\boldsymbol{R}_{k+1} = h(\boldsymbol{R}_k, \alpha_k)$ and $h(x, \alpha)$ is a polynomial in $x$, it follows that $\boldsymbol{R}_\ell$ is symmetric for all $\ell$. Therefore,

$$\|\boldsymbol{R}_\ell\|_2 = \max_i |\lambda_{\ell,i}| \text{ for all } \ell,$$

where $\lambda_{\ell,i}$ denotes the $i$-th eigenvalue of $\boldsymbol{R}_\ell$. We will assume without loss of generality that the eigenvalues are ordered in such a way that $\lambda_{\ell+1,i} = h(\lambda_{\ell,i}, \alpha_\ell)$. For $\ell = 0$, because $\|\boldsymbol{X}_0\|_2 \leq 1$ and $\boldsymbol{X}_0^2$ is symmetric, the eigenvalues of $\boldsymbol{X}_0^2$ are all real-valued and lie in the interval $[0, 1]$, and therefore $0 \leq \lambda_{0,i} < 1$ for all $i$.

Let $\boldsymbol{S}_\ell \in \mathbb{R}^{p \times n}$ be random matrices consisting of i.i.d Gaussian entries $[\boldsymbol{S}_\ell]_{i,j} \sim \mathcal{N}(1, 1/p)$ with $p \geq 48(\log n + \log(1/\delta) + \log k + 41.4)$. Using standard result in randomized numerical linear algebra, for example, Proposition 3.7 in Balabanov & Nouy (2019), we know that $\boldsymbol{S}_\ell$ is a $(6, \epsilon, \frac{\delta}{kn})$-OSE for $\epsilon = 0.405$. Now fix $\ell \in \{0, 1, \ldots, k\}$ and $\boldsymbol{R}_\ell$. Let $\boldsymbol{r}_\ell^{(i)}$ denote the $i$-th column of $\boldsymbol{R}_\ell$, and let $\boldsymbol{r}_{\ell+1}^{(i)}$ denote the $i$-th column of $\boldsymbol{R}_{\ell+1} = h(\boldsymbol{R}_\ell, \alpha)$ for any $\alpha$. Since $\boldsymbol{R}_{\ell+1} = h(\boldsymbol{R}_\ell, \alpha)$ is a degree-5 polynomial with respect to $\boldsymbol{R}_\ell$, we get $\boldsymbol{r}_{\ell+1}^{(i)} \in \text{span}\{\boldsymbol{e}_i, \boldsymbol{v}_1, \boldsymbol{v}_2, \ldots, \boldsymbol{v}_5\}$ where $\boldsymbol{e}_i$ is the $i$-th standard basis vector, and $\boldsymbol{v}_j$ is the $j$-th column of $\boldsymbol{R}_\ell^j$. This means that each $\boldsymbol{r}_{\ell+1}^{(i)}$ lives in a 6-dimensional subspace of $\mathbb{R}^n$. Since we can express the squared Frobenius norm as the sum of squared column $\ell_2$ norms

$$\|\boldsymbol{R}_{\ell+1}\|_F^2 = \sum_{i=1}^n \|\boldsymbol{r}_{\ell+1}^{(i)}\|_2^2,$$

we use the $(6, \epsilon, \frac{\delta}{kn})$-OSE property of $\boldsymbol{S}_\ell$ and a union bound over $i \in \{1, 2, \ldots, n\}$ to get, with probability at least $1 - \delta/k$,

$$(1 - \epsilon)\|\boldsymbol{R}_{\ell+1}\|_F^2 = \sum_{i=1}^n (1 - \epsilon)\|\boldsymbol{r}_{\ell+1}^{(i)}\|_2^2 \leq \sum_{i=1}^n \|\boldsymbol{S}_\ell \boldsymbol{r}_{\ell+1}^{(i)}\|_2^2 = \|\boldsymbol{S}_\ell \boldsymbol{R}_{\ell+1}\|_F^2$$

and

$$(1 + \epsilon)\|\boldsymbol{R}_{\ell+1}\|_F^2 = \sum_{i=1}^n (1 + \epsilon)\|\boldsymbol{r}_{\ell+1}^{(i)}\|_2^2 \geq \sum_{i=1}^n \|\boldsymbol{S}_\ell \boldsymbol{r}_{\ell+1}^{(i)}\|_2^2 = \|\boldsymbol{S}_\ell \boldsymbol{R}_{\ell+1}\|_F^2.$$

Let $\alpha_\ell^*$ and $\tilde{\alpha}_\ell$ be computed according to Equation (3) and Equation (4), respectively, that is

$$\alpha_\ell^* = \arg\min_{\alpha \in [1/2,1]} \|h(\boldsymbol{R}_\ell, \alpha)\|_F^2, \quad \tilde{\alpha}_\ell = \arg\min_{\alpha \in [1/2,1]} \|\boldsymbol{S}_\ell h(\boldsymbol{R}_\ell, \alpha)\|_F^2,$$

then we get that, with probability at least $1 - \delta/k$,

$$(1 - \epsilon)\|h(\boldsymbol{R}_\ell, \tilde{\alpha}_\ell)\|_F^2 \leq \|\boldsymbol{S}_\ell h(\boldsymbol{R}_\ell, \tilde{\alpha}_\ell)\|_F^2 \leq \|\boldsymbol{S}_\ell h(\boldsymbol{R}_\ell, \alpha_\ell^*)\|_F^2 \leq (1 + \epsilon)\|h(\boldsymbol{R}_\ell, \alpha_\ell^*)\|_F^2,$$

and hence

$$\|h(\boldsymbol{R}_\ell, \tilde{\alpha}_\ell)\|_F^2 \leq \left(\frac{1 + \epsilon}{1 - \epsilon}\right) \|h(\boldsymbol{R}_\ell, \alpha_\ell^*)\|_F^2 < (1 + \gamma)\|h(\boldsymbol{R}_\ell, \alpha_\ell^*)\|_F^2$$

where $\gamma < 1.37$ for $\epsilon = 0.405$. Since

$$\|h(\boldsymbol{R}_\ell, \tilde{\alpha})\|_F^2 = \sum_{i=1}^n h(\lambda_{\ell,i}, \tilde{\alpha})^2,$$

we may apply Lemma C.1 and get that, for each $0 \leq \ell \leq k$,

- $\|\boldsymbol{R}_{\ell+1}\|_2 \leq \|\boldsymbol{R}_\ell\|_2^2$ if $\|\boldsymbol{R}_\ell\|_2 \geq 1/2$;

- $\|\boldsymbol{R}_{\ell+1}\|_2 \leq 1/4$ if $\|\boldsymbol{R}_\ell\|_2 \leq 1/2$;

- $\|\boldsymbol{R}_{\ell+1}\|_2 \leq 2.95\|\boldsymbol{R}_\ell\|_2^2$ with probability at least $1 - \delta/k$.

Let $\ell_1$ be such that $\|\boldsymbol{R}_{\ell_1}\|_2 \leq 1/4 < \|\boldsymbol{R}_{\ell_1-1}\|_2$. Because

$$h(h(0.75, 0.5), 0.5) < 0.25 < \|\boldsymbol{R}_{\ell_1-1}\|_2,$$

using the monotone properties of $h(x, \alpha)$ for $x \in [1/2, 1]$ and $\alpha \in [1/2, 1]$ from the proof of Claim 1, we must have

$$\|\boldsymbol{R}_{\ell_1-3}\|_2 > 0.75 > 2.95\|\boldsymbol{R}_{\ell_1}\|_2.$$

Then by induction and a union bound for $0 \leq \ell_2 \leq k$, we get that with probability at least $1 - \delta$,

$$\|\boldsymbol{R}_{\ell_1+\ell_2}\|_2 \leq \left(2.95\|\boldsymbol{R}_{\ell_1}\|_2\right)^{2^{\ell_2}} \leq \left(\|\boldsymbol{R}_{\ell_1-3}\|_2\right)^{2^{\ell_2}} \leq \|\boldsymbol{R}_0\|_2^{2^{\ell_1+\ell_2-3}},$$

for $1 \leq \ell_1 + \ell_2 \leq k$. This proves the required result for the rate of convergence. The convergence to $\text{sign}(\boldsymbol{A})$ follows exactly the same way as before.

# D. On Coefficient Bound $[\ell, u]$ for Polynomial Optimization in Higher-Order Newton-Schulz Variants

When using PRISM to accelerate the convergence of Newton-Schulz iteration for matrix sign, polar decomposition, and square roots, one needs to enforce a bound $[\ell, u]$ on the polynomial coefficient $\alpha_k$. We now discuss how to choose this bound in a principled way.

Let us start by reminding the reader with the notations that we used in Appendix A.1 for the relevant Newton-Schulz iterations. We write $f(\xi) = (1 - \xi)^{-1/2}$ and $f_d(\xi)$ the $d$-th order Taylor approximation of $f(\xi)$ around $\xi = 0$. Most notably, we have

$$f_1(\xi) = 1 + 0.5\xi, \quad f_2(\xi) = 1 + 0.5\xi + 0.375\xi^2,$$

which give rise to the 3rd- and 5th-order Newton-Schulz iteration, respectively, for the matrix sign, and hence for polar decomposition and square roots due to their connections with the matrix sign (cf. Theorem 5.2 and Theorem 5.1). In PRISM, we replace $f_d(\xi)$ with $g_d(\xi; \alpha) = f_{d-1}(\xi) + \alpha\xi^d$. For example,

$$g_1(\xi; \alpha) = 1 + \alpha\xi, \quad g_2(\xi; \alpha) = 1 + 0.5\xi + \alpha\xi^2.$$

For computing the matrix sign function with Newton-Schulz, at the $k$-th iteration, given the current iterate $\boldsymbol{X}_k$, residual matrix $\boldsymbol{R}_k = \boldsymbol{I} - \boldsymbol{X}_k^2$, and sketch matrix $\boldsymbol{S}$, finding a good value of $\alpha_k$ by naively minimizing

$$m(\alpha) = \left\| \boldsymbol{S}\left( \boldsymbol{I} - \boldsymbol{X}_k^2 g_d(\boldsymbol{R}_k; \alpha)^2 \right) \right\|_F^2$$

may lead to performance degradation. To address this, for the case $d = 1$, in Section 4 we suggested minimizing $m(\alpha)$ over an interval $\alpha \in [\ell, u]$, where $\ell = 0.5$ and $u = 1$. In Appendix C, we showed that, by constraining $\alpha_k$ in this way, the resulting polynomial has some good properties that enable us to prove Theorem 4.1 and Theorem 4.2, respectively, which state that PRISM-based 3rd-order Newton-Schulz converges quadratically in the worst case, matching the convergence rate of classical Newton-Schulz. The proof relies on Lemma C.1, which establishes several desired properties for the polynomial

$$h(x, \alpha) = 1 - (1 - x)(1 + \alpha x)^2.$$

This polynomial arises from the PRISM-based 3-rd order matrix sign iteration $\boldsymbol{X}_{k+1} = \boldsymbol{X}_k(\boldsymbol{I} + \alpha_k \boldsymbol{R}_k)$ where $\boldsymbol{R}_k = \boldsymbol{I} - \boldsymbol{X}_k^2$. One may easily verify that $\boldsymbol{R}_{k+1}$ and $\boldsymbol{R}_k$ are related by the recurrence relation

$$\boldsymbol{R}_{k+1} = h(\boldsymbol{R}_k, \alpha_k).$$

Because the residual matrix is symmetric (recall our assumption that the input matrix $\boldsymbol{A}$ satisfies $\boldsymbol{A}^2$ is symmetric), roughly speaking, the polynomial $h(x, \alpha)$ models how each individual residual value $x$ (think of an eigenvalue of the residual matrix) progresses from one iteration to the next, given the coefficient $\alpha$ in the current recurrence relation. For a precise treatment, we refer the reader to Appendix C.2. Hence, in rough words, the properties that we would like $h(x, \alpha)$ to have are the following:

1. Initial phase:

   - *Large residuals contract rapidly:* If $x$ is "large", then $h(x, \alpha)$ should be contracting towards 0 at the desired rate, for any $\alpha$ in the interval $[\ell, u]$. See, for example, Item 1 in Lemma C.1.
   - *Small residuals remain small:* If $x$ is "small", then $h(x, \alpha)$ should remain small, for any $\alpha$ in the interval $[\ell, u]$. See, for example, Item 2 in Lemma C.1.

2. Convergence phase:

   - *Point-wise convergence with optimal $\alpha$:* Let $\{x_1, x_2, \ldots, x_n\}$ be the set of all residuals (think of the set of all eigenvalues of the residual matrix). Then $h(x_i, \alpha^*)$ should be converging towards 0 at the desired rate, for all $i$, where $\alpha^* = \arg\min_{\alpha \in [\ell, u]} \sum_{i=1}^{n} h(x_i, \alpha)^2$. See, for example, Item 3 of Lemma C.1.

In general, for any $d \geq 1$, the bound $[\ell, u]$ that constrains the coefficient $\alpha$ in $g_d(\xi; \alpha)$ should be set to allow these properties. We have shown an explicit example for the case $d = 1$ in Lemma C.1. It enables us to prove Theorem 4.1 on the worst-case convergence guarantee for PRISM-based 3rd-order Newton-Schulz for matrix sign (which readily applies to polar

decomposition and square roots computation). Therefore, in Algorithm 1 and Algorithm 3, we set the input parameters to be $\ell = 0.5$ and $u = 1$. Note that these are universal constants in the sense that they do not depend on the spectral properties (or any other characteristics such as the size) of the input matrix.

For $d \geq 2$ that gives rise to higher-order PRISM Newton-Schulz variants, we can repeat the same line of reasoning and find appropriate choices for $[\ell, u]$. In the next, we give an example on how to do this for $d = 2$. The resulting $[\ell, u]$ is used in Algorithm 2 and Algorithm 4 for the PRISM-based 5th-order Newton-Schulz for polar decomposition and square root computation, respectively.

### D.1. Choosing $[\ell, u]$ for PRISM-based 5th-order Newton-Schulz and beyond

The PRISM-based 5th-order Newton-Schulz iteration for matrix sign (and similarly for polar decomposition and square root computation, due to Theorem 5.2 and Theorem 5.1) is given as

$$\boldsymbol{R}_k = \boldsymbol{I} - \boldsymbol{X}_k^2, \quad \boldsymbol{X}_{k+1} = \boldsymbol{X}_k(\boldsymbol{I} + 0.5\boldsymbol{R}_k + \alpha_k \boldsymbol{R}_k^2).$$

Hence $\boldsymbol{R}_{k+1}$ and $\boldsymbol{R}_k$ are related by the recurrence relation

$$\boldsymbol{R}_{k+1} = \boldsymbol{I} - \boldsymbol{X}_{k+1}^2 = \boldsymbol{I} - (\boldsymbol{I} - \boldsymbol{R}_k)(\boldsymbol{I} + 0.5\boldsymbol{R}_k + \alpha_k \boldsymbol{R}_k^2)^2.$$

The polynomial of interest in this case is thus

$$h(x, \alpha) = 1 - (1 - x)\left(1 + \tfrac{1}{2}x + \alpha x^2\right)^2.$$

By applying a similar line of reasoning on the analysis of monotone behaviors of $h(x, \alpha)$ and its derivatives, we can show that $h(x, \alpha)$ have the following properties:

1. *Large residuals contract rapidly:* $h(x, \alpha) \in [-3/8, x^3]$ for all $x \in [3/4, 1]$ and for all $\alpha \in [3/8, 29/20]$;

2. *Small residuals remain small:* $h(x, \alpha) \in [-3/8, 3/8]$ for all $x \in [-3/8, 3/4]$ and for all $\alpha \in [3/8, 29/20]$;

3. Let $n \geq 1$ and $x_1, x_2, \ldots, x_n \in [-3/8, 3/8]$ and $\alpha^* = \arg\min_{\alpha \in [3/8, 29/20]} \sum_{i=1}^n h(x_i, \alpha)^2$, we have

$$\max_i |h(x_i, \alpha^*)| \leq C \max_i |x_i|^3 \text{ for some constant } C < 1.7.$$

By following a similar argument in Appendix C.2 and using the above properties on $h(x, \alpha)$, one can show that, if we set $\ell = 0.375$ and $u = 1.45$, then PRISM-based 5th-order Newton-Schulz is cubically convergent in the worst case, matching the rate of classical 5th-order Newton-Schulz. This gives the worst-case convergence guarantee for Algorithm 2 and Algorithm 4, and explains why we choose $\ell$ and $u$ to be these values.

This should in principle apply to any $d$. For example, for $d = 3$, which corresponds to the 7th-order Newton-Schulz, one may analogously show that $\ell = 5/16$ and $u = 1$ is a good choice. In practice, these bounds need not to be tight. For example, for the PRISM-based 5th-order Newton-Schulz iteration, we find that setting $\ell = 3/8$ and $u = 1$ yields similar performance. The general rule is that the interval $[\ell, u]$ should include the original coefficient of the Taylor polynomial. In approximating $f(\xi) = (1 - \xi)^{-1/2}$, since the Taylor polynomials $f_d(\xi)$ approximate $f(\xi)$ from the below, we simply set the lower bound $\ell$ to be value of the coefficient in the corresponding Taylor polynomial. For example, for $d = 1$, we set $\ell = 0.5$ as $f_1(\xi) = 1 + 0.5\xi$; for $d = 2$, we set $\ell = 0.375$ as $f_2(\xi) = 1 + 0.5\xi + 0.375\xi^2$.

## E. Details of Numerical Experiments

The empirical evaluation of polar decomposition algorithms for Gaussian random matrices and HTMP random matrices is run in single precision (i.e., `torch.float32`) on an Nvidia A100 GPU. For the numerical experiment in Figure 4, the matrix $\boldsymbol{A} \in \mathbb{R}^{n \times m}$ has size $n = 8000$ and $m = 4000$.

In the experiment with the Shampoo optimizer, we made the standard ResNet-20 and ResNet-32 slightly larger to demonstrate more clearly the difference between different algorithms for matrix square root when the input matrix has reasonably large size, e.g., larger than 100. For both ResNet-20 and ResNet-32, we kept the stride at one for each convolutional layer; for

ResNet-20, we additionally removed the average pooling layer before the final fully connected layer. We set the maximum preconditioner dimension to 2048 in the Distributed Shampoo optimizer (Shi et al., 2023), so the matrices of which we need to compute the inverse square root have dimension at most 2048 x 2048. We also tested setting the maximum preconditioner dimension to 1024 and 4096, respectively, and the results are similar. Generally, we find that the larger dimension the preconditioner has, the better performance PRISM has relative to eigenvalue decomposition and PolarExpress. We set the learning rate to 0.001 and the weight decay to 0.0005. We did not tune hyperparameters for this experiment. In this experiment we used the 5th-order Newton-Schulz iteration accelerated by PRISM. Other variants of Newton-Schulz, such as the one that uses a degree-3 polynomial update at each iteration, can also be accelerated by PRISM. In our experiment, we also tried to use five iterations of this variant to compute the inverse square root of Shampoo's preconditioners, and we got similar results. The PolarExpress algorithm that we use in our experiment is the one that is optimized for $\sigma_{\min} = 10^{-3}$ (i.e. Algorithm 1 of Amsel et al. (2026)).

In the experiment with Muon optimizer in Figure 6, we use PolarExpress (Algorithm 1 of Amsel et al. (2026)), PRISM-based Newton-Schulz iteration with degree-5 polynomial (PRISM-5), and PRISM-based Newton-Schulz with degree-3 polynomial (PRISM-3) to compute the polar decomposition of the gradient matrices. In this experiment, we use five iterations for PolarExpress and PRISM-3, and three iterations for PRISM-5. Empirically, we observed that PRISM-5 with 3 iterations attains lower average residual error than PolarExpress with 5 iterations. For instance, at final step in Figure 6, the average Frobenius norm error over all layers for PRISM-5 with 3 iterations and PolarExpress with 5 iterations are 12.7 and 23.4. Hence, for the computational efficiency, we only take 3 iterations for PRISM-5 at each step of training. In addition, based on the computed $\alpha_k$ in Figure 4, we observe that at the initial several iterations, the coefficient $\alpha_k$ is attained at the upper bound $u$ in (4). Hence, when we implement PRISM-3 and PRISM-5 in Figure 6, we decide to use the highest value of $\alpha_k$ for the initial three iterations for efficiency. This means that we set $\alpha_k = 1$ for PRISM-3 and $\alpha_k = 29/20$ for PRISM-5 for the first three iterations. Note that, by Lemma C.1, setting $\alpha_k$ in this way preserves the initial quadratic convergence of PRISM-3. Additionally, we choose weight decay 0.01, momentum parameter 0.95, and initial learning rate $6 \times 10^{-3}$, and micro-batch size 4. We also compare these experiments with the baseline using AdamW with initial learning rate $3 \times 10^{-4}$ and weight decay 0.1. All experiments are run on NVIDIA A100-SXM4-80GB with global batch size 32.

## F. Additional Empirical Results

- Figure 7 and Figure 8 show the convergence of degree-5 polynomial methods with respect to the number of iterations. These figures compensate for Figure 3 and Figure 4, respectively, where we illustrate the convergence of algorithms with respect to the wall-clock time when running on a single Nvidia A100 GPU.

- Figure 9 and Figure 10 show the convergence of degree-5 polynomial methods for computing square roots.

- Figure 11 shows the accelerated convergence behavior of the PRISM-based DB Newton iteration (see Appendix A.2 for details). We observe that, when accelerated by PRISM, the Newton iteration can converge faster than the PRISM-based Newton-Schulz. However, because of the requirement to perform matrix inversion at every iteration, Newton iteration is generally less stable than the Newton-Schulz variants. An interesting future work is to exploit the faster convergence of PRISM-based Newton iteration while maintaining good numerical stability.

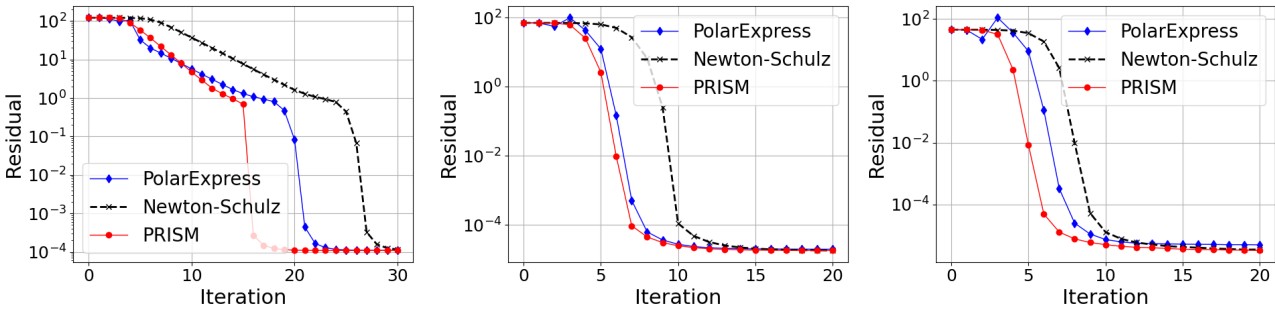

*Figure 7.* Convergence (with respect to iterations) of degree-5 polynomial methods for orthogonalizing a Gaussian random matrix $A \in \mathbb{R}^{n \times m}$ with varying aspect ratio $\gamma = n/m$. The figures show the Frobenius norm error $\|I - X_k^T X_k\|_F$ for $\gamma = 1, 4, 50$, from left to right respectively.

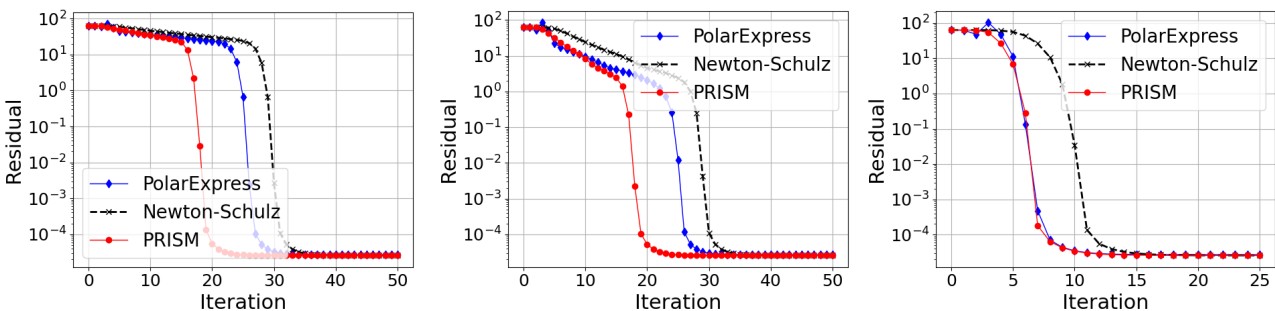

*Figure 8.* Convergence (with respect to iterations) of degree-5 polynomial methods for othogonalizing random matrices generated by HTMP ([Hodgkinson et al., 2025](#)) with different parameter $\kappa$. Smaller $\kappa$ indicates heavier tail in the spectral distribution. The figures show the Frobenius norm error $\|\boldsymbol{I} - \boldsymbol{X}_k^T \boldsymbol{X}_k\|_F$ for $\kappa = 0.1, 0.5, 100$, from left to right respectively.

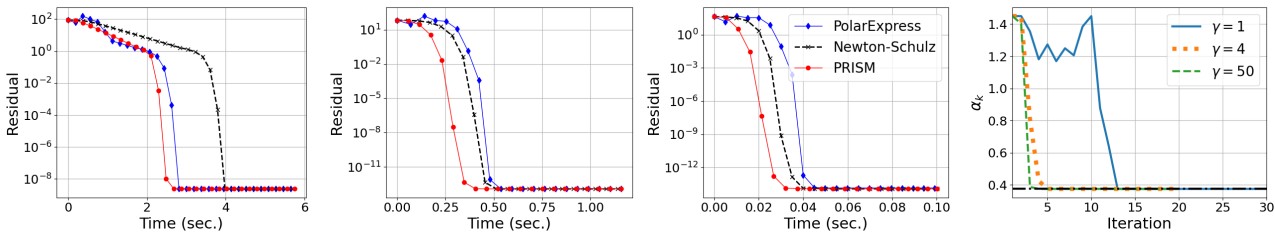

*Figure 9.* Convergence of degree-5 polynomial methods for computing the square root and inverse square root of $\boldsymbol{A} = \boldsymbol{G}^T \boldsymbol{G}$, where $\boldsymbol{G} \in \mathbb{R}^{n \times m}$ is a Gaussian random matrix with varying aspect ratio $\gamma = n/m$. That is, $\boldsymbol{A}$ is a Wishart matrix. The figures from left to right show the Frobenius norm error $\|\boldsymbol{I} - \boldsymbol{X}_k^{-2} \boldsymbol{A}\|_F$ for $\gamma = 1, 4, 50$, respectively. The last figure on the right shows the $\alpha_k$'s computed by ([4](#)) in PRISM for different aspect ratios.

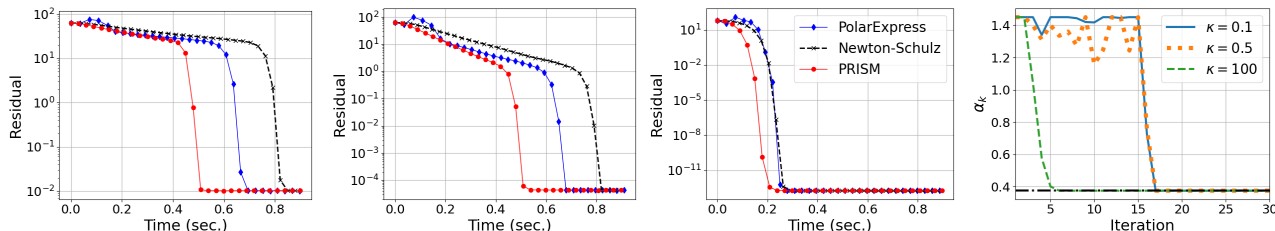

*Figure 10.* Convergence of degree-5 polynomial methods for computing the square root and inverse square root of $\boldsymbol{A} = \boldsymbol{G}^T \boldsymbol{G}$, where $\boldsymbol{G} \in \mathbb{R}^{n \times m}$ is a random matrix generated by HTMP ([Hodgkinson et al., 2025](#)) with different parameter $\kappa$. Smaller $\kappa$ indicates heavier tail in the spectral distribution. The figures from left to right show the Frobenius norm error $\|\boldsymbol{I} - \boldsymbol{X}_k^{-2} \boldsymbol{A}\|_F$ for $\kappa = 0.1, 0.5, 100$, respectively. The last figure on the right shows the $\alpha_k$'s computed by ([4](#)) in PRISM for different $\kappa$'s.

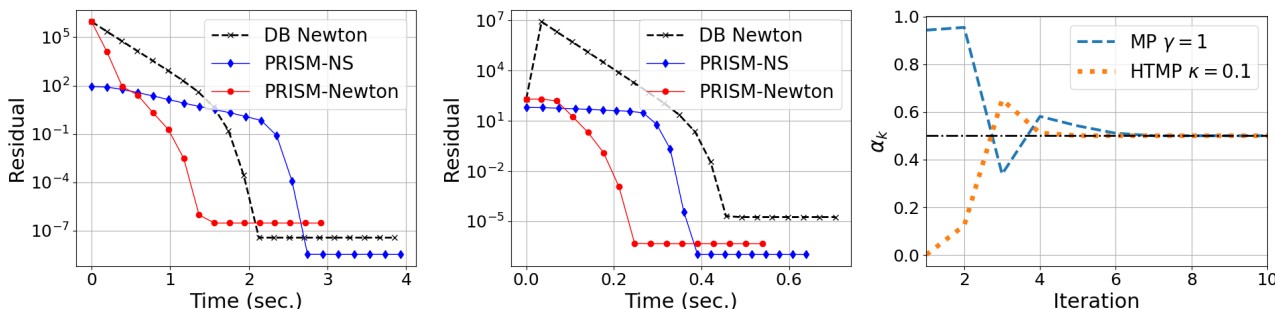

*Figure 11.* Convergence of PRISM-based DB Newton (PRISM-Newton, cf. Table [1](#) and Appendix [A.2](#)) for computing the square root and inverse square root. We compare with the classical DB Newton iteration. For reference we also compare with the PRISM-based Newton-Schulz (PRISM-NS) that we tested in the previous experiment. We select two representative input matrices from the previous experiment: (Left) Wishart matrix with aspect ratio $\gamma = 1$; (Middle) random matrix generated by HTMP with $\kappa = 0.1$. The rightmost plot shows the coefficient $\alpha_k$ computed by PRISM-Newton.

