# OpenReview forum: "PRISM: Distribution-free Adaptive Computation of Matrix Functions for Accelerating Neural Network Training"
_ICML.cc/2026/Conference — ICML 2026 regular_

### Official Review · Reviewer_sDQC · 2026-03-04

**Soundness:** 2
**Presentation:** 3
**Significance:** 3
**Originality:** 2
**Overall Recommendation:** 4
**Confidence:** 3

**Summary:**

The paper introduces a new algorithm called PRISM, designed to accelerate the computation of matrix functions, especially the matrix sign and square-root functions. PRISM builds a residual term and utilizes an iterative polynomial approximation. To further enhance the convergence rate, PRISM incorporates a spectrum-adaptive polynomial by solving a least-squares problem at each step. A sketching method is used to reduce the computational cost of the subproblem. Numerical experiments show that the proposed algorithm has good performance on orthogonalizing random matrices and training neural networks.

**Compliance With Llm Reviewing Policy:**

Affirmed.

**Final Justification:**

The rebuttal addresses my concerns, so I will keep the score of 4.

**Key Questions For Authors:**

1.Numerical experiments on neural network training only consider ResNet and GPT-2, which are much smaller in scale compared to currently used language models. More experiments on large-scale LLMs (perhaps 1B to 8B) may enhance the generalizability of the algorithm.

2.For acceleration of the Taylor expansion, the method appears to adapt only the highest-order coefficient in $g(R_k)$. An ablation study on this choice may demonstrate the reason for it. A discussion on the correlation between the number of learnable parameters in $g(R_k)$ and the convergence rate might be helpful to clarify why this parameterization is chosen.

3.The theorems mainly focus on analyzing the case $d=1$ and fixed upper and lower bounds $\ell$ and $u$. However, in practice there might be usage of higher-order Taylor expansions and different $\ell$ and $u$. Could there be related theorems designed for more practical use of these approximation methods?

4.For $d=2$, the choice of the upper and lower bounds seems tricky as $[3/8,29/20]$. More discussion on different choices of this kind would help strengthen the understanding of the proposed algorithm and the effect of the eigenvalues on the algorithm.

**Limitations:**

yes

**Strengths And Weaknesses:**

The paper provides a well-designed algorithm called PRISM for matrix functions which are commonly occurred in optimization algorithms such as Muon and Shampoo. It provides a comprehensive review on GPU-friendly methods to calculate matrix square root and polar decomposition. PRISM consists of a high-level structure form of a meta-algorithm and a practical one which incorporates diverse acceleration techniques to further accelerate the method.

The method analyzes the problem from a residual perspective. A modified polynomial approximation is used to deal with the case when the eigenvalue is close to zero. A sketch-based method is utilized to reduce the computational dimension of the parameter $\alpha$. Relative theorems are provided to validate the choice of the hyper-parameters and numerical design.

The paper is clearly structured and each part is tightly linked to the next. The algorithm shows good performance on diverse test problems. It shows great enhancement compared to PolarExpress when the smallest eigenvalue is relatively large.

The paper also has some weaknesses, listed as follows.

1. The proposed algorithm is only tested on small-scale problems. There is no experiment training large language models (size of nearly 1B) using the proposed algorithm with Muon.

2. Next, although it is described as spectrum-adaptive without prior spectral knowledge, the method also relies on tuning hyperparameters related to prior spectral knowledge, such as $\ell$ and $u$. Besides, proposed theoretical guarantees are also strongly connected to this choice of parameters. There are few ablation studies on these hyperparameters from an experimental perspective.

3. The proposed algorithm seems to be just a combination of residual-based iterative algorithms and a sketching method. Ablation studies on the effect of these parts may offer more insight on this proposed algorithm. More discussion on the benefits of the residual based algorithm structure would be helpful for the algorithm class.

4. Last but not least, there are typos in the paper, which partially undermine the soundness of the overall paper. For line 192, the update formula $X_{k+1} = \frac{1}{2}X_k - \frac{3}{2} X_k^3$ is not consistent with the derivation in the surrounding text and might be a typo. For line 242, ``PRIME meta-algorithm'' is not consistent with the rest of the paper. The authors should check typos of this kind in the paper carefully.

---

> ### Author Rebuttal · Authors · 2026-03-31
>
> We thank the reviewer for the careful reading of our manuscript and favorable feedback. We have fixed the typos the reviewer mentioned in the paper. In the following, we address the reviewer's questions and concerns.
>
> **W1\& Q1:** We agree that large-scale LLM experiments are an important stress test. The focus of this work is on methodology and numerical experiments at reasonable scale. PRISM is a general-purpose acceleration framework for matrix-function, not an optimizer itself. Our experiments focus on GPT-2, which is a good testbed for early-phase optimizer development before scaling up. Many other variants of Muon [1,2,3] also only consider GPT-2 to test their methods. In the revision we'll add more experiments: GPT-2-Large (774M) and Llama-1.2B model studied in [4]. In a follow-up work, we plan to work with an external team and scale up the experiments [4] for large-scale LLMs (we’re limited by compute to tune hyperparameters, as [3,4] did).
>
> **W2 \& Q4:** We clarify that the bounds $[\ell, u]$ are not tuning hyperparameters and do not rely on spectral knowledge. They are selected analytically using algebraic properties of the polynomial. The goal is to ensure that the polynomial $g_d(\xi) = f_{d-1}(\xi) + \alpha_k^* \xi^d$ retains convergence guarantees comparable to the Taylor polynomial, so that an analogue of Theorem 4.1 holds for all $d \ge 2$. From the proof of Theorem 4.1, the key requirement is that for all $\alpha \in [\ell,u]$: (i) large residuals contract rapidly (item 1 Lemma B.1), and (ii) small residuals remain bounded (item 2 Lemma B.1). For $d \ge 2$, we enforce analogous conditions. When $d=2$, with $h(x,\alpha) = 1 - (1-x)(1 + 0.5x + \alpha x^2)^2,$choosing $[\ell,u] = [3/8, 29/20]$ ensures:
> - $h(x,\alpha) \in [-3/8, x^3]$ for $x \in [3/4,1]$ (rapid contraction),
> - $h(x,\alpha) \in [-3/8, 3/8]$ for $x \in [-3/8,3/4]$ (stability),
> - for $\alpha^* = \arg\min_{\alpha \in [\ell,u]} \sum_i h(x_i,\alpha)^2$, $\max_i |h(x_i,\alpha^*)| \le C \max_i |x_i|^3$ with $C<1.7$.
>
> These conditions guarantee cubic convergence. In practice, the bounds need not be tight; empirically, $[3/8,1]$ performs similarly. The same construction extends to $d \ge 3$ (e.g., $d=3$: $[\ell,u]=[5/16,1]$). We set $\ell$ to the Taylor coefficient since $f(\xi)=(1-\xi)^{-1/2}$ is approximated from below, so increasing this coefficient improves approximation. Thus, $[\ell,u]$ are analytically derived to enforce favorable polynomial behavior, independent of spectral information. We will include a general derivation in the revision.
>
> **W3:** Thank you for the suggestion. Sketching and PRISM are complementary: while PRISM improves convergence per iteration, sketching is essential for reducing per-iteration cost and achieving wall-clock gains.
>
> We evaluate polar decomposition on an $8000 \times 4000$ matrix (Fig 4, left). To reach $10^{-4}$ Frobenius error:
>
> - **Iterations:** NS: 32; PRISM NS: 20; PRISM NS (no sketch): 20; PolarExpress: 28
>
> - **Wall-clock time (s):** NS: 1.138; PRISM NS: 0.722; PRISM NS (no sketch): 1.311; PolarExpress: 0.998
>
> Thus, without sketching, PRISM retains fast convergence (same iteration count) but becomes the slowest due to higher per-iteration cost, whereas sketching yields a $\sim$1.8× speedup.
>
> Conceptually, PRISM’s minimum-residual formulation connects to sketched least-squares methods in RandNLA and optimization (e.g., Newton–Sketch [5]). But, since the approximation space is naturally represented in factorized matrix form, vectorizing into a standard least-squares problem is inefficient. Our method applies sketching directly to matrix residuals, preserving structure while improving efficiency.
>
> **Q2:** We optimize a single coefficient to retain a closed-form solution; optimizing multiple coefficients leads to a multivariate polynomial problem requiring numerical solvers. We choose the highest-degree coefficient because, in the $d$-th order Taylor expansion of $f(\xi) = (1-\xi)^{-1/2}$ around $\xi=0$, perturbing the $\xi^d$ term minimally affects the approximation near $\xi \approx 0$. This preserves local accuracy while allowing greater flexibility to improve approximation near $\xi \approx 1$, leading to better overall performance.
>
> **Q3:** Yes! For each $d\ge1$, we recommend using fixed $\ell$ and $u$ based on analogous theorems to Theorem 4.1, which covers $d=1$. The same reasoning applies to $d\ge2$ and guarantees analogous convergence. For details, see our response to W2. In the revised paper, we’ll provide derivations of $\ell,u$ for $d=2$ and $d=3$, so practitioners can use them.
>
> ---
> [1] The polar express: Optimal matrix sign methods and their application to the muon algorithm
>
> [2] Accelerating Newton-Schulz Iteration for Orthogonalization via Chebyshev-type Polynomials
>
> [3] Beyond the ideal: Analyzing the inexact muon update
>
> [4] Fantastic pretraining optimizers and where to find them
>
> [5] Newton Sketch: A Linear-time Optimization Algorithm with Linear-Quadratic Convergence.

---

> > ### Author Rebuttal · Reviewer_sDQC · 2026-04-02
> >
> > Thank you for addressing my questions and for including additional experiments. I will keep the score of 4.

---

### Official Review · Reviewer_Mwff · 2026-03-11

**Soundness:** 4
**Presentation:** 4
**Significance:** 3
**Originality:** 4
**Overall Recommendation:** 5
**Confidence:** 4

**Summary:**

PRISM is a meta-algorithmic framework for accelerating iterative matrix function algorithms (square roots, orthogonalization) without requiring prior knowledge of the spectral range. It adaptively fits a polynomial to the current spectrum using cheap randomized sketching, replacing the fixed polynomials used in methods like Newton-Schulz. The framework is integrated into the Muon optimizer for numerical evaluation, where outperforms PolarExpress (and the standard AdamW optimizer).

**Compliance With Llm Reviewing Policy:**

Affirmed.

**Final Justification:**

I like the idea of the paper; the rebuttal was short but partially addressed my concern. I keep my score at "accept"

**Key Questions For Authors:**

In Section 6.2 you write "The validation accuracy throughout the first 50 epochs is shown in Figure 5. The ranking stays the same when we keep training longer." Also in Figure 6 you only show results for the first 800 seconds of training. Your writing indicates that you ran training for longer, why do you only report the losses and accuracies in the beginning? Does the gap in performance (Figure 6) persist at later epochs?

**Limitations:**

yes the authors have an "Impact statement" but limitations are not discussed in their conclusion.

**Strengths And Weaknesses:**

Soundness:
- The overall meta-algorithm is well-justified and properly theoretically investigated
- Randomized sketching is justified with a Johnson-Lindenstrauss argument preserving the convergence rate.
- Experiments are limited to relatively small models; scalability to large-scale settings is not directly demonstrated.

Presentation:
- Clearly written; the meta-algorithm is well-described and the linear algebra is easy to follow
- Table 1 outlines iterations for different matrix functions, which is very useful

Significance:
- Matrix functions in optimizers are a bottleneck at scale; a drop-in distribution-free replacement is practically valuable.
- Integration into Muon shows benefits transfer to large-scale training workloads.

Originality:
- The combination of adaptive polynomial fitting and randomized sketching in a unified meta-algorithm template is original.
- Clearly distinguished from previous approaches like PolarExpress

---

> ### Author Rebuttal · Authors · 2026-03-31
>
> We thank the reviewer for the careful reading of our manuscript and favorable evaluation. In the revision of our paper, we will add more experiments: GPT-2-Large model (774M parameters) on 1 Billion tokens of the Fineweb data and Llama-1.2B model studied in [1] to enhance the generalizability of the algorithm. In the following, we address the reviewer's question. We are happy to discuss more if the reviewer has further questions.
>
> **Q1:** Yes, we did train longer. For the experiments shown in Figure 5, we trained for 100 epochs, but the losses remain roughly the same for all methods towards the end. So we plot the loss trajectory for the first 50 epochs for illustration purposes. For the GPT2 experiment in Figure 6, we trained the model for a single epoch, similar to the empirical setting used in [1]. This is in part due to limited compute we had access to. For a follow-up work, we are planning to work with an external team and scale up the experiments.
>
> ---
> [1] Wen et al. Fantastic pretraining optimizers and where to find them.

---

> > ### Author Rebuttal · Reviewer_Mwff · 2026-03-31
> >
> > Thank you for adding the additional experiments.

---

### Official Review · Reviewer_q22h · 2026-03-12

**Soundness:** 3
**Presentation:** 4
**Significance:** 2
**Originality:** 2
**Overall Recommendation:** 3
**Confidence:** 2

**Summary:**

This paper introduces PRISM, a novel framework for the adaptive computation of matrix functions (such as matrix sign, square root, and polar decomposition) using polynomial fitting and randomized sketching. The method primarily relies on General Matrix Multiplications (GEMMs), which are highly efficient on modern GPU accelerators. The authors propose a two-part meta-algorithm: Part I establishes a baseline iterative procedure (e.g., Taylor expansion-based Newton-Schulz), and Part II accelerates it by dynamically fitting a polynomial surrogate to the current residual's spectrum via sketched least-squares. The authors integrate PRISM into second-order optimizers like Shampoo and Muon. Experimental results demonstrate that PRISM achieves faster convergence in terms of wall-clock time compared to existing baselines like PolarExpress.

**Compliance With Llm Reviewing Policy:**

Affirmed.

**Final Justification:**

The authors' response has addressed one of my main previous concerns: that similar methods might have appeared in numerical analysis literature. The authors clarified that such methods cannot be incorporated into classical approximation methods and highlighted the differences in the prior paradigm. Therefore, I believe the authors have made a contribution in terms of originality, which has led to an increase in the final score.

**Key Questions For Authors:**

I do not fully understand why adaptivity is only applied to the highest-degree polynomial. Is this due to algorithmic cost considerations? Why was the highest degree chosen?

As the authors emphasize in the claims regarding the Dion algorithm, communication and computation are equally important in the time cost of the Muon optimizer. What is the impact of the additional sketching component introduced in PRISM_Distribution_free_-6.pdf on communication?

Regarding Theorem 4.1, does the convergence rate differ when utilizing the vanilla NS?

**Limitations:**

Beyond the weaknesses mentioned above, I do not see any additional major limitations.

**Strengths And Weaknesses:**

Strengths:

The paper clearly explains its motivation and target.

The overall logic of the paper is very clear.

The mathematical formulas are well-utilized, and the layout of the paper is excellent.

Weaknesses:

Methodological: The relationship between this method and prior work is insufficiently discussed. Although the related work mentions many previous matrix approximation and adaptive methods, it lacks a comparative analysis regarding the relationship with the proposed approach. The existence of adaptive, matrix-multiplication-based polynomial approximation methods is not deeply explored. Readers would expect a clear assertion on whether such methods previously existed or what specifically distinguishes this method from those prior works.

Experimental Scope: As a paper proposing a general-purpose method, it does not seem to explore applications beyond optimization. I believe providing such applications is necessary for a method claimed to be general.

Experimental Setup: In the optimization experiments, the number of tokens used is relatively small. This is not a concern regarding the volume of experiments, but rather a concern about the fairness of the comparison. In the later stages of training, the statistical distribution of momentum singular values tends to stabilize. In such scenarios, non-adaptive methods might have an advantage. Furthermore, the period where statistics are stable constitutes the majority of the optimization process.

---

> ### Author Rebuttal · Authors · 2026-03-31
>
> We thank the reviewer for providing valuable comments on our work. In the following, we address the reviewer's questions and concerns. We are happy to discuss more if the reviewer has further questions. We hope the our clarifications address the reviewer’s concerns.
>
> **W1:** To our knowledge, this is the first adaptive, matrix-multiplication-based polynomial approximation method (when PRISM is applied to accelerate Newton-Schulz) that does not make any assumption on input eigenvalues. There are adaptive rational approximation methods, which require matrix inversion at every iteration, such as the scaled Newton iteration for matrix sign under 2-norm scaling, spectral scaling and determinantal scaling [1]. We also note that there is substantial prior work on adaptive polynomial approximation via Krylov methods for matrix-function actions $f(A)b$, especially for linear solves $A^{-1}b$. However, these methods are fundamentally first-order and operate in a different computational regime: they build approximants through matrix-vector products and the incremental expansion of an approximation space, with adaptivity mainly through flexible preconditioning and deflation strategies. We will make this point clear in the revised paper.
>
> **W2:** The primary goal in this work is to apply PRISM and accelerate the matrix computations used in optimizers. We expect to see the most amount of gain in this application because optimizers that utilize matrix-based preconditioners, such as Muon and Shampoo, require repeated computation of matrix functions. Having said that, we completely agree that PRISM can be very useful more broadly.
>
> Here are two additional examples beyond optimization. The first is ZCA whitening. Let $\hat{\Sigma}$ denote the estimated sample covariance matrix, one needs to compute $\hat{\Sigma}^{-1/2}$ explicitly. We follow the empirical setup of [2] to compute $\hat{\Sigma}^{-1/2}$ where $\hat\Sigma$ is the sample covariance matrix from a batch of flattened tinyImageNet images. The size of $\hat\Sigma$ is 12288 x 12288. Here is the time (in seconds) it takes for each method to converge to machine precision. Original NS: 14.3; PRISM NS: 9.6; PolarExpress: 10.3. As expected, we see that PRISM-accelerated Newton-Schulz is significantly faster than the original Newton-Schulz. Note that this aligns with the plots in Figure 9 and Figure 10, where we show speedup of PRISM-based NS for computing square roots. The other example is in the computation of the 2-Wasserstein metric for Gaussians [3], which requires computing two matrix-square-roots sequentially. Again, we expect a similar speedup as illustrated in Figures 9 and 10.
>
> **W3:** Even if the singular values tend to stabilize in later stages of training, their bounds (i.e. the smallest and the largest singular values for each weight matrix) as well as their statistical distribution can still differ dramatically across different weight matrices that correspond to different layers. In addition, these values depend on, among many other factors, model architecture and data. Therefore, assuming that the singular values do not tend to change much over time, it is still necessary to use adaptive methods, as the empirical singular value distribution will inevitably differ across different locations in the model architecture.
>
> **Q1:** The reviewer is right; we only optimize for a single coefficient because otherwise no closed-form solution exists, and one has to call a numerical solver and minimize a multi-variate polynomial. We chose the coefficient that corresponds to the highest degree term because, from the d-th order Taylor's expansion of $f(\xi) = (1-\xi)^{-1/2}$ around $\xi = 0$, a unit change in the coefficient of $\xi^d$ leads to the minimum distortion in the approximation for small $\xi$ near 0. Intuitively, this allows the resulting polynomial to provide a better approximation of $f(\xi)$ when $\xi$ is close to 1, while maintaining a good approximation of $f(\xi)$ when $\xi$ is close to 0.
>
> **Q2:** In this work, our focus is on improving the computation cost rather than the communication cost. We agree that designing communication-efficient implementation can be an interesting follow-up work. In addition, please allow us to clarify that we only mentioned DION on line 163, and there is no claim that ''communication and computation are equally important...''.
>
> **Q3:** No, the convergence rate is the same. The purpose of Theorem 4.1 is to provide a robustness guarantee that, in the worst case, PRISM-accelerated Newton-Schulz converges at least as fast as the vanilla Newton-Schulz. (Although in all of our numerical experiments, PRISM-accelerated version always converges much faster.)
>
> ---
> [1] On Scaling Newton's Method for Polar Decomposition and the Matrix Sign Function.
>
> [2] MatRL: Provably Generalizable Iterative Algorithm Discovery via Monte-Carlo Tree Search.
>
> [3] Learning from uncertain curves: The 2-Wasserstein metric for Gaussian processes.

---

> > ### Author Rebuttal · Reviewer_q22h · 2026-04-08
> >
> > I thank the authors for their detailed response. They clarified the distinction between their method and prior work, which addressed my main concern about originality, and accordingly I raised my final score.

---

### Official Review · Reviewer_QRht · 2026-03-13

**Soundness:** 3
**Presentation:** 3
**Significance:** 3
**Originality:** 3
**Overall Recommendation:** 5
**Confidence:** 3

**Summary:**

This paper aims to accelerate the computation of fundamental matrix functions, such as $A^{-1}$, $A^{-1/2}$, and $UV^\top$ (i.e., orthogonalization), using General Matrix Multiplications (GEMMs). Unlike prior works, which are restricted in their broader applicability and depend heavily on the singular value distribution of the input matrix, the proposed PRISM method can be applied to a wider class of problems without requiring any prior singular value information. The PRISM algorithm replaces the coefficient of the $d$-th degree term in the Taylor approximation with an optimized value, $\alpha^\*$, by minimizing the residual norm. To reduce computational costs, the method approximates $\alpha^*$ as $\tilde{\alpha}_k$ via matrix sketching. The authors conducted comprehensive experiments to validate the algorithm, demonstrating how quickly the residual can be reduced relative to actual wall-clock time. Furthermore, neural network experiments, specifically training ResNet on CIFAR datasets and GPT-2 on the FineWeb dataset, confirm that PRISM achieves faster improvements in validation accuracy and more rapid loss reduction over training time compared to baselines.

**Compliance With Llm Reviewing Policy:**

Affirmed.

**Final Justification:**

During the rebuttal period, the authors’ response and the newly provided experimental results addressed my main concerns, including the choice of $[\ell, u]$, the missing training accuracy, and the question of whether the loss gap increases over time. Therefore, I increase the score to 5.

**Key Questions For Authors:**

1. How does the convergence speed of PRISM scale with respect to the matrix dimensions $n$ and $m$ for an input matrix $A \in \mathbb{R}^{n \times m}$? I would appreciate a discussion or empirical comparison of this scaling behavior against baseline methods.
2. Based on the description in Appendix C, it appears that PRISM5 was only run for 3 iterations in the experiments shown in Figure 6. Does this not result in the Frobenius norm error (residual) remaining at a substantially large value (according to the result from Figure 7)? I would like to see the actual residual values at each training step when PRISM is applied in the neural network experiments, especially in comparison to the original Newton-Schulz (NS) and PolarExpress methods.
3. Why was the highest possible value of $\alpha_k$ explicitly chosen for the first few iterations in the practical neural network experiments? Was this a heuristic decision made specifically to reduce training costs? I am curious to see what the performance plots would look like if the algorithm did not rely on this prior knowledge.

**Limitations:**

The authors do not adequately discuss the limitations of their work. It would be highly beneficial to include a dedicated section addressing the potential failure modes of PRISM, specifically highlighting scenarios or types of matrices where the algorithm struggles to perform optimally.

**Strengths And Weaknesses:**

### **Strengths**

1. The paper is well-organized and easy to follow. Although some sections are highly technical, this is largely unavoidable given the nature of the field. The authors make a commendable effort to break down these complex concepts using clear examples and informative plots. They effectively highlight the limitations of prior works, specifically regarding their restricted applicability and reliance on singular value distribution information, which clearly establishes the need for a new approach. The subsequent sections introduce the PRISM algorithm logically, providing detailed explanations of how it operates and justifying the necessity of each component. Finally, the authors thoroughly validate their claims, demonstrating both empirically and theoretically that PRISM converges rapidly, and successfully proving its practical effectiveness in neural network training scenarios.
2. The proposed method demonstrates strong novelty and successfully resolves the limitations of prior works identified in the introduction. The core approach adaptively tunes the coefficient of the $d$-th order term in the Taylor approximation by dynamically updating it at each iteration. Crucially, to mitigate the computational overhead of this process, the authors cleverly employ randomized sketching to project the $n \times n$ matrix calculations into a lower-dimensional $p \times n$ space. This effectively reduces the overall computational cost from $O(n^3)$ to $O(n^2 p)$. Furthermore, the authors provide robust theoretical and empirical validation to support these algorithmic improvements.

### **Weaknesses**

1. Concerns regarding the experimental evaluation:
    - Given that the title explicitly highlights the acceleration of neural network training, the scope of the neural network experiments is limited. The paper needs a broader range of experiments across different architectures and datasets to robustly support its main claims.
    - Figures 5 and 6 only display a zoomed-in view of a specific, narrow time period. The authors should provide plots over an extended time axis. Because PRISM is designed to reduce the computational time per iteration, it is important to see if the performance gap between PRISM and the baselines continues to widen as training progresses. Additionally, providing a plot of wall-clock time versus iteration count would clearly illustrate the per-iteration speedup.
    - In the Figure 6 experiments, the most critical baseline---the Newton-Schulz based Muon optimizer---is absent. Comparing the proposed method against AdamW is an unfair and incomplete comparison, especially for a paper focused on matrix orthogonalization techniques.
    - For the image classification experiments in Figure 5, the authors only plot validation accuracy. Since the core objective of PRISM is to accelerate the *training* process itself, it is essential to plot the training accuracy alongside the validation metrics to properly demonstrate the method's effectiveness.
2. As I understand it, the algorithm finds the optimal coefficient $\alpha_k^\*$ for the $d$-th degree term by minimizing the Frobenius norm of the residual matrix at each $k$-th iteration. However, isn't this inherently a greedy approach? Could you clarify or provide theoretical justification on whether greedily optimizing $\alpha_k^*$ at each individual step guarantees global optimality or ensures optimal convergence across the entire iterative process?
3. The choice of the bounds $[\ell, u]$ for constraining $\alpha_k$ is questionable. The authors mention that these values were selected empirically, but they do not detail the selection process or provide an analysis of the algorithm's robustness to different interval choices. Because PRISM is proposed as a meta-algorithm designed to be applied on top of existing or future algorithms, it is unclear if this empirically chosen range will generalize well to new methods. Furthermore, the paper lacks guidance on what constraint values should be used for higher-order polynomials ($d \geq 3$). Finally, while the manuscript explicitly discusses the necessity of the upper bound $u$, it entirely omits any rationale or justification for enforcing the lower bound $\ell$.
4. To enhance both the reproducibility and readability of the paper, the authors should include explicit pseudo-code demonstrating exactly how the PRISM algorithm is integrated with specific baseline methods. At a minimum, providing the pseudo-code for representative algorithms, such as the Newton-Schulz iteration, is highly recommended.


**Minor Corrections and Typos**

1. Line 86: “RRISM” → “PRISM”
2. Line 161: “$X_{k+1} = X_k f_1(R_k)$” → “$X_{k+1} = X_k f_d(R_k)$”
3. Figure 2 (Left): Please include a y-axis label (e.g. “Function value”)
4. Figure 4, 5, 6, and 7 (Captions): “othogonalizing” → “orthogonalizing”

---

> ### Author Rebuttal · Authors · 2026-03-31
>
> We thank the reviewer for the careful reading and constructive feedback. We'll add pseudo-code for all representative algorithms and release code. Below we address all concerns and questions:
>
> **W1:**
> - We'll add a diffusion model, FLEX, which integrates a vision transformer with traditional U-Net-style ResNet for modeling spatio-temporal physical systems [1]. We use FLEX-medium (about 200M parameters) in our experiment in the revision. We train it on a spatio-temporal super-resolution task using a subset of the ERA5 reanalysis dataset from the SuperBench benchmark. We compare different methods for computing the inverse root preconditioner. The table below shows the best validation loss up to a certain time. Computing Shampoo’s preconditioner with PRISM yields the lowest loss because it’s faster than eigendecomposition and more accurate than the original NS (we ran both for 5 iterations). We’ll include a plot of the loss trajectories in the revised paper.
> | Time (sec) | Eigendecomp | Newton-Schulz (NS) | PRISM |
> |--|--|--|--|
> | 1000 | 0.2415 | 0.2277 | **0.2242**|
> | 2000 | 0.2198 | 0.2156 | **0.2132**|
> | 3000 | 0.2061 | 0.2072 | **0.2024**|
> | 4000 | 0.1932 | 0.1982 | **0.1885**|
> | 5000 | 0.1827 | 0.1853 | **0.1751**|
>
> - We'll add full training plots in the revision and iteration-wise wall-clock time speedup. For the inverse square root computation used inside the Shampoo, the speedup depends on Shampoo's configuration on the maximum size of its preconditioner (the larger the better). We will also add per-iteration speedups (Shampoo inverse square root): max size 2048: 12.8→11.4 (1.12×); max size 4096: 15.1→13.0 (1.16×).
>
> - We'll include Muon in the revision. Fig 6 does not present Muon since the performance of Muon by Jordan is really close to PolarExpress. In the revision, we will add more experiments for GPT-2 Medium (350M) and GPT-2 Large (774M) with PRISM, Muon, PolarExpress, AdamW. Preliminary GPT-2-Large (1B tokens, 0.2 epochs) results: AdamW, 6.657; PolarExpress (NS5), 4.129; Muon (NS5), 4.247; PRISM-5, **4.031**.
>
> - PRISM is even more effective for training accuracy! Validation accuracy (Fig 5) is more indicative of model quality. We’ll add plots of training accuracy to our paper. Here’s how training accuracy evolves over wall-clock training time for ResNet20 on Cifar10:
> | Time (sec) | Eigendecomp | PolarExpress | PRISM |
> |--|--|--|--|
> | 500 | 69.4 | 72.8 | **73.0** |
> | 1000 | 79.0 | 81.7 | **83.3** |
> | 1500 | 84.5 | 85.9 | **87.9** |
> | 2000 | 87.1 | 88.6 | **90.1** |
>
> **W2:** Yes, this greedy approach is essential to adapt to the input matrix spectrum without explicit access. PRISM doesn’t guarantee optimal convergence, but it provides robust acceleration for all inputs due to its adaptivity. In contrast, PolarExpress is optimal under an uncheckable assumption on the smallest singular values, failing to provide consistent speedup over a wide range of inputs. Practically, non-globally-optimal methods can often lead to near-optimal performance, e.g. the scaled Newton method of [2] and QDWH [3] for polarization.
>
> **W3:** The choice of bounds  $[\ell, u]$ was principled, but not empirical. There is a systematic way to pick the bounds for any $d\ge1$ according to a set of easy-to-check conditions. We will provide in-depth selection details in the revised paper. (Due to character limit, we kindly ask the reviewer the check our response to **Reviewer sDQC**, point **W2** for more details.)
>
> **Q1:** The convergence rate of PRISM is independent from the size of the input matrix. It depends solely on the distribution of singular values and most notably on the smallest singular value.
>
> **Q2:** We use 3 PRISM-5 iterations to improve training efficiency, reflecting the trade-off between speed and orthogonalization accuracy. PRISM achieves higher per-iteration accuracy than NS by adapting its polynomial to the current spectrum, so fewer iterations suffice. Empirically, PRISM-5 (3 iters) attains lower residual error than PolarExpress (5 iters):12.7 vs. 23.4 (Frobenius norm error, Fig 6). We’ll also follow [4] and add an experiment to train GPT-2 models using different PRISM and NS iteration numbers to find the most practical and effective number of iterations for PRISM.
>
> **Q3:** In Figs 3-4, right, we plot $\alpha_k$ values for different matrices and $k$, and it shows that $\alpha_k$ is close to the upper bound in the initial iterations. Fixing $\alpha_k$ initially reduces computation with negligible impact. Experiments without this constraint show similar training time and slightly better performance. We will add more experiments and treat early $\alpha_k$ fixing as an optional choice.
>
> ---
> [1] FLEX: A Backbone for Diffusion-Based Modeling of Spatio-temporal Physical Systems
>
> [2] A New Scaling for Newton's Iteration for the Polar Decomposition and its Backward Stability
>
> [3] Optimizing Halley's Iteration for Computing the Matrix Polar Decomposition
>
> [4] Beyond the ideal: Analyzing the inexact muon update.

---

> > ### Author Rebuttal · Reviewer_QRht · 2026-04-01
> >
> > Thank you for the response and for providing new experimental results. Some of my concerns have been addressed, but a few points still remain:
> >
> > - If I understand correctly, PRISM is designed to reduce the **per-iteration** computation time. If that is the case, then, as I noted in the second bullet of W1 in my initial review, one would naturally expect the performance gap relative to the baselines to widen over wall-clock time as training progresses. For this reason, I would be interested in seeing results over a broader time scale than what is currently shown. As the authors mention in the paper, it appears that longer training runs have already been conducted, so it would be helpful to include those plots. If the performance gap does not widen over time, or remains roughly constant, I think that should also be explicitly shown, together with some justification for why this happens.
> > - In the response to the reviewer, the authors discuss preliminary GPT-2 Large results and compare methods at the same epoch. As above, my understanding is that PRISM is primarily intended to reduce per-iteration time. Therefore, if the residual is negligible, then comparing at the same epoch would naturally be expected to yield similar values. If so, it is unclear why PRISM would achieve better performance at the same epoch. If this is indeed the case, then I believe the authors should report this result more clearly and provide an explanation for it.
> >
> > If I have misunderstood any aspect of PRISM, I would appreciate clarification. Otherwise, I believe it would be helpful to include the corresponding plots together with a more detailed explanation.
> >
> >
> > ---
> > **Edited**
> >
> >
> > Thank you for the response and the convincing experimental results. My concerns have been fully addressed during the review and rebuttal period, and I am happy to increase my score to 5.

---

> > > ### Author Response · Authors · 2026-04-06
> > >
> > > We thank the reviewer for following up with two additional questions! We address them below.
> > >
> > > ## On the evolution of performance gap as training progresses
> > >
> > > tl;dr: Some performance gap indeed widens as training progresses, but it is not the gap the reviewer was thinking of. We provide additional results.
> > >
> > > The evolution of performance gap is coupled with the dynamics of training, so a faster per-iteration time does not mean a widening performance gap. In fact, the gap should gradually narrow as the training progresses. This is because the rate at which the loss decreases is diminishing over time (loss decreases most rapidly at the beginning, and then the rate slows down). Therefore, if we train the model long enough, the loss trajectories from different baselines should eventually get closer, rather than having a gap that keeps getting larger. For example, the vertical performance gaps (Figure 5, left) between PRISM and Eigendcomp are 3.4%, 2.4%, 1.8% at wall-clock time 2000, 2500, 3000, respectively. The gap is decreasing as expected.
> > >
> > > On the other hand, there is an alternative presentation of the same results that aligns with the reviewer's intuition. PRISM speeds up training in the sense that it trains the model to reach a certain performance level in the fastest way. In the table below, we list the wall-clock time (in sec) it took each method to reach a target accuracy, when training ResNet20 on Cifar10. This is an alternative presentation of the same results shown in Figure 5, left, and it more aligns with the reviewer's intuition that "some gap that should widen over time". The last two columns show the time gap and the speedup factor for PRISM to achieve the target train accuracy compared to Eigendecomp and PolarExpress, respectively. Observe that **the time gap indeed widens as training progresses**, while the speedup factor stays roughly constant. This widening gap corresponds to the *horizontal* gap between the curves shown in Figure 5. The reviewer was likely thinking of the *vertical* gap between the curves in Figure 5, which should not widen due to the nature of the accuracy trajectories being concave w.r.t. wall-clock time (i.e., diminishing gain in accuracy over time).
> > >
> > > |Train Accuracy | Eigendecomp | PolarExpress | PRISM | Time Gap  | Speedup |
> > > |---|---|---|---|---|---|
> > > |65 | 345.2 | 297.4 | 254.0 | 91.2 / 43.4 | 1.36x / 1.21x |
> > > |70 | 486.2 | 412.0 | 350.2 | 136.0 / 61.8 | 1.39x / 1.23x |
> > > |75 | 732.5 | 593.6 | 549.7 | 182.8 / 43.9 | 1.33x / 1.18x |
> > > |80 | 1045.3 | 882.5 | 729.9 | 315.4 / 152.6 | 1.43x / 1.26x |
> > > |85 | 1647.9 | 1399.7 | 1154.9 | 493.0 / 244.8 | 1.43x / 1.25x |
> > > |90 | 2601.6 | 2228.8 | 1838.0 | 763.6 / 390.8 | 1.42x / 1.24x |
> > > |94 | 4275.8 | 3757.1 | 3065.1 | 1210.7 / 692.0 | 1.39x / 1.23x |
> > >
> > >
> > > ## On comparing loss at the same epoch
> > >
> > > tl;dr: Residual is not negligible. PRISM has a lower residual error and hence leads to a lower loss at the same epoch. Additional results are provided.
> > >
> > > When used inside Muon, the baseline methods are run for only a few iterations (both PolarExpress and the NS variant used in the original Muon are only run 5 iterations). Because of this, compared to the baselines, PRISM does not always lead to faster time per training step. PRISM is better because, within the same amount of time, it computes polar decomposition to a higher accuracy and a lower residual. As shown in our initial response, in Fig 6, PRISM has a residual error of 12.7, whereas the second-best performing PolarExpress has a residual of 23.4. This distinction in residual directly translates into a difference in validation loss when we train for the same epoch for all methods. The key point is that the residual is not negligible. If a method has a lower residual for polar decomposition, then it will generally have a lower validation loss.
> > >
> > > In order to examine how the residual error for polar decomposition affects the training and validation loss, we carried out an additional ablation study using Muon to train nanoGPT 124M for 0.2 epoch. We vary the number of PRISM iterations used inside Muon from 1 to 5. The table below summarizes the average residual error and the validation loss. Observe that a lower residual indeed translates into a lower loss (but there is a tradeoff: moving from Iter 3 to Iter 5 costs us a 20\% increase in total training time and only marginal improvement in loss). For a complete set of plots, including layer-level residual error, we refer the reader to this anonymous repo, [link](https://anonymous.4open.science/r/prism-additional-plots-DAEA).
> > >
> > > | Iterations | 1 | 2 | 3 | 4 | 5 |
> > > |---|---|---|---|---|---|
> > > |Val loss | 4.926 | 4.403 | 4.015 | 3.890 | 3.885 |
> > > |Time (min) | 26.5 | 29.2 | 32.3 | 35.3 | 38.2 |
> > > |Avg residual | 26.87 | 23.20 | 8.97|  1.55 | 0.57 |
> > >
> > > ---
> > > We will add all additional plots (extended time) and details to further improve the presentation of the results. If our response has fully addressed the reviewer's questions, we'd appreciate an increase in score.

---

### Decision · Program_Chairs · 2026-04-30

**Decision:**

Accept (regular)

**Comment:**

# Summary

This paper proposes PRISM, an iterative meta-algorithm that improves and speeds up matrix operations such as matrix square roots and orthogonalization. The proposed method is agnostic and adaptive to the distribution of the spectrum, and uses sketching to allow faster iterations.

# Comments

I carefully read the paper, the reviews, and the discussion. The questions raised by the reviewers were well addressed by the authors. I believe that this paper is a solid contribution to accelerating matrix operations in modern optimizers, so I recommend acceptance. I would like to ask the authors to carefully incorporate the points raised during the discussion, including larger-scale LLM experiments, concrete pseudocode, and the guidelines for the choice of parameters $\ell$ and $u$.